# Discrimination of cell-intrinsic and environment-dependent effects of natural genetic variation on Kupffer cell epigenomes and transcriptomes

Hunter Bennett [1,9], Ty D. Troutman [1,2,3,9] ✉, Enchen Zhou [1,2,9], Nathanael J. Spann[1], Verena M. Link[4], Jason S. Seidman[1], Christian K. Nickl[1], Yohei Abe[1], Mashito Sakai[1,8], Martina P. Pasillas[1], Justin M. Marlman[5], Carlos Guzman[1], Mojgan Hosseini[6], Bernd Schnabl[2,7] & Christopher K. Glass [1,2] ✉

Noncoding genetic variation drives phenotypic diversity, but underlying mechanisms and affected cell types are incompletely understood. Here, investigation of effects of natural genetic variation on the epigenomes and transcriptomes of Kupffer cells derived from inbred mouse strains identified strain-specific environmental factors influencing Kupffer cell phenotypes, including leptin signaling in Kupffer cells from a steatohepatitis-resistant strain. Cell-autonomous and non-cell-autonomous effects of genetic variation were resolved by analysis of F1 hybrid mice and cells engrafted into an immunodeficient host. During homeostasis, non-cell-autonomous *trans* effects of genetic variation dominated control of Kupffer cells, while strain-specific responses to acute lipopolysaccharide injection were dominated by actions of *cis*-acting effects modifying response elements for lineage-determining and signal-dependent transcription factors. These findings demonstrate that epigenetic landscapes report on *trans* effects of genetic variation and serve as a resource for deeper analyses into genetic control of transcription in Kupffer cells and macrophages in vitro.

Genome-wide association studies have been highly successful in linking common forms of genetic variation to risk of disease and providing starting points for identifying new therapeutic targets. However, most of the genetic variants identified by these studies reside in noncoding regions of the genome, limiting their interpretability[1]. Defining the causal variants, the cell types in which they exert their effects and their mechanism of action remain major challenges. Common forms of noncoding genetic variation, including single-nucleotide

[1]Department of Cellular and Molecular Medicine, University of California, San Diego, La Jolla, CA, USA. [2]Department of Medicine, University of California, San Diego, La Jolla, CA, USA. [3]Division of Allergy and Immunology, Center for Inflammation and Tolerance, Cincinnati Children's Hospital Medical Center, University of Cincinnati College of Medicine, Cincinnati, OH, USA. [4]Metaorganism Immunity Section, Laboratory of Host Immunity and Microbiome, National Institute of Allergy and Infectious Diseases, NIH, Bethesda, MD, USA. [5]Division of Allergy and Immunology, Cincinnati Children's Hospital Medical Center, Cincinnati, OH, USA. [6]Department of Pathology, University of California, San Diego, San Diego, CA, USA. [7]Department of Medicine, VA San Diego Healthcare System, San Diego, CA, USA. [8]Present address: Department of Biochemistry and Molecular Biology, Nippon Medical School, Tokyo, Japan. [9]These authors contributed equally: Hunter Bennett, Ty D. Troutman, Enchen Zhou. ✉e-mail: ty.troutman@cchmc.org; ckg@ucsd.edu

polymorphisms (SNPs) and short insertions and deletions (indels), can alter gene expression by changing the sequences of DNA recognition elements for transcription factors within enhancers and promoters. For example, a SNP that reduces the binding of a required transcription factor to a cell-specific enhancer may reduce expression of the corresponding gene in that cell type. Thus, the variant exerts a cell- and gene-specific impact via a form of *cis* regulation. Such variants may have *trans* effects dependent on the affected gene. For example, a noncoding variant affecting expression of a transcription factor in *cis* may drive cell-autonomous *trans* effects on its target genes. Alternatively, a *cis*-acting genetic variant affecting intercellular communication may promote non-cell-autonomous *trans* regulation in other cell types.

At the *cis*-regulation level, substantial progress has been made in linking SNPs and indels responsible for gene regulation to promoters and cell-specific regulatory elements through creation of cell-specific *cis*-regulatory atlases. Dependent on context, ≤80% of allele-specific differences in *cis*-regulatory activity occurs through local variants[2]. By contrast, few studies have quantitatively assessed *trans* regulation in specific cell types of genetically diverse vertebrates in vivo[3]. Notably, *cis*-regulatory variation may only contribute ~30% of heritability in gene expression, with the remaining heritability being driven by *trans* effects[4,5]. An additional barrier is the limited systematic approaches for investigating mechanisms by which genetic variation exerts non-cell-autonomous effects on cellular phenotypes[6].

To address this gap, we integrated two distinct experimental strategies to distinguish cell-autonomous and non-cell-autonomous effects of genetic variation. The first strategy builds on the prior use of genetically diverse macrophages to investigate mechanisms of enhancer selection and activation and gene expression[7–9]. In these studies, macrophages were differentiated in vitro from different inbred strains of mice and used for transcriptomic and/or genomic studies. Roles for ~80 transcription factors were inferred through interrogating the effects of SNPs and indels on transcription factor binding. These roles include macrophage lineage-determining factors, signal-dependent transcription factors and other collaborative binding partners that promote the selection and activation of macrophage-specific enhancers[2]. By establishing an identical differentiation program and cell culture environment, these studies largely excluded possible effects of genetic variation on other cell types and enabled direct assessment of cell-autonomous effects; notably, studies in F1 hybrid mice indicated that ≤90% of strain-specific enhancer activity exhibited *cis* regulation[2].

The second experimental strategy leveraged the roles of enhancers and promoters as sensors and transcriptional effectors of the internal and external signals that establish cellular identity and function[3,10–12]. Signal-dependent changes in gene expression generally result from altered binding and/or function of transcription factors at *cis*-regulatory elements. The selection and activation of enhancers and promoters can be quantitatively measured on a genome-wide scale using assays for open chromatin and histone modifications associated with activity, such as acetylation of histone H3 at lysine 27 (H3K27ac)[13]. Motif enrichment analysis of these dynamically controlled enhancers and promoters enables prediction of the transcription factors underlying gene activation. For example, quantitative analysis of dynamic enhancer landscapes in monocytes undergoing Kupffer cell differentiation in vivo revealed transcription factor motifs linked to validated pathways responsible for Kupffer cell-specific gene expression, including the Notch, transforming growth factor-β and liver X receptor (LXR) signaling pathways[12,14].

We combined these experimental strategies to define cell-autonomous and non-cell-autonomous effects of genetic variation on enhancer activity and gene expression in Kupffer cells. Kupffer cells are the major population of liver-resident macrophages and play important roles in immunity and physiology, including detoxifying gut-derived lipopolysaccharide (LPS) and regulating iron metabolism[15]. These abundant macrophages are suitable for various genomics

assays requiring deep sequencing and thus are a powerful model for evaluating the impact of genetic variation on a tissue-resident macrophage population in vivo. Kupffer cells are implicated in pathological processes of liver diseases, including non-alcoholic steatohepatitis (NASH) development[16,17]. To increase generalizability, we selected three strains of mice with different sensitivities to a NASH-inducing diet for analysis of Kupffer cell gene expression[18]. We provide evidence that strain-specific differences in Kupffer cell transcriptomes and enhancer activity states can be used to infer consequences of genetic variation and their mechanism of action in other cell types that influence Kupffer cell gene expression. These findings suggest a general approach to investigating non-cell-autonomous effects of genetic variation that may be broadly applicable to diverse cell types.

## Results

### Gene–environment interactions affecting gene expression

To establish a model system for analyzing effects of natural genetic variation on Kupffer cell gene expression, we selected three common strains of inbred mice that recapitulate major phenotypic differences observed in human liver disease[18]. Each strain (A/J, BALB/cJ and C57BL/6J) has a publicly available genome[19] and positionally defined SNPs and indels (Extended Data Fig. 1a). Comparing strain susceptibility/resistance to NASH confirmed documented trait segregation for developing obesity, steatosis, steatohepatitis and fibrosis (Extended Data Fig. 1b–e). Transcriptional alteration of total hepatic Kupffer cells was most pronounced in NASH-susceptible strains, with minimal observed changes in cells from NASH-resistant mice (Extended Data Fig. 1f).

These findings prompted us to systematically evaluate the effects of genetic variation on the transcriptomes of Kupffer cells in these mouse strains. In parallel, we generated transcriptomic data for strain-specific bone marrow-derived macrophages (BMDMs) to assess the specificity of genetic variation effects to Kupffer cells. Unsupervised clustering and principal component analysis of these data demonstrated cell type as the dominant determinant of clustering, as expected[20], but each cell type also segregated by strain (Fig. 1a, Extended Data Fig. 2a–f and Supplemental Table 1). Pairwise comparisons of Kupffer cells from each strain indicated 194–362 differentially expressed genes, and 57–80% of genes with differential regulation by strain were uniquely altered in only Kupffer cells or BMDMs (Fig. 1b–d). Examples of transcripts with strain-unique patterns of expression in BMDMs (Fig. 1e–g) included notable regulators of inflammation (*Rgs1, Ifi203* and *Aoah*), responses to lipids (*Ch25h, Abcg1* and *Sdc1*) and polarization (*Arg2* and *Marco*). Likewise, examples of transcripts with strain-unique patterns of expression in Kupffer cells (Fig. 1h–j) included transcriptional (*Atf5*) and inflammatory (*Cd300e, Irak3, Cxcl14, Cd40* and *Mefv*) regulators. Functional grouping of strain-unique gene expression by Kupffer cells linked evolutionary plasticity in programs controlling antigen processing and presentation, chemokine signaling and chemotaxis suppression (Fig. 1k–m)[21]. These results provide functional insights into genetic control of macrophage transcription in a common in vitro model and a natural in vivo environment.

### Effect of natural genetic variation on enhancer landscapes

To investigate the impact of genetic variation on potential transcription factor binding regulatory elements in Kupffer cells, we performed assay for transposase-accessible chromatin followed by sequencing (ATAC-seq)[22]. Pairwise comparisons of ATAC-seq signal by strain (Fig. 2a and Extended Data Fig. 3a) identified from ~1,000 to >7,000 differential ATAC-seq peaks; the number of differential peaks scaled with the level of genetic diversity. As open chromatin does not necessarily reflect the activity of a putative regulatory element, we performed chromatin immunoprecipitation followed by sequencing (ChIP–seq) for H3K27ac, which is highly correlated with enhancer and promoter activity[13]. Approximately 5,000–7,000 ATAC-seq peaks had differential H3K27ac signal by strain (Fig. 2b and Extended Data Fig. 3b). Over

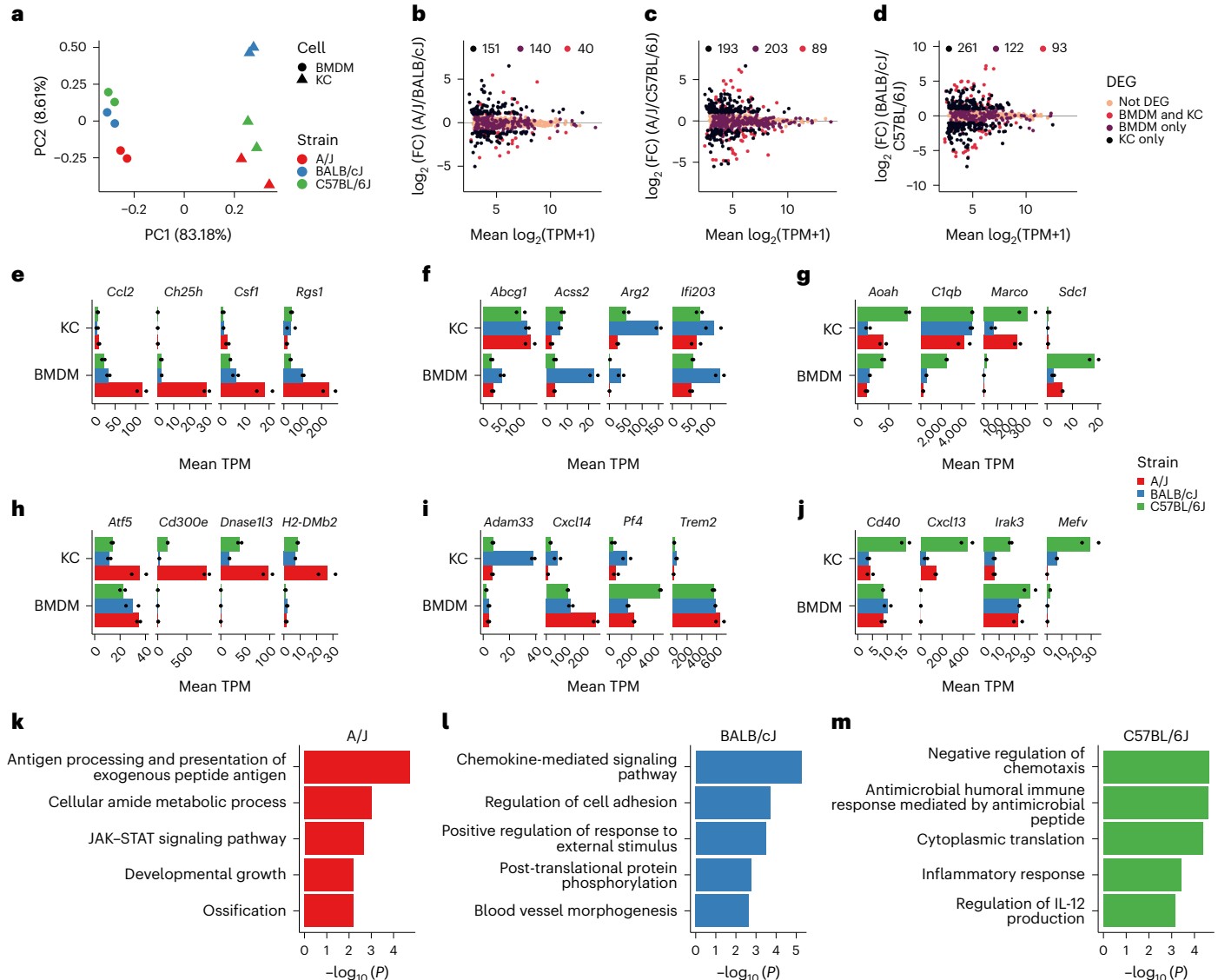

**Fig. 1 | Gene-by-environment transcriptional regulation of primary mouse macrophages. a**, Global principal component analysis for RNA-seq data from BMDMs and Kupffer cells from the indicated strains. Data represent any gene expressed above a TPM threshold of 8; $n = 2$ per group; PC, principal component; KC, Kupffer cells. **b–d**, Comparison of the mean TPM and the DeSeq2 $\log_2$ (fold change) ($\log_2$ (FC)) value for RNA-seq data from Kupffer cells purified from the indicated strains. Differentially expressed genes were identified using a $\log_2$ (fold change) of >1, an adjusted *P* value of <0.05 and a TPM of >8, as identified using DeSeq2 *P* value (Wald's test with multiple testing correction using the Benjamini–Hochberg method) and HOMER (TPM normalization);

DEG, differentially expressed gene. **e–j**, Expression of representative genes in BMDMs (**e**, **f** and **g**) and Kupffer cells (**h**, **i** and **j**) identified as A/J specific (left; **e** and **h**), BALB/cJ specific (middle; **f** and **i**) or C57BL/6J specific (right; **g** and **j**). Genes were defined as strain specific if they were expressed at a significantly higher level in one strain than in both other strains. Data represent the mean TPM. **k–m**, Gene ontology enrichment for strain-specific genes, which were defined as genes with significantly increased expression ($\log_2$ (fold change) > 1, adjusted *P* < 0.05) in one strain compared to both other strains, for example, increased expression in A/J mice relative to BALB/cJ mice and in A/J mice relative to C57BL/6J mice; $n = 2$ per subgroup for transcriptional analysis.

10,000 of the ~66,000 putative regulatory elements identified were defined as under genetic control by analysis of these intersected data (Fig. 2c). Examples of strain-specific ATAC-seq and H3K27ac ChIP–seq signals that correlated with strain-specific gene expression included *Cd300e*, *Trem2* and *Irak3* (Fig. 2d).

To estimate the extent to which local genetic variants contribute to strain-specific differences in ATAC-seq and H3K27ac, we determined the frequency of SNPs and indels within ATAC-defined open chromatin at strain-similar ATAC-seq and H3K27ac peaks in comparison to peaks exhibiting a greater than twofold change in ATAC-seq or H3K27ac signal. Strain-similar peaks exhibited a background SNP/indel frequency of 15–18%, whereas peaks differing by greater than twofold exhibited a

variant frequency of 46–56% (Fig. 2e). The variant frequency increased to 57–64% at a fourfold difference between strains and was similar at an eightfold difference. Overall, ~30–50% of regions' quantitative differences in open chromatin or histone acetylation were associated with nearby variants controlling chromatin accessibility or activity in *cis*.

Motif enrichment analysis of the common set of ATAC-seq peaks exhibiting H3K27ac yielded motifs corresponding to previously established Kupffer cell lineage-determining factors, including PU.1, MAF/MAFB, NF-κB, TFEB/TFEC, LXR, RBPJ and SMADs (Extended Data Fig. 4a,b). To identify potential transcription factors driving strain-specific enhancer selection, we performed motif enrichment of strain-specific enhancers. In each case, PU.1 binding sites were the

most enriched motifs, consistent with the general role of PU.1 in selecting macrophage-specific enhancers (Fig. 2f). Additionally, putative enhancers exhibiting preferential H3K27ac between strains included enrichment for motifs recognizing NFIL3, AP-1 and ETS factors, among others (Fig. 2f), and indicating a likely causal relationship between enhancer activity and the sequence-specific transcription factor(s) in the active strain.

To gain further insight into mechanisms by which SNPs and indels exert local effects on enhancer selection and function, we assessed the quantitative impact of the genetic variation provided by these three strains of mice on open chromatin and H3K27ac using MAGGIE, a motif mutation analysis tool[23]. MAGGIE associates changes of epigenomic features at homologous sequences (for example, enhancer activation or enhancer repression) with motif mutations caused by genetic variation to prioritize motifs that likely contribute to local regulatory function. Although the genetic variation provided by comparing A/J, BALB/cJ and C57BL/6J mice is substantially less than that used in previous MAGGIE applications[9,23], systematic analysis of JASPAR[24] transcription factor binding motifs identified >100 motifs with variants significantly associated with altered ATAC-seq and/or H3K27ac signal (Supplemental Table 2). Many motifs identified by MAGGIE are binding sites for related transcription factors (for example, ETS, AP-1 and IRF families; Fig. 2g). Overall, effects of motif mutations on open chromatin and H3K27ac were highly correlated. Significance of motif mutation enrichments were greater when associated with ATAC-seq than H3K27ac ChIP–seq, likely due to the larger size of the ATAC-seq input set. Mutations affecting PU.1 and related ETS family motifs were the most deleterious for chromatin accessibility and acetylation, consistent with the role of PU.1 as a macrophage lineage-determining transcription factor[2,25–27]. Motif mutations affecting IRF, STAT, AP-1/ATF, CREB, MAF and C/EBP families were also identified as drivers of enhancer selection and function in Kupffer cells. Identifying a motif for LXR/RXR heterodimers was also consistent with its established role as a Kupffer cell lineage-determining factor that participates in chromatin opening and enhancer activation[12]. Not all enriched motif mutations could be associated with corresponding transcription factors expressed by Kupffer cells (for example, ZFP57, ZKSCAN, PAX and NKX; Supplemental Table 2), suggesting their involvement in other biological contexts or the interaction of currently undefined transcription factors with these DNA elements. RBPJ and SMAD both regulate Kupffer cell differentiation[12,14]; however, the SNP and indel abundance affecting these motifs is limited, and analyses did not detect enrichment of motif mutations for these factors associated with altered chromatin accessibility or acetylation. This may reflect a selection pressure to preserve the function of these elements. Collectively, these studies define the effect of natural genetic variation on the enhancer landscapes of Kupffer cells from three strains of mice and support functional roles of major motifs enriched in these regulatory elements.

## Inference of environmental influence on transcription

Sinusoidal endothelial cells and stellate cells regulate the expression and activity of Kupffer cell lineage-determining transcription factors via the *Dll4*–Notch and *Bmp9*–ALK/SMAD pathways[12,14]. To gain insights

into how these and other signaling molecules affect strain-specific transcription in Kupffer cells, we applied NicheNet, a computational model of intercellular signaling built from public data sources of ligand–receptor and intracellular networks[28]. The NicheNet model can infer active ligands in cell–cell communication by comparing expressed ligand–receptor pairs in sender–receiver cells to differentially expressed genes in receiver cells.

Using newly generated RNA-sequencing data (RNA-seq) from hepatocytes, stellate cells and liver sinusoidal endothelial cells (LSECs) from A/J, BALB/cJ and C57BL/6J mice, we determined whether ligands expressed by hepatic cells with cognate receptor expression in Kupffer cells could predict strain-specific Kupffer cell gene expression (Fig. 3a and Extended Data Fig. 5). We also included a selection of hormonal ligands that may alter Kupffer cell transcription via portal blood exposure. NicheNet scored each ligand by correlating receiver cell gene expression predicted as induced by a given ligand to the set of strain-specific differentially expressed genes, and we summarized top-scoring NicheNet ligands (Fig. 3b) and ligand–target gene connections (Fig. 3c). Hepatocyte-derived ligands included ApoE, an apolipoprotein that binds lipoprotein receptors (for example, LDLR and TREM2), which was predicted to induce BALB/cJ-specific Kupffer cell gene expression (for example, *Fads1* and *Cxcr4*; Fig. 3c). LSEC-derived ligands included bone morphogenetic protein 2 (BMP2), a BMP ligand family member, which was predicted to control A/J-specific Kupffer cell gene expression. Predicted niche ligands of LSECs and hepatic stellate cells included proteins encoded by *Adam17* and *App*. *App*, the gene encoding amyloid precursor protein (APP), was linked to C57BL/6J-specific gene expression, including the inflammation response genes *Ccl5* and *Tnfaip3*. *Lep*, encoding the adipokine leptin, was a top-scoring ligand that predicts BALB/cJ-specific Kupffer cell gene expression, and *Lepr*, encoding the leptin receptor, was expressed highest in BALB/cJ Kupffer cells (Fig. 3d). Altogether, these analyses predicted altered expression or activity of several niche ligand–receptor signaling pathways that regulate strain-specific Kupffer cell gene expression.

We investigated the importance of differential expression of *Lepr*, which is expressed highest by Kupffer cells among other cell types in the hepatic niche (Fig. 3d). Endogenous leptin is exclusively derived from adipose tissue and signals via the leptin receptor through phosphorylation and activation of STAT3 (refs. 29,30). Acute intraperitoneal injection of leptin into fasted mice induced detectable STAT3 phosphorylation in liver tissue from C57BL/6J and BALB/cJ mice but not A/J mice (Fig. 3e), which had the lowest *Lepr* expression in Kupffer cells of the assessed strains (Fig. 3d). This result was consistent with the NicheNet-predicted role for leptin in controlling strain-specific differences in Kupffer cell gene expression. The leptin–STAT3 pathway suppressing inflammatory signaling in obese mice[29] and Kupffer cells facilitating the acute effects of leptin on hepatic lipid metabolism[31] suggest this finding's functional relevance. Furthermore, resident (Tim4+) Kupffer cells had significantly lower *Lepr* expression in an experimental NASH model than in control mice, and expression of *Lepr* was minimal in peripheral blood monocytes and recruited monocyte-derived (Tim4−) Kupffer cells in C57BL/6J mice with NASH (Fig. 3f)[32]. *Lepr* is also one of

**Fig. 2 | Local and global effect of natural genetic variation on the Kupffer cell epigenome. a**, Scatter plots of log₂ (tag counts) for ATAC-seq signal at the union set of irreproducible discovery rate (IDR) ATAC-seq peaks across all strains; $n = 4$ per group. **b**, Scatter plots of log₂ (tag counts) for H3K27ac ChIP–seq signal at the union set of IDR ATAC-seq peaks across all strains; tags are annotated with a window size of 1,000 base pairs (bp) centered on the middle of the IDR peak; $n = 3$ per group. **c**, Overlap of active and accessible genomic loci for each strain. Accessible loci are defined as sites with >16 HOMER-normalized ATAC-seq tags. Active loci are defined as sites with >32 HOMER-normalized H3K27ac ChIP–seq tags. **d**, Strain-specific epigenetic signals associated with transcriptional activation. *Cd300e* expression and H3K27ac acetylation of nearby enhancers

are specific to A/J Kupffer cells. *Trem2* is preferentially expressed in BALB/cJ Kupffer cells and is associated with increased acetylation of an intronic enhancer. *Irak3* is preferentially expressed in C57BL/6J Kupffer cells and is associated with C57BL/6J-specific ATAC-seq peaks and increased acetylation of nearby enhancers; kb, kilobases. **e**, Enhancers were categorized into strain similar or strain specific for both ATAC-seq and H3K27ac ChIP–seq data. The table denotes percentages of enhancers at each fold change cutoff that harbor local genetic variation within the 200 bp of the IDR peak. **f**, Motifs associated with strain-specific active enhancers, defined as loci that had strain-specific increases in H3K27ac. **g**, MAGGIE motif mutation analysis on strain differentially accessible and active enhancers.

 

the few genes not induced in monocyte-derived Kupffer cells repopulating the niche following experimental ablation of resident Kupffer cells (Fig. 3g)[12]. Thus, our data indicate that genetic, developmental and environmental factors regulate *Lepr* expression in Kupffer cells.

## *Cis* and *trans* effects of genetic variation on transcription

Next, we performed RNA-seq on the Kupffer cells from the first-generation intercross of C57BL/6J male and BALB/cJ female mice to assess underlying mechanisms for genetic variation on transcription.

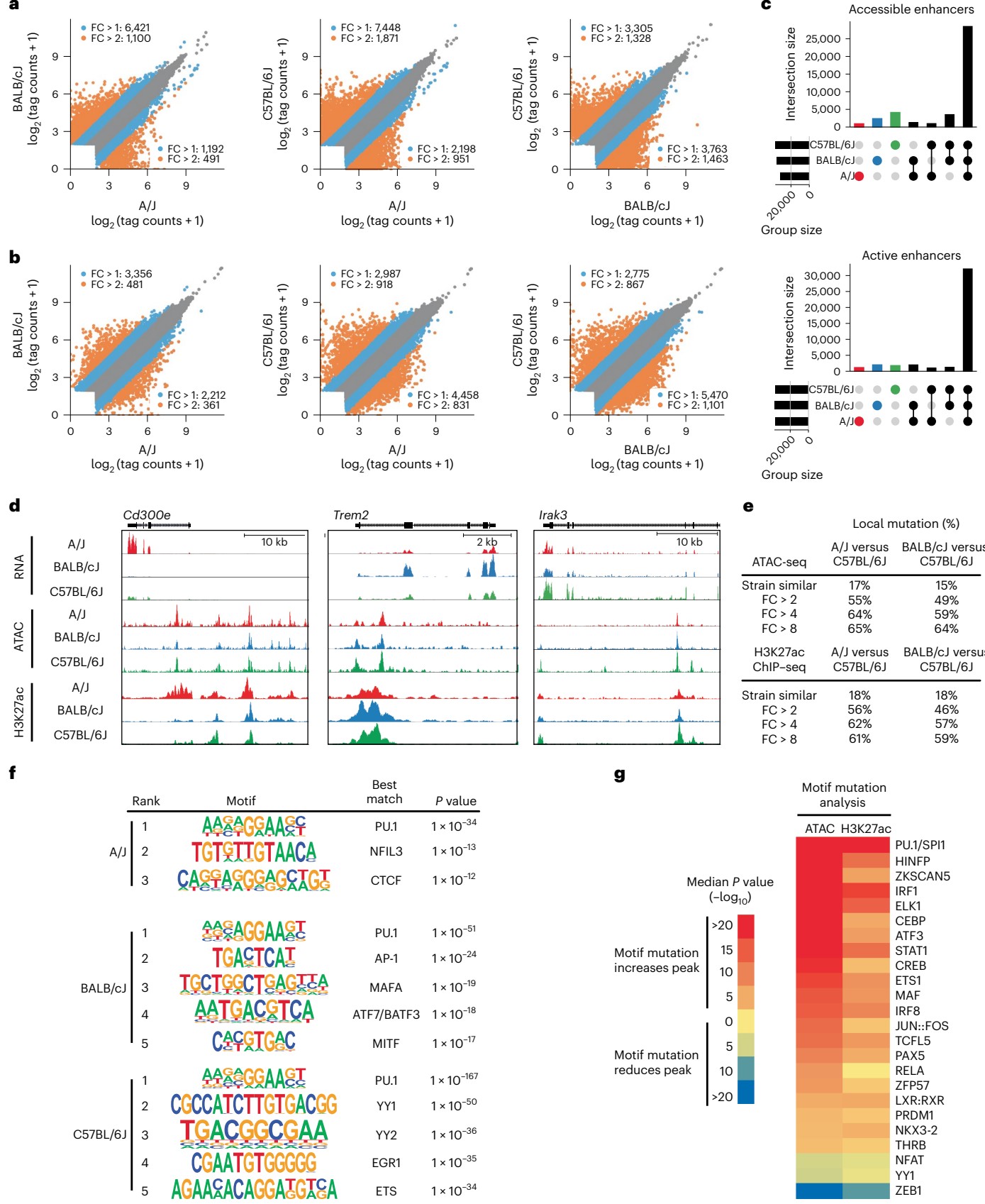

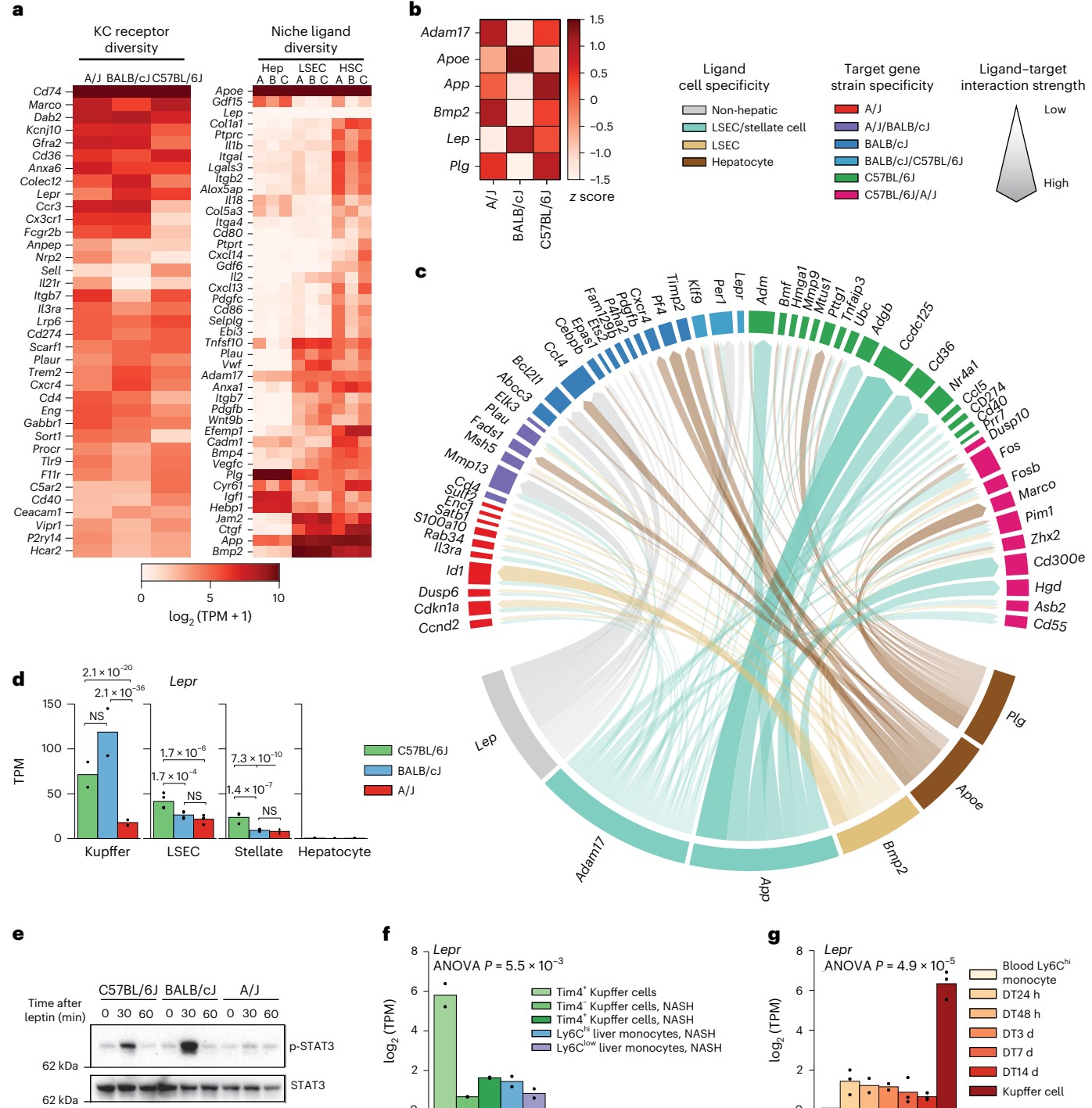

**Fig. 3 | Hepatic Kupffer cell niche differences predicted using network analysis. a**, Gene expression of receptors by Kupffer cells (left) or ligand by niche companion cells (right) from the indicated cell types ('A', 'B' and 'C' annotations on the right indicate expression in A/J, BALB/cJ and C57BL/6J mice, respectively); Hep, hepatocytes; HSC, hematopoietic stem cells. **b**, Top NicheNet ligand activity scores for each strain. Significance was normalized to $z$ score across strains; $n = 2$ samples per group for Kupffer cell RNA-seq, and $n = 4$ samples per subgroup for niche companion cell RNA-seq. **c**, Circos plot demonstrating gene targets of the top six NicheNet ligands from **b**. The width of arrows represents the NicheNet activation score for a given ligand–target gene pair. **d**, Strain-specific expression of the leptin receptor in hepatic cells. $P$ values are derived from DESeq2 (Wald's test with multiple testing correction using the Benjamini–Hochberg method); $n = 2$ samples per group for Kupffer cell RNA-seq, and $n = 4$ samples per subgroup for niche companion cell RNA-seq; NS, not significant.

**e**, Immunoblot assessment of total and phosphorylated STAT3 in liver tissue from mice injected with 1 mg per kg (body weight) leptin via the intraperitoneal route. Data are representative of three experiments. **f**, Expression of the gene encoding the leptin receptor in Kupffer cells from healthy C57BL/6J mice (left) and myeloid cells including macrophages and monocytes isolated from mice fed an Amylin liver NASH (AMLN) NASH-inducing diet for 20 weeks. Data were analyzed by one-way analysis of variance (ANOVA); $P = 5.5 \times 10^{-3}$; $n = 2$ samples per subgroup. **g**, Expression of the gene encoding the leptin receptor in embryonic-derived Kupffer cells (far right) and bone marrow-derived monocytes repopulating the liver at specific time points following depletion of resident Kupffer cells with diptheria toxin. Data were analyzed by one-way ANOVA; $P = 4.9 \times 10^{-5}$; $n = 2$ samples for Ly6C$^{hi}$ monocytes, DT48 h, DT3 d and DT14 d subgroups; $n = 3$ samples for DT24 h, DT7 d and Kupffer cell subgroups.

In CB6F1/J hybrid mice, both parental BALB/cJ and C57BL/6J genomes coexist within a matched extracellular and intracellular environment. To identify allelic biases, we mapped RNA-seq data to each parental genome and compared the levels of perfectly mapped reads spanning mutations between the parental strains, as described previously[2,7,9,33].

We identified 245 genes with significant allelic bias in the F1 hybrid; 83 genes shared gene expression bias toward the same strain in RNA-seq data from parental Kupffer cells (Fig. 4a). We defined *cis/trans* expression patterns by directly comparing the relative expression between parental cells and allelic data[2,3,34,35]. Strain-specific gene expression was classified as *cis* driven or *trans* driven if expression was conserved or not conserved, respectively, at the allelic level. With this approach, we found that *trans* genes outnumbered *cis* genes in healthy Kupffer cells (Fig. 4b), but *trans* genes were associated with smaller expression fold changes in the parental strain (Fig. 4c), mirroring findings in human expression quantitative trait loci studies[5]. Functional studies of expression quantitative trait loci have shown that *trans*-regulatory genetic variants associated with complex traits converge on modules of coexpressed genes sharing common upstream regulators, yielding insights into mechanisms of pathogenesis[5,36,37]. We asked whether this feature of *trans* genes could be used to identify transcriptional pathways driving strain-specific Kupffer cell gene expression. Indeed, C57BL/6J and BALB/cJ *trans* genes were enriched for distinct biological functions, antigen presentation and Toll-like receptor (TLR) signaling in C57BL/6J mice and chemotaxis and wound healing in BALB/cJ mice (Fig. 4d). De novo motif analysis of *trans* gene promoters identified distinct transcriptional regulators in Kupffer cells, including NF-κB in C57BL/6J mice and RXRα in BALB/cJ mice (Fig. 4e).

As *trans* effects might be driven by cell-autonomous and non-cell-autonomous mechanisms, we determined their relative contributions by transplanting bone marrow from C57BL/6J mice, BALB/cJ mice or CB6F1/J F1 hybrid mice into busulfan-conditioned NOD *scid* gamma (NSG) recipients[38]. Busulfan treatment depletes resident Kupffer cells, allowing engraftment of monocyte-derived Kupffer cells from donor progenitors. Both the F0–NSG and F1–NSG transplant Kupffer cells share a similar hepatic extracellular environment. Thus, genes with strain-specific gene expression bias in parental Kupffer cells and equivalent expression in F0–NSG and F1–NSG Kupffer cells are likely driven by environmental differences unique to the parental liver (Fig. 4f). By contrast, donor-derived Kupffer cells from the F1–NSG model contain both parental genomes in a shared cellular and nuclear environment. Consequently, genes exhibiting strain-specific expression in Kupffer cells from parental mice and the F0–NSG transplant model and no significant allelic imbalance in the donor-derived F1–NSG model likely arise from differences in intracellular signaling or transcription factor activity. Regarding strain-specific genes, F0 and F0–NSG mice shared more differential genes than did F1–NSG mice, consistent with the role of the intracellular environment in maintaining a portion of strain-specific gene expression (Fig. 4g and Extended Data Fig. 6).

In both the F1 hybrid and NSG models, most transcripts with allelic bias (71/121; Fig. 4h) were *trans* regulated and likely driven by environmental *trans* effects in the parental hepatic environment that were lost in the models. Many remaining transcripts (42/121; Fig. 4h) were defined as *trans* regulated due to allelic biases in chimeras made with cells from F1 hybrid mice but not cells from parental mice, indicating control by cell-autonomous differences upstream of transcription factor binding. From these collective data, we estimate that *trans*-mediated genetic variation in Kupffer cells is driven 60% by non-cell-autonomous differences in environmental signals and 40% by cell-autonomous differences in signaling or transcriptional activity. Notably, these estimates are based on semiquantitative cutoffs, and transcription is regulated by varying degrees of both *trans* and *cis* effects, but we do not have a model to quantify the relative impact of these effects using F0 and F1 data.

### *Cis* and *trans* effects of genetic variation on enhancers

These findings were consistent with genetically determined differences in the hepatic environment exerting additional regulation on Kupffer cells in vivo, and predicted *trans* effects of genetic variation may be dominant drivers of chromatin landscapes in Kupffer cells. We performed ATAC-seq and ChIP-seq for H3K27ac in Kupffer cells isolated from CB6F1/J F1 hybrid mice and evaluated allelic biases as discriminated by the presence of genetic variants (Extended Data Fig. 3c). As with gene expression, *trans* effects were dominant for both ATAC-seq peaks (442 *cis*, 1,184 *trans*; Fig. 5a) and overlayed H3K27ac ChIP–seq signal (936 *cis*, 1,493 *trans*; DeSeq2: $\log_2$ (fold change) > 1, adjusted $P < 0.05$; Fig. 5a). Overall, allelic bias in H3K27ac ChIP–seq data positively correlated with allelic biases in both ATAC-seq and RNA-seq data, suggesting that allele-specific changes in epigenetic signals and gene expression correlate (Fig. 5b,c).

To identify transcription factors driving *trans* regulation of these effects, we performed motif enrichment analysis of the *trans*-regulated 1,184 ATAC peaks and 1,493 H3K27ac loci with convergent allelic signals in F1 hybrid Kupffer cells. Motifs for ETS factors were preferentially enriched in *trans*-regulated regions of open chromatin specific to BALB/cJ Kupffer cells, whereas motifs for CTCF and IRFs were enriched in open chromatin specific to C57BL/6J Kupffer cells (Extended Data Fig. 7a). Motifs recognized by AP-1 factors, MAF factors and FOXO1 were enriched in ATAC-seq peaks exhibiting higher H3K27ac levels in BALB/cJ Kupffer cells, whereas motifs recognized by IRF, LXR and NF-κB were enriched in H3K27ac regions in C57BL/6J Kupffer cells (Fig. 5d). LXRα was more highly expressed in C57BL/6J Kupffer cells (Extended Data Fig. 7b), potentially explaining the LXRE enrichment in these strain-specific chromatin regions. However, the majority of transcription factors associated with strain-specific ATAC or H3K27ac peaks were similarly expressed between strains (Extended Data Fig. 7b).

To link these findings with transcriptional regulation, we performed de novo motif enrichment in active or open enhancers associated with identified *trans* genes (Fig. 4b). BALB/cJ enhancers associated with *trans* genes were enriched for SpiC, AP-1 (representative loci in Fig. 5f)

---

**Fig. 4 | Cell-autonomous and non-cell-autonomous *trans* interactions contribute to gene expression differences in Kupffer cells. a**, Differential allelic expression in F1 hybrid mice with significantly biased genes colored by strain (left) or allelic expression in F1 hybrid mice with overlayed F0 strain-specific genes (right). The dark-colored dots represent genes with strain-specific bias in F0 mice and allelic bias in F1 hybrid mice. The light-colored dots indicate genes that lost strain specificity in F1 hybrid mice. Data are presented in an 'MA plot' format. On the *x* axes, data depict the $\log_2$-transformed average TPM between samples from the displayed comparison. On the *y* axes, data depict relative expression differences ($\log_2$ (fold change)) calculated using DeSeq2. **b**, Comparison of gene expression ratios using Kupffer cells from parental mice (*x* axis) to Kupffer cells from F1 mice (*y* axis). Genes with maintained expression differences in both comparisons are labeled '*cis*' and are colored red; genes differentially expressed in parental cells and without allelic bias in F1 cells are labeled '*trans*' and are colored light purple; genes expressed similarly in parental data and with allelic bias are labeled 'mixed' and are colored dark purple; genes expressed similarly in both comparisons are labeled 'same' and are colored light beige. **c**, Cumulative distribution of allelic fold change in parental strains associated with *cis* and *trans* genes. **d**, Gene ontology enrichment of BALB/cJ and C57BL/6J-specific *trans* genes. **e**, HOMER de novo motif enrichment in promoters associated with BALB/cJ- or C57BL/6J-specific *trans* genes. **f**, Experimental schematic for chosen strategies used to predict cell-autonomous and non-cell-autonomous patterns of strain-specific gene expression by Kupffer cells. **g**, UpSet plot displaying intersections of BALB/cJ and C57BL/6J strain-specific genes across the parental, F0–NSG transplant and F1–NSG transplant conditions. Vertical bar plots illustrate the number of genes in a set, with colored dots below the plots representing the experiments sharing a given set of differential genes. **h**, Overlap of *trans* genes identified in F1 and NSG models. The *trans* genes were identified as in **b**. NSG *trans* genes were filtered to only consider genes harboring a mutation, allowing discrimination of allelic reads in F1 Kupffer cells.

and MITF motifs, whereas *trans* gene-associated C57BL/6J enhancers were enriched for SpiB, Rel/RelA NF-κB (representative loci in Fig. 5f) and heterodimer PU.1–IRF motifs (Fig. 5e and Extended Data Fig. 7c). These observations implicate strain-specific signaling pathways upstream of AP-1 and factors in BALB/cJ Kupffer cells and strain-specific activity of NF-κB in C57BL/6J Kupffer cells.

### Genetic variation impacts LPS responses

The differential sensitivities of C57BL/6J and BALB/cJ mice to a NASH-inducing diet (Extended Data Fig. 1) implies strain-specific differences in cell-type responses within and outside the liver during dietary intervention. In NASH models, dietary effects on Kupffer cell gene expression may be indirect, as substantial gene expression changes in C57BL/6J Kupffer cells may not occur until after histological changes develop in the liver[39]. To gain insights into how genetic variation affects Kupffer cell responses to an environmental perturbation in vivo, we assessed changes in the transcriptomes and open chromatin landscapes of Kupffer cells from BALB/cJ and C57BL/6J mice 2 h following intraperitoneal injection of LPS, a time point capturing immediate transcriptional consequences of TLR signaling. A total of 3,268

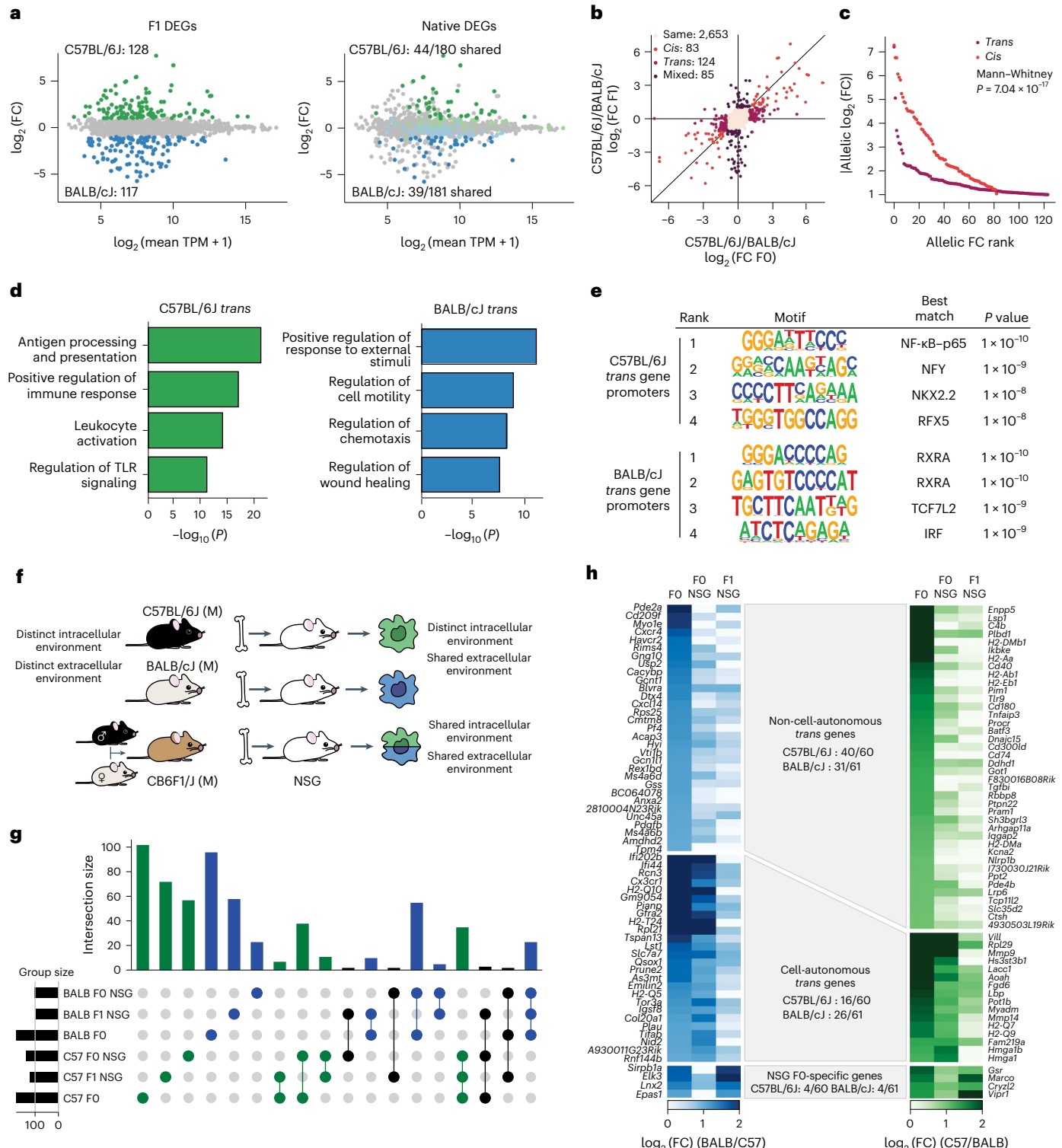

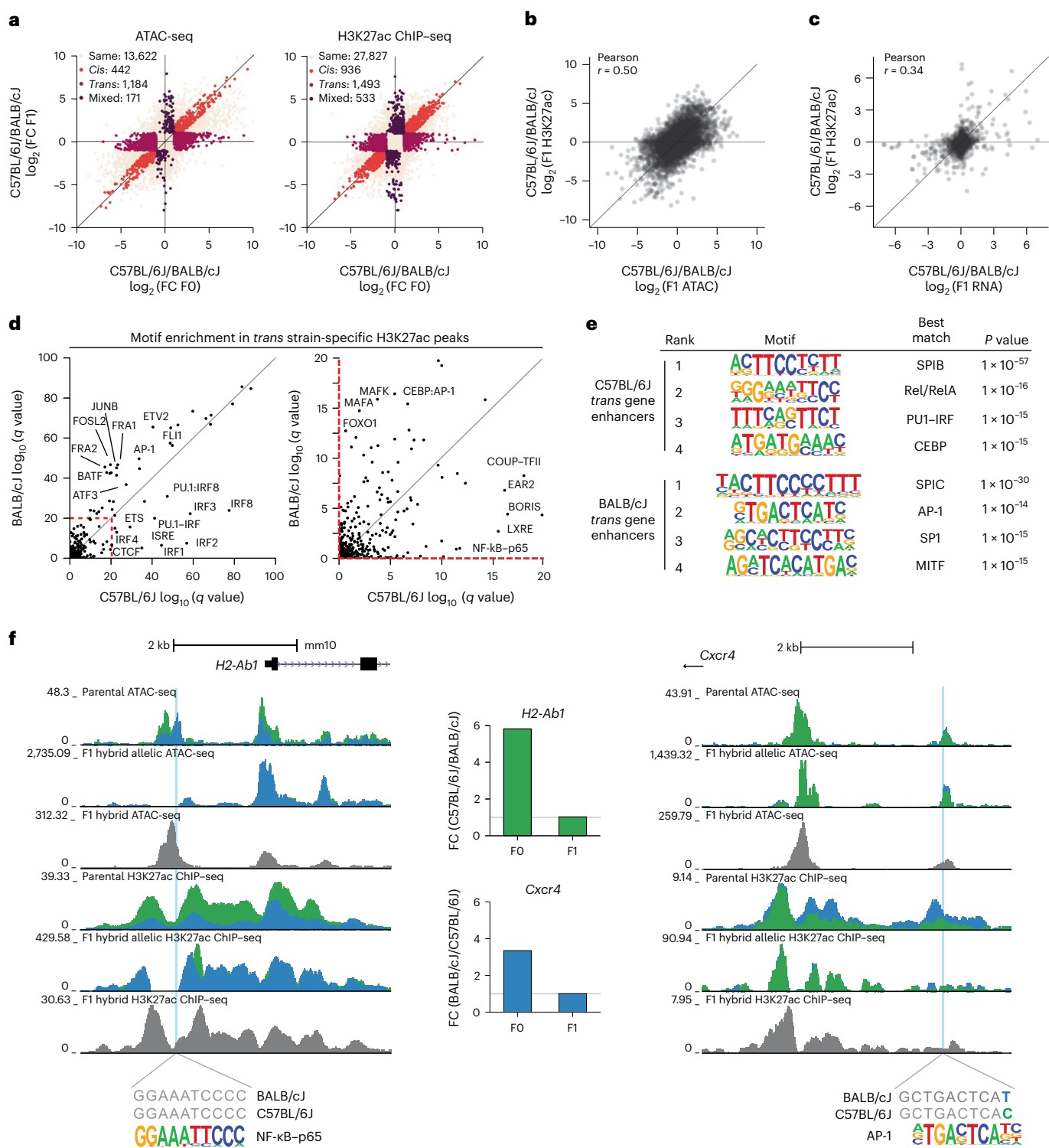

**Fig. 5 | *Cis* and *trans* analysis of epigenetic loci reveals upstream regulators of *trans* transcriptional diversity. a**, Comparison of ATAC-seq (left) and H3K27ac ChIP–seq (right) tag ratios using Kupffer cells from parental mice (*x* axes) and F1 hybrid mice (*y* axis). All IDR ATAC-seq peaks harboring a mutation were considered for analysis. Peaks with maintained ATAC-seq tag differences in both comparisons are labeled '*cis*' and are colored red; peaks with differential ATAC-seq tags in parental cells but not in the F1 model are labeled '*trans*' and are colored purple; peaks with similar ATAC-seq tags in parental data and allelic bias in ATAC-seq tags in F1 mice are labeled 'mixed' and are colored dark purple; peaks with similar ATAC-seq tags in both comparisons are labeled 'same' and are colored beige. **b**, Correlation of allelic ATAC-seq and H3K27ac ChIP–seq reads at IDR ATAC-seq peaks with at least 16 ATAC-seq and H3K27ac reads in at least one

sample. **c**, Correlation of allelic RNA-seq reads and promoter H3K27ac ChIP–seq reads for transcripts expressed at TPM > 4 and promoters with greater than eight H3K27ac ChIP–seq reads in at least one sample. **d**, Enrichment score of known transcription factor motifs in *trans* BALB/cJ-specific enhancers (*y* axis) and in *trans* C57BL/6J H3K27ac enhancers (*x* axis). **e**, De novo motif enrichment at active enhancers (>32 H3K27ac tags) associated with C57BL/6J or BALB/cJ *trans* genes. **f**, Examples of *trans*-regulated epigenetic loci upstream of *trans*-regulated genes. A strain-invariant NF-κB motif lies within a *trans*-regulated C57BL/6J-specific enhancer upstream of the C57BL/6J *trans* gene *H2-Ab1* (left). An AP-1 motif with a mutation outside the core binding sequence lies in a *trans*-regulated BALB/cJ-specific enhancer upstream of the BALB/cJ-specific *trans* gene *Cxcr4* (right). Colored tracks/bars denote BALB/cJ (blue) or C57BL/6J (green) data.

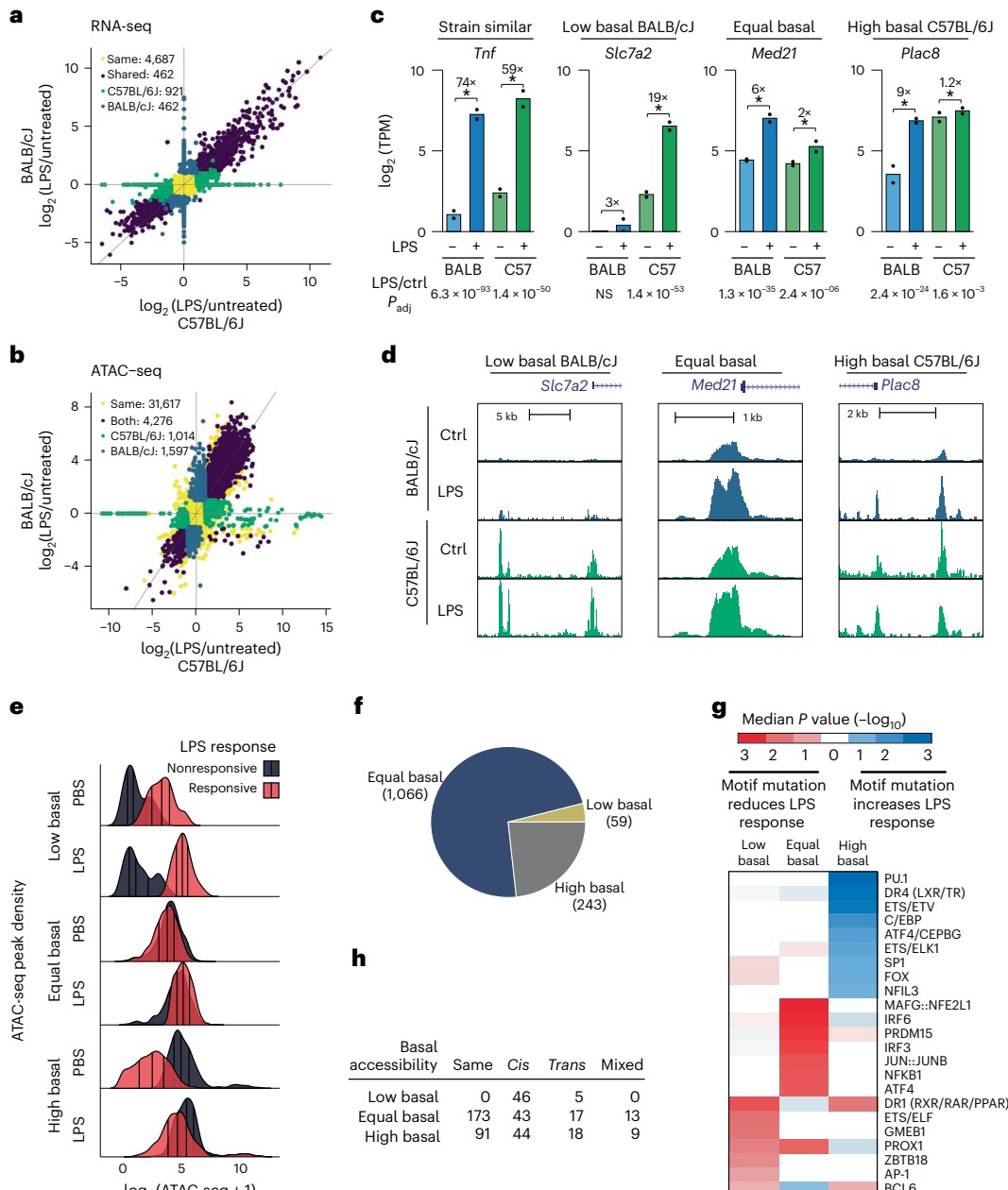

**Fig. 6 | Strain-specific response to LPS determined by *cis*-acting changes in motif affinity. a**, Comparison of transcriptional response to LPS in C57BL/6J (*x* axis) and BALB/cJ (*y* axis) Kupffer cells. Mice were treated with 0.1 mg per kg (body weight) LPS by intraperitoneal injection for 2 h before Kupffer cell isolation. Genes were considered differentially expressed if their expression differed with an absolute log₂(fold change) > 1 and adjusted *P* < 0.05. Genes were separated into C57BL/6J- or BALB/cJ-specific LPS-induced genes (green and blue, respectively) or 'same' if LPS induced equivalent expression changes in both C57BL/6J and BALB/cJ Kupffer cells (purple); *n* = 2 samples per subgroup. **b**, Comparison of ATAC-seq response to LPS in C57BL/6J (*x* axis) and BALB/cJ (*y* axis) Kupffer cells; *n* = 4 samples for basal C57BL/6J and BALB/cJ ATAC-seq data, and *n* = 2 and *n* = 3 for LPS-treated C57BL/6J and BALB/cJ ATAC-seq data, respectively. **c**, Examples of strain-similar and strain-differential transcriptional responses to LPS. The asterisk (*) indicates a DESeq2-adjusted *P* value of <0.05

(Wald's test with multiple testing correction using the Benjamini–Hochberg method) for the comparison between LPS and basal conditions; *n* = 2 mice per group for the LPS-treated RNA-seq data; ctrl, control; *P*ₐdⱼ, adjusted *P* value; BALB, BALB/cJ; C57, C57BL/6J. **d**, ATAC-seq signal at nearby enhancers associated with 'low basal', 'equal basal' and 'high basal' genes. **e**, Distribution of ATAC-seq signal in 'low basal', 'equal basal' and 'high basal' composite enhancers. **f**, Number of enhancers in each basal accessibility category. **g**, MAGGIE motif mutation analysis comparing motif scores between strains associated with each category of enhancer. 'Motif mutation' refers to reduced motif scores in one strain relative to the comparison strain. Heatmap *P*-values refer to the motif scores in the LPS non-responsive strain relative to the responsive strain. **h**, Overlap of three enhancer categories with *cis* and *trans* enhancer analysis performed in CB6F1/J Kupffer cells isolated from mice treated with LPS for 2 h; *P* < 3 × 10⁻²⁴ (Pearson chi-squared *P* value for association of basal states and *cis*/*trans* regulation).

genes were differentially regulated by LPS injection in Kupffer cells from the two parental strains (DESeq2: log₂ (fold change) > 1, adjusted *P* < 0.05, minimum average TPM > 8), with a conserved response of 1,885 genes and strain divergence in 1,383 genes responsive in only one

strain (Fig. 6a and Extended Data Fig. 8a). Regarding LPS-responsive changes in chromatin accessibility (DESeq2: log₂ (fold change) > 1, adjusted *P* < 0.05, minimum average tags > 4), 6,887 ATAC-seq peaks had significant changes in accessibility due to LPS injection in Kupffer

 

cells from the two parental strains (Fig. 6b and Extended Data Fig. 3c,d); 4,276 peaks were similarly LPS responsive in both strains, whereas 2,611 peaks were LPS responsive in only one strain. Interestingly, differences in known motif enrichment at *trans* ATAC-seq peaks were less marked with LPS treatment (Extended Data Fig. 8b), and de novo motif analysis for strain-specific enhancers demonstrated enrichment of AP-1, ETS and IRF motifs in both strains (Extended Data Fig. 8c,d).

Because fold change is determined by the ratio of expression under LPS and control treatment conditions, strain-specific variation in either or both values contribute to detecting strain-specific differences (LPS-responsive strain-similar and strain-specific genes in Fig. 6c). For example, *Tnf* is categorized as a strain-similar gene for exhibiting nearly identical fold changes to LPS injection (~60-fold) in BALB/cJ and C57BL/6J Kupffer cells despite basal and induced levels of *Tnf* being approximately 50% in BALB/cJ compared to in C57BL/6J Kupffer cells. Thus, from a functional perspective, responses to Kupffer cell-derived *Tnf* would be expected to be greater in C57BL/6J than in BALB/cJ mice.

We assessed strain-specific transcriptional and epigenetic mechanisms by partitioning strain-specific genes by basal expression levels (approach from Hoeksema et al.[9]). Strain-specific LPS-responsive genes were sorted into 'low basal' (for example, *Slc7a2*), 'equal basal' (for example, *Med21*) and 'high basal' (for example, *Plac8*) states by comparing Kupffer cell gene expression in untreated BALB/cJ or C57BL/6J mice (Fig. 6c). ATAC-seq peaks for the *Slc7a2*, *Med21* and *Plac8* promoters and nearby enhancers illustrated corresponding low, equal and high basal normalized tag counts, respectively (Fig. 6d). Normalized tag count and absolute number distributions for each ATAC-seq peak category are illustrated in Fig. 6e. Equal, high and low basal enhancers accounted for ~77%, ~17% and ~4%, respectively, of ATAC-seq peaks exhibiting strain-specific LPS responses (Fig. 6f).

Applying MAGGIE to these basal status peak categories revealed clear segregation of motif mutations. Deleterious motif mutations for signal-dependent transcription factors, including IRFs, JUN/JUNB and NF-κB, which have established transcriptional roles in LPS-mediated TLR4 signaling[40–42], were enriched in equal basal ATAC-seq peaks from the non-responsive strain (Fig. 6g). Additionally, MAFG:NFE2L1 were enriched, suggesting previously unrecognized roles in LPS-induced Kupffer cell responses. Conversely, high basal peaks from the non-responsive strain were associated with higher motif scores for lineage-determining transcription factors, including PU.1, ETS and C/EBP family members (Fig. 6g). A DR4 element was the second-most significant motif mutation identified in high basal ATAC-seq peaks. Among nuclear receptors recognizing this motif, LXRα is the most highly expressed in Kupffer cells and an established Kupffer cell lineage-determining transcription factor[10,12,43]. The enrichment of mutations improving the motif scores for Kupffer cell lineage-determining factors in the high basal ATAC-seq peaks is consistent with enhanced binding of these factors under basal conditions. A DR1 element recognized by RXR homodimers and heterodimers was enriched in low basal ATAC-seq peaks, suggesting a function in selecting a small subset of LPS-responsive enhancers; however, this interpretation is limited by the low number of low basal peaks. Finally, we intersected LPS treatment peaks with *cis* regulation, *trans* regulation or mixed regulation with each basal peak category. Discerning parental-specific alleles reduced the strain-specific peak set from 1,261 to 459, where 133 peaks exhibited *cis* regulation, 40 exhibited *trans* regulation, and 22 exhibited mixed regulation (Pearson chi-squared $P < 3 \times 10^{-24}$; Fig. 6h). This indicated that strain-specific LPS responsiveness primarily results from local motif mutations in binding sites for signal-dependent transcription factors.

## Discussion

We characterized the impact of natural genetic variation on gene expression and transcriptional regulatory elements in Kupffer cells derived from three widely used inbred strains of mice that exhibit different sensitivities to diet-induced liver disease. The degree of interstrain variation was comparable to that between a given human individual and the human reference genome[44], suggesting that the magnitude of gene expression effects observed herein will be comparable to common genetic variation effects in human Kupffer cells. Our finding that strain-specific gene expression differences in Kupffer cell gene expression are distinct from those in primary cultured macrophages reinforces that interpreting common genetic variants associated with traits and disease risk requires characterizing cell-specific regulatory landscapes, even for related cell types[45–47].

By acquiring and analyzing transcriptomic data for hepatocytes, stellate cells and endothelial cells in each strain and considering candidate hormonal signaling molecules, we inferred ligand–receptor pairs predicted to contribute to strain-specific differences in Kupffer cells. Among these, we validated the prediction of enhanced leptin signaling in livers of BALB/cJ mice using the preferential expression of *Lepr* in BALB/cJ Kupffer cells. Further investigating the physiologic significance of leptin signaling in Kupffer cells is of interest given that Kupffer cells facilitate the acute effects of leptin on hepatic lipid metabolism[31]. Intriguingly, *Lepr* is one of the few genes that do not become expressed in monocyte-derived cells engrafting the liver following loss of embryonic Kupffer cells, raising questions of whether its expression requires embryonic origin and whether its function in Kupffer cells is limited to early life before hematopoietic stem cell-derived Kupffer cell replacement.

NSG transplant and F1 hybrid models provided evidence that most strain-specific differences in gene expression, open chromatin and histone H3K27ac resulted from *trans* effects of genetic variation, agreeing with previous work showing that *trans* effects mediate 70% of the heritability of gene expression[4,5] (Extended Data Fig. 9). By performing cell-type-specific analysis, we further demonstrated that >50% of *trans* effects were mediated by extracellular factors. A major objective was to investigate whether strain-specific differences in enhancers could be used to infer mechanisms underlying *trans* effects of genetic variation. From analyzing ATAC-seq and H3K27ac ChIP–seq peaks exhibiting criteria for *trans* regulation in F1 hybrid mice, we detected clear biases for AP-1 and MAF family activation in BALB/cJ Kupffer cells and IRF, NF-κB and LXR activity in C57BL/6J Kupffer cells. Apart from increased expression of MAF in BALB/cJ Kupffer cells and LXRα in C57BL/6J Kupffer cells, these differential motif enrichments cannot be explained by differences in mRNA levels for the corresponding factors. Therefore, the strain-specific differences in *trans*-regulated enhancer landscapes are most consistent with differences in extracellular environmental signals and intracellular signaling pathway activity. Although the genomic markers selected distinguish open and active chromatin, further insight into strain-specific genetic regulation could be gained with markers of poised or primed chromatin, such as H3K4me1 or H3K4me2.

Although we found a dominant role of *trans* regulation in determining basal states of strain-specific gene expression and transcriptional regulatory elements, it was possible to exploit *cis* effects to establish the functional significance of motifs required for enhancer selection and function, as previously documented in BMDMs[2]. Furthermore, we found that *cis* regulation predominated in the strain-specific responses to LPS. By segregating strain-specific responses according to the effects of genetic variation on relative activity under basal conditions, we identified qualitatively different patterns of motif mutations associated with each category. Mutations disrupting binding sites for transcription factors mediating transcriptional responses to LPS were significantly associated with impaired activation, as expected, whereas mutations improving suboptimal binding sites for Kupffer cell lineage-determining factors resulted in high constitutive basal activity and a reduced dynamic range in response to LPS. This latter observation, similar to recent findings in IL-4-treated BMDMs[9], suggests the

 

importance of suboptimized binding sites for lineage-determining factors in conferring low levels of basal activity but enabling selection of primed enhancers receptive to binding and activation by signal-dependent transcription factors.

A key future objective is investigating whether the observed effects of natural genetic variation on Kupffer cell gene expression are linked to liver disease mechanisms. Overall, our findings suggest that interactions of multiple genes and cell types quantitatively contribute to phenotypic differences, corresponding with implications of most genome-wide association studies. However, our findings also indicated that strain-specific pathways correlate with sensitivity or resistance to the NASH phenotype. Notable candidate pathways identified here include signaling through the leptin receptor, which has anti-inflammatory roles in Kupffer cells and is expressed highest by Kupffer cells from NASH-resistant BALB/cJ mice. Additionally, Kupffer cells from the NASH-sensitive C57BL/6J strain had increased *trans*-acting chromatin activity at elements predicted to bind NF-κB, a transcription factor strongly linked to macrophage-mediated pathogenesis in many chronic inflammatory diseases.

Systematically analyzing the transcriptomes and regulatory landscapes of Kupffer cells in A/J, BALB/cJ and C57BL/6J mice, NSG chimeras and F1 hybrid mice revealed dominant *trans* effects of natural genetic variation under homeostatic conditions and dominant *cis* effects on the response to LPS. The resulting transcriptomic and genomic datasets are valuable resources for further understanding the mechanisms by which genetic variation affects tissue-resident macrophage phenotypes and susceptibility to diseases in which macrophages play pathogenic or protective roles.

## Quantification and statistical analyses

### Sequencing data analysis

Data collection and analysis were not performed blind to the conditions of the experiments.

**Preprocessing and mapping.** Sequencing data were assessed for quality using fastqc and unsupervised principal component analysis. ATAC-seq and ChIP–seq data were mapped using Bowtie2, and RNA-seq data were mapped using STAR[48,49]. ATAC-seq data were trimmed to 30 bp to remove sequencing adapters, which improved mapping efficiency. Strain-specific pseudogenomes for BALB/cJ and A/J cells were generated by replacing invariant positions of mm10 sequence with alleles reported in the Mouse Genome Project strain-specific VCF files. Importantly, this strategy allows for mapping of SNPs and indels but does not consider larger structural variants present in BALB/cJ and A/J mice. While these structural variants may contain regulatory elements, the number of structural variants is two orders of magnitude less than the number of SNPs and indels captured by the pseudogenome alignment strategy[19]. mm10 was used as the C57BL/6J strain-specific genome. Samples from parental strains of mice were mapped to the strain-specific genome. Mapped reads were shifted to the chromosome coordinates of the mm10 genome build using MARGE.pl shift with -ind set to balbcj or aj for reads mapped to the BALB/cJ or A/J genome, respectively[33].

For samples from CB6F1/J samples, reads were mapped to the mm10 and BALB/cJ genome builds. The BALB/cJ mapped reads were then shifted to the mm10 build with MMARGE, as described above. Perfectly mapped reads spanning genetic mutations between BALB/cJ and mm10 were identified using the MMARGE.pl allele_specific_reads command with -ind set to BALB/cJ and a second time with -ind set to mm10, resulting in two SAM files for each biological sample: one SAM file containing reads perfectly mapped to the mm10 genome that spanned known DNA sequence differences relative to the BALB/cJ genome and a second SAM file containing reads perfectly mapped to the BALB/cJ genome that spanned known DNA sequence differences relative to the reference mm10 genome.

**ATAC-seq analyses.** Strain-specific ATAC-seq SAM files were used to generate HOMER tag directories, and allelic (irreproducible discovery rate) IDR peaks were identified using each biological replicate. ATAC-seq tags were quantified for differential peak analysis by annotating merged IDR peaks with ATAC-seq tag directories using the HOMER command 'annotatePeaks.pl' with parameters -size 1000 -raw[25]. ATAC tag counts were quantified for visualization in heat maps using 'annotatePeaks.pl' with the following parameters: -size 1000 -norm 1e7 (ref. 25). Alterations in allelic signals from pooled IDR peaks were detected using DeSeq2 and required the following thresholds: minimum normalized average tag depth > 16, absolute $\log_2$ (fold change) > 1 and adjusted $P$ < 0.05. Data quality metrics for ATAC-seq libraries are provided in Supplemental Table 4, and DESeq2 results for ATAC-seq experiments included in this study are provided in Supplemental Table 5.

**ChIP–seq analyses.** In F0 mice, H3K27ac ChIP–seq tags were quantified for differential peak analysis by annotating merged ATAC-seq peaks with ChIP–seq tag directories using the HOMER command 'annotatePeaks.pl' with parameters -size 1000 -raw[25]. H3K27ac tag counts were quantified for visualization in heat maps using 'annotatePeaks.pl' with the following parameters: -size 1000 -norm 1e7 (ref. 25). In F1 mice, strain-specific SAM files were used to generate strain-specific H3K27ac tag directories that only contained perfectly aligned reads spanning a mutation between the intercrossed strains. H3K27ac reads spanning mutations were aggregated over IDR peaks using a 1,000-bp window centered on the IDR peak. Alterations in allelic tag counts from pooled peaks were detected using DESeq2 with the following thresholds: normalized average tag depth > 16, absolute $\log_2$ (fold change) > 1 and adjusted $P$ < 0.05. Data quality metrics for H3K27ac ChIP–seq libraries are provided in Supplemental Table 4, and DESeq2 results for H3K27ac ChIP–seq experiments included in this study are provided in Supplemental Table 5.

**RNA-seq analyses.** Gene expression data were quantified using the HOMER command analyzeRepeats. Raw count data were aggregated using the following parameters: rna mm10 -condenseGenes -count exons -noadj. TPM count data were aggregated using the following parameters: rna mm10 -count exons -tpm[25]. TPM values were matched to the isoforms with the highest raw count values. Only genes with an average expression level of >8 TPM were considered for differential gene analysis. Differentially expressed genes were identified using DESeq2 with betaPrior set to TRUE[50]. DESeq2 results for RNA-seq experiments included in this study are provided in Supplemental Table 5. The *trans* gene ontology analysis was performed using HOMER findGO.pl[25]. All other gene ontology enrichment analyses were performed using Metascape[21].

**Motif enrichment analysis.** Enrichment of known transcription factor binding motifs in ATAC-seq peaks was performed using HOMER. DNA sequences associated with peaks containing no detectable difference in LPS responsiveness in both BALB/cJ and C57BL/6J cells were used as the background for enrichment analysis of IDR ATAC-seq peaks from F0 Kupffer cells. A randomly generated GC-matched background was used for enrichment analysis of allele-specific IDR ATAC-seq peaks from F1 Kupffer cells. Motifs were selected for visualization if the probability of enrichment over background had a $q$ value of <0.05 in only one strain or allele. Known motif enrichment analysis results for experiments included in this study are provided in Supplemental Table 6.

**MAGGIE analysis.** MAGGIE analysis was performed with modifications to allow analysis of three pairwise comparisons[23]. For each strain, positive sequences (either associated with increased ATAC-seq signal or increased H3K27ac ChIP–seq signal) were extracted, in addition to the corresponding negative sequence from the comparator strain. Positive and negative sequences from all six possible foreground and background strains were concatenated, maximal motif scores

were calculated for each motif available in the JASPAR database[24], and the net difference in max motif score between positive and negative sequences was calculated. Following calculation of net difference in max motif score for all sequences, a non-parametric Wilcoxon signed-rank two-sided test was used to assign significance to putative motifs by comparing the distribution of motif score differences to a null distribution centered on zero.

**NicheNet.** NicheNet is a computational model built on publicly accessible cellular network data (KEGG, PathwayCommons and ENCODE) that scores the ability of extracellular ligands to induce expression of target genes using network propagation[28]. To assess putative strain-specific ligand activity, we first filtered the NicheNet ligand–target matrix to only consider ligands in which the following criteria were true:

1. The ligand was expressed by a cell of the hepatic niche within that strain at >10 TPM.
2. The receptor was expressed by Kupffer cells from that strain at >10 TPM.

We also included selected metabolic ligands for which expression data were not available. We did not require that ligands or their receptors be differentially expressed by sender or receiver cells. Target genes were selected to be any gene that had significantly higher expression in a pairwise comparison of that strain ('union' gene set, adjusted $P < 0.05$, log (fold change) > 2 and TPM > 10 expression in Kupffer cells). As background, we considered all genes that were expressed at TPM > 10 in Kupffer cells. The NicheNet ligand activity score was then computed as the Pearson correlation coefficient between the ligand–target score and the binary vector, indicating whether a target gene was differentially expressed. For heat maps, select top-scoring ligands from each strain were aggregated and displayed with ligand $z$ scores. Ligand–receptor interaction scores were displayed for ligand–receptor pairs, with a receptor expressed by Kupffer cells in at least one strain. For the Circos plot analysis, ligand–target interaction scores were displayed as arrow thicknesses linking a ligand to its target gene.

**Statistical analysis**

Data collection and analysis were not performed blind to the conditions of the experiments. No data were excluded from the analyses. No statistical methods were used to predetermine sample sizes, but our sample sizes are similar to those reported in previous publications[12,32]. Fastqc, unsupervised clustering and mean variance analysis were used to verify data quality before further analysis. Genome-wide signals for RNA-seq, ATAC-seq and ChIP–seq were evaluated for differential levels using DESeq2 (ref. 50). The raw $P$ values from DESeq2 for a given peak or gene were corrected for multiple testing using the Benjamini–Hochberg procedure. In some cases, an ANOVA was used to assess expression differences of a priori defined candidate genes between many groups. Association of cis/trans with low, equal and high basal regulation was performed with a Pearson chi-squared test (Python scipy.stats.contingency.chi2_contingency). The effect of strain on NASH CRN and fibrosis score was assessed with a Kruskal–Wallis test (R; Kruskal.test). The effect of NASH or control diets on mass between strains across time was assessed with a linear mixed model fit by maximum likelihood. The hierarchical relationship of repeated measures on individual mice over time was controlled using a random effect term in the model (R; lme4::lmer(Mass ~ 1 + Week * Diet * Genotype + (1 + Week | Mouse), data = df1, REML = F). The significance of main effects or their interactions was assessed using a type III ANOVA with Satterthwaite's method (R; anova(fittedModel)). Within-strain interactions of Week*Diet were assessed by subsetting data on strain and running a reduced linear mixed model (R; lmer(Mass ~ 1 + Week * Diet + (1 + Week | Mouse), data = strain_subset, REML = F).

**Data visualization.** Data were visualized using the University of California Santa Cruz genome browser[51] and custom R and Python scripts. UpSet plots were generated using the UpSetPlot Python package[52].

**Reporting summary**

Further information on research design is available in the Nature Portfolio Reporting Summary linked to this article.

## Online content

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

## Methods

### Mice

A/J, BALB/cJ, C57BL/6J and CB6F1/J strains of *Mus musculus* used in this study were sourced directly from Jackson Laboratories, with the exception of C57BL/6 (lab-maintained) background mice used in NASH experiments. Immunodeficient NOD *scid Il2rg*^null mice were obtained from the University of California San Diego Moore's Cancer Center. Mice were housed in individually ventilated cages under standard conditions at 22 °C with 40 ± 5% relative humidity and a 12-h light/12-h dark cycle. Water and a standard laboratory diet were available ad libitum, unless indicated otherwise. All male animals were used in these studies given the sex-specific response of laboratory mice to NASH-inducing diets[53]. Eight- to 12-week-old mice were used for all experiments. All animal maintenance protocols and procedures performed were approved by the University of California San Diego Animal Care and Use Committee in accordance with an approved animal study protocol meeting AALAC standards.

### Bone marrow chimeras

NSG mice were conditioned with the myeloablative agent busulfan at 25 mg per kg (body weight) for 2 consecutive days, as previously described[54]. On the third day, mice were engrafted via retro-orbital injection (BALB/cJ and C57BL/6J mice) or tail vein injection (CB6F1/J mice) with magnetically enriched, lineage-negative hematopoietic stem cells (Miltenyi) from BALB/cJ, C57BL/6J or CB6F1/J hybrid donors. Engraftment efficiency was monitored in peripheral blood after 4 and 8 weeks. Livers from chimeric recipients were collected 4 months after transplant, and graft-derived Kupffer cells were purified by fluorescence-activated cell sorting (FACS). In BALB/cJ chimeras, viable graft-derived Kupffer cells were distinguished by major histocompatibility complex haplotype as CD146^−, F4/80^+, Cd11b^+, H2-D^b− (KH95) and H2-D^d+ (34-2-12). In C57BL/6J chimeras, graft-derived cells were distinguished as H2-K^d− (SF1-1.1) and H2-K^b+ (AF6-88.5). Kupffer cells from each group (*n* = 3) were used downstream for RNA-seq.

### NASH model diets

Mice were fed for up to 30 weeks with a NASH model diet (Research Diets, D09100301) composed of 40 kcal% from fat, 20 kcal% from fructose and 2% cholesterol by mass, or a custom defined control diet (Research Diet, D15100601) composed of 10 kcal% from fat with 50 g of inulin (a dietary fiber) per 4,057 kcal.

### Histology and pathologic scoring

Samples from NASH and control mouse livers were incubated at 37 °C in 1% paraformaldehyde for 24 h, paraffin embedded and sectioned by the University of California San Diego Histology Core. Sections were stained with hematoxylin and eosin or picrosirius red to evaluate for steatosis and fibrosis, respectively. Samples were scored by a board-certified pathologist blinded to the sample group using the NASH Clinical Research Network (CRN) scoring and fibrosis scoring systems[55].

### In vivo response to leptin

Leptin stocks were prepared in Tris-HCl (pH 8.0) buffer and diluted to 0.333 mg ml^−1 in PBS for intraperitoneal injection at 1 mg per kg (body weight) into overnight fasted mice (18:00 to 06:00 h). At the indicated times, mice were humanely killed by $CO_2$, and liver tissues were removed, minced and homogenized in lysis buffer (50 mM HEPES-KOH (pH 7.9), 150 mM NaCl, 1.5 mM $MgCl_2$, 1% NP-40, 1 mM phenylmethylsulfonyl fluoride (Sigma-Aldrich), protease inhibitor cocktail (Sigma-Aldrich) and PhosSTOP (Roche)). Lysates were sonicated by ultrasound homogenizer (Bioruptor, Diagenode) for 10 min at 4 °C and centrifuged for 10 min at -20,000*g* at 4 °C. The supernatant was used as tissue homogenate for immunoblotting. After a protein assay (Bio-Rad Laboratories), the homogenate was

boiled at 95 °C for 5 min in NuPAGE LDS Sample Buffer (Thermo Fisher Scientific) with NuPAGE Sample Reducing Agent (Thermo Fisher Scientific), subjected to SDS–PAGE and transferred to immobilon-P transfer membranes (Merck Millipore). Immunodetection was performed with anti-phospho-STAT3 (Tyr 705; Cell Signaling Technology, 9145; 1:2,000) or anti-STAT3 (Cell Signaling Technology, 9139; 1:2,000), bound antibodies were visualized with peroxidase-conjugated affinity-purified donkey anti-mouse or anti-rabbit IgG (Dako) using Luminate Forte Western HRP Substrate (Merck Millipore), and luminescence images were analyzed with a ChemiDoc XRS+ System (Bio-Rad Laboratories).

### In vivo response to LPS

Littermate male mice were fasted overnight and randomized to treatment with an intraperitoneal injection of 0.1 mg per kg (body weight) *Escherichia coli* O114:B4 LPS (Sigma) or no injection. After 2 h, mice were killed by $CO_2$ exposure, and transcription was halted by hepatic perfusion with flavopiridol (1 mM). Liver tissue was digested in situ with Liberase in the presence of flavopiridol, and immunolabeled Kupffer cells were purified by FACS[12,32].

### Hepatocyte preparation

Hepatocytes were prepared by perfusion digestion in a retrograde fashion through the inferior vena cava to the portal vein. In brief, livers were blanched with clearing buffer (HBSS + 10 mM HEPES) and digested with a collagenase solution (HBSS supplemented with 0.3 mg ml^−1 collagenase D, 10 mM HEPES and one protease inhibitor cocktail complete-EDTA free tablet per 50 ml) at 39 °C. Livers were perfused for 18 min at 5 ml min^−1. The perfusion and digestion steps were performed in the presence of 1 mM flavopiridol to offset transcriptional changes associated with digestion. After digestion, individual livers were gently dissociated using forceps in 20 ml of Medium 199 supplemented with 5% fetal bovine serum (FBS) and penicillin/streptomycin/gentamicin. Crude hepatocyte preps were carefully strained through a 100-μm strainer into a 50-ml tube. An equal volume of isotonic Percoll (90% Percoll and 10% 10× HBSS) was added, followed by gentle mixing and centrifugation at 100*g* for 7 min at 4 °C. The supernatant was discarded, and the cells were gently resuspended with 50 ml of Medium 199 and centrifuged at 100*g* for 2 min at 4 °C. Hepatocytes were gently resuspended in Medium 199 and counted.

### Hepatic non-parenchymal cell preparation

Non-parenchymal cells from digested livers were prepared as previously described[12,32,56]. In brief, livers were retrograde perfused for 3 min at a rate of 5–7 ml min^−1 through the inferior vena cava with HBSS without $Ca^{2+}$ or $Mg^{2+}$ supplemented with 0.5 mM EGTA, 0.5 mM EDTA and 20 mM HEPES. Perfusions were then switched to 40 ml of a digestion buffer (held at 37 °C) comprised of HBSS with $Ca^{2+}$ and $Mg^{2+}$ supplemented with 0.033 mg ml^−1 Liberase (Roche), 20 mg ml^−1 DNase I (Worthington) and 20 mM HEPES. Livers were then excised, minced and digested for an additional 20 min in vitro at 37 °C with gentle rotation in 20 ml of fresh digestion buffer. The perfusion and digestion steps were performed in the presence of 1 mM flavopiridol to offset transcriptional changes associated with digestion. After tissue digestion, cells were passed through a 70-μm cell strainer, and hepatocytes were removed by two low-speed centrifugation steps at 50*g* for 2 min. Non-parenchymal cells in the supernatant were further separated from debris by pelleting for 15 min at 600*g* in 50 ml of 20% isotonic Percoll (Sigma-Aldrich) at 37 °C. Cells were then washed from Percoll-containing buffer, suspended in 10 ml of 28% OptiPrep (Sigma-Aldrich) and carefully underlaid under 3 ml of wash buffer. The resulting gradient was centrifuged at 1,400*g* for 25 min at 4 °C with no break, and cells enriched at the interface were saved and subjected to isotonic erythrocyte lysis. Cells were washed after erythrocyte lysis and immediately purified by cell sorting.

## Cell sorting and flow cytometry

Hepatic non-parenchymal cells were labeled with fluorescent antibodies, and desired cell populations were purified using a Beckman Coulter Mo-Flo Astrios EQ configured with spatially separated 355-, 405-, 488-, 561- and 642-nm lasers. Kupffer cells were defined as 355:494/20$^{lo}$, SSC$^{lo}$, CD146$^{neg}$, CD45$^{pos}$, F4/80$^{hi}$, CD11b$^{Intermediate}$, live and singlets. LSECs were defined as 355:494/20$^{lo}$, SSC$^{lo}$, CD45$^{neg}$, CD146$^{pos}$, live and singlets. Hepatic stellate cells were defined as 355:494/20$^{hi}$, SSC$^{Intermediate}$, live and singlets.

## BMDMs

Femur, tibia and iliac bones from male mice of 8–12 weeks of age ($n$ = 2 per group) were flushed with DPBS, and red blood cells were lysed using red blood cell lysis buffer (Sigma-Aldrich). Bone marrow cells were seeded in 10-cm non-tissue culture plates in RPMB with 10% FBS, 30% L929 cell-conditioned laboratory-made medium (as a source of macrophage colony-stimulating factor (M-CSF)), 100 U ml$^{-1}$ penicillin–streptomycin (Thermo Fisher Scientific) and 16.7 ng ml$^{-1}$ M-CSF (BioLegend). After 2–3 d of differentiation, cells were fed with 16.7 ng ml$^{-1}$ M-CSF. After an additional 2 d of culture, non-adherent cells were washed off with 37 °C PBS, and adherent macrophages were obtained by scraping. Cells were counted, density adjusted with RPMI supplemented with 10% FBS and 100 U ml$^{-1}$ penicillin–streptomycin, seeded into multiwell plates and rested at 37 °C overnight. The following day, macrophages were treated with 100 ng ml$^{-1}$ Kdo$_2$-lipid A (Avanti lipids), a highly purified *E. coli* LPS[57].

## Next-generation sequencing libraries

**ATAC-seq.** Transposase reactions and sequencing libraries were generated as described previously[12,22,56] using 25,000 to 50,000 FACS-purified Kupffer cells. Tagmented DNA was cleaned using Zymo ChIP Clean & Concentrate columns and PCR amplified for 14 cycles using barcoding primers. Libraries were size selected to 175–225 bp using gel excision and purified as described in Texari et al.[58]. For F1 samples, dual-indexed libraries were pooled for a targeted depth of 100 million reads per sample.

**ChIP–seq.** ChIP–seq libraries were generated as previously described[58] with modifications to lysis, immunoprecipitation buffer and washing buffer as described in refs. [32,59]. In brief, FACS-purified cells were fixed with 1% paraformaldehyde for 10 min at 37 °C. Next, 2.625 M glycine was added to 125 mM to quench fixation, and cells were collected by centrifugation with the addition of 0.01% Tween 20 at 1,200$g$ for 10 min at 4 °C. Cells were washed once with 0.01% Tween 20 in PBS and collected by centrifugation at 1,200$g$ for 10 min at 4 °C. Cell pellets were then snap-frozen and stored at −80 °C. For ChIP reactions, cell pellets were thawed on ice and lysed in 80 ml of LB3 (10 mM Tris-HCl (pH 7.5), 100 mM NaCl, 1 mM EDTA, 0.5 mM EGTA, 0.1% deoxycholate, 0.5% sarkosyl, 1× protease inhibitor cocktail and 1 mM sodium butyrate). Lysate was sonicated using a Covaris for 12 cycles with the following settings: time, 60 s; duty, 5.0; PIP, 140; cycles, 200; amplitude, 0.0; velocity, 0.0; dwell, 0.0. Samples were collected, and 10% Triton X-100 was added to a final concentration of 1%. One percent of the sonicated lysate was saved as ChIP input. For each ChIP, aliquots of ~500,000 cells were added to 20 μl of Dynabeads Protein A with 2 μg of anti-H3K27ac (Active Motif) and incubated with slow rotation at 4 °C overnight. The following day, beads were collected using a magnet and washed three times each with wash buffer I (20 mM Tris-HCl (pH 7.5), 150 mM NaCl, 1% Triton X-100, 0.1% SDS, 2 mM EDTA and 1× protease inhibitor cocktail) and wash buffer III (10 mM Tris-HCl (pH 7.5), 250 mM LiCl, 1% Triton X-100, 0.7% deoxycholate, 1 mM EDTA and 1× protease inhibitor cocktail). Beads were then washed twice with ice-cold 10 mM Tris-HCl (pH 7.5), 1 mM EDTA and 0.2% Tween 20. Sequencing libraries were prepared for ChIP products while bound to Dynabeads Protein A[58]. For F1 samples, dual-indexed libraries were pooled for a targeted depth of 100 million reads per sample.

**RNA-seq.** Poly(A) RNA-seq libraries were generated using 50,00 to 100,000 FACS-purified cells stored in lysis/Oligo d(T) Magnetic Beads binding buffer and stored at −80 °C or 500 ng of purified RNA using the Zymo Research Direct-zol RNA microprep kit[60,61]. In brief, mRNAs were enriched by incubation with Oligo d(T) Magnetic Beads (New England Biolabs, S1419S) and fragmented/eluted by incubation at 94 °C for 9 min. Poly(A)-enriched mRNA was fragmented in 2× Superscript III first-strand buffer with 10 mM DTT (Invitrogen) by incubation at 94 °C for 9 min and immediately chilled on ice before the next step. The 10 μl of fragmented mRNA, 0.5 μl of random primer (Invitrogen), 0.5 μl of Oligo(dT) primer (Invitrogen), 0.5 μl of SUPERase-In (Ambion), 1 μl of dNTPs (10 mM) and 1 μl of DTT (10 mM) were heated at 50 °C for 3 min. At the end of incubation, 5.8 μl of water, 1 μl of DTT (100 mM), 0.1 μl of actinomycin D (2 mg ml$^{-1}$), 0.2 μl of 1% Tween 20 (Sigma) and 0.2 μl of Superscript III (Invitrogen) were added and incubated in a PCR machine using the following conditions: 25 °C for 10 min, 50 °C for 50 min and a hold at 4 °C. The product was then purified with RNAClean XP beads according to manufacturer's instructions and eluted with 10 μl of nuclease-free water. The RNA/cDNA double-stranded hybrid was then added to 1.5 μl of Blue Buffer (Enzymatics), 1.1 μl of dUTP mix (10 mM dATP, dCTP and dGTP and 20 mM dUTP), 0.2 μl of RNase H (5 U ml$^{-1}$), 1.05 μl of water, 1 μl of DNA polymerase I (Enzymatics) and 0.15 μl of 1% Tween 20. The mixture was incubated at 16 °C for 1 h. The resulting dUTP-marked double-stranded DNA (dsDNA) was purified using 28 μl of Sera-Mag Speedbeads (Thermo Fisher Scientific) diluted with 20% PEG8000 and 2.5 M NaCl to a final concentration of 13% PEG, eluted with 40 μl of EB buffer (10 mM Tris-Cl, pH 8.5) and frozen at −80 °C. The purified dsDNA (40 μl) underwent end repair by blunting, A-tailing and adaptor ligation, as previously described[25], using indexed barcoding adapters. Libraries were PCR amplified for 9–14 cycles, size selected by gel extraction, quantified using a Qubit dsDNA HS Assay kit (Thermo Fisher Scientific) and sequenced on a HiSeq 4000, NextSeq 500 or NOVA-seq (Illumina) according to the manufacturer's instructions. For F1 samples, dual-indexed libraries were pooled for a targeted depth of 100 million reads per sample.

## Data availability

All data required to evaluate the conclusions in the paper are present in the paper. The datasets generated during and/or analyzed during the current study are available in the Gene Expression Omnibus repository (accession number GSE216164). Processed data can be found on Zenodo (https://zenodo.org/record/7829622). Additional data related to this paper may be requested from the corresponding authors T.D.T. or C.K.G. Source data are provided with this paper.

## Code availability

Python and R scripts for analysis and figure generation can be found on GitHub (https://github.com/HunterBennett/KupfferCell_NaturalGeneticVariation).

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

## Acknowledgements

These studies were supported by NIH grants DK091183 and HL147835 and a Leducq Transatlantic Network grant 16CVD01 to C.K.G. Sequencing costs were partially supported by DK063491. H.B. was supported by the NIH Predoctoral Training Grants T32GM007198, T32DK007202 and F30DK124980. T.D.T. was supported by P30DK063491, T32DK007044, P30DK078392, NRSA T32CA009523 and the Center for Inflammation and Tolerance through the Cincinnati Children's Research Foundation. J.S.S. was supported by an American Heart Association Fellowship (16PRE30980030) and an NIH Predoctoral Training Grant (5T32DK007541). M.S. was supported by the Manpei Suzuki Diabetes Foundation of Tokyo, Japan, and the Osamu Hayaishi Memorial Scholarship for Study Abroad, Japan. This study was also supported by NIH Grant DK120515. We thank S. Hottinger for editorial assistance.

## Author contributions

Conceptualization: H.B., T.D.T., J.S.S., M.S. and C.K.G. Formal analysis: H.B., T.D.T., E.Z., J.S.S., V.M.L. and C.K.G. Investigation: H.B., T.D.T., E.Z., M.S., J.S.S., Y.A., C.K.N., N.J.S., V.M.L., M.P.P., J.M.M., C.G. and M.H. Writing: H.B., T.D.T., E.Z. and C.K.G. Visualization: H.B., T.D.T. and Y.A. Supervision: C.K.G., T.D.T. and B.S. Funding acquisition: T.D.T. and C.K.G.

## Competing interests

C.K.G. is a cofounder, equity holder and member of the Scientific Advisory Board of Asteroid Therapeutics. The other authors declare no competing interests.

## Additional information

**Extended data** is available for this paper at https://doi.org/10.1038/s41590-023-01631-w.

**Correspondence and requests for materials** should be addressed to Ty D. Troutman or Christopher K. Glass.

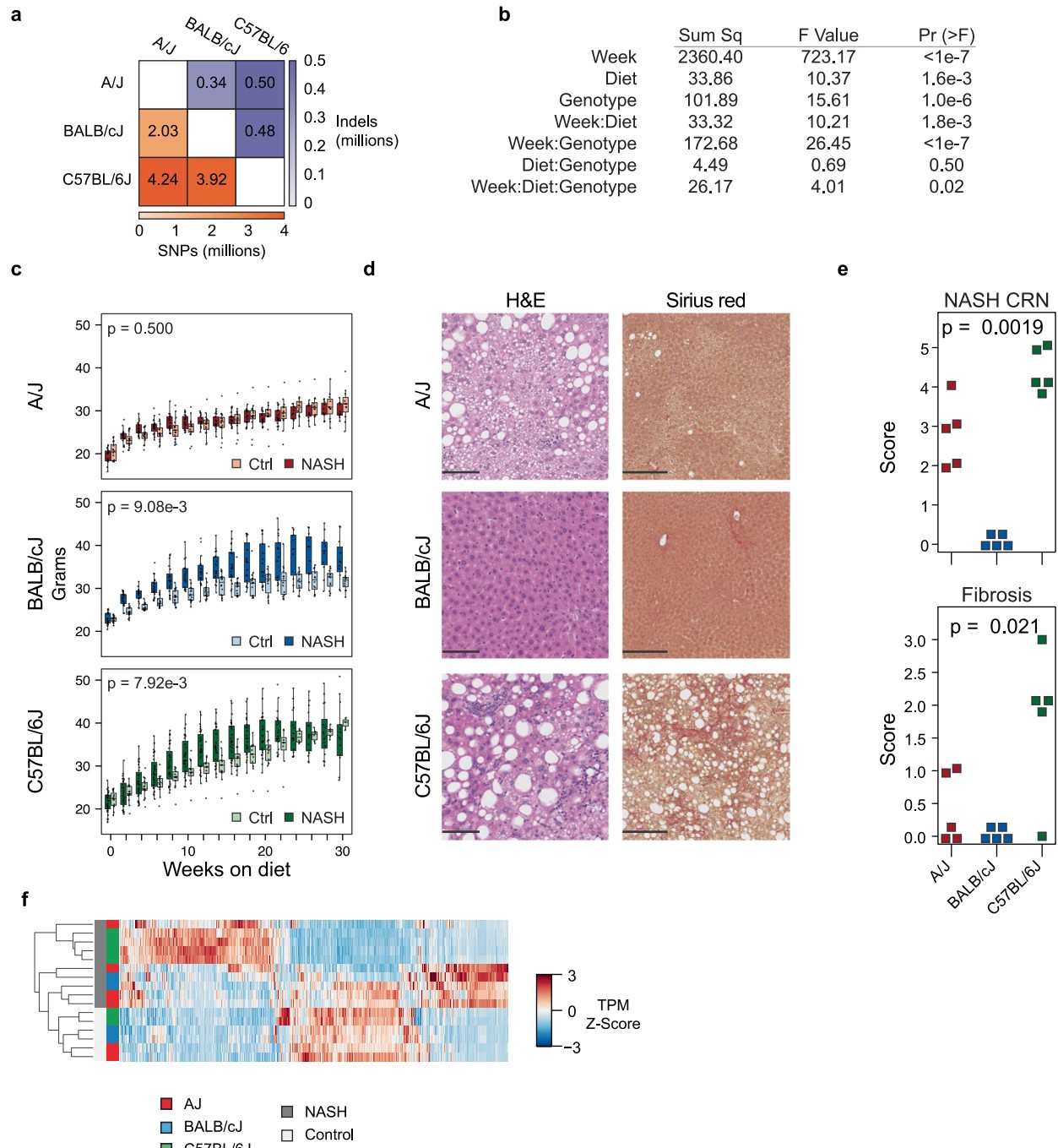

**Extended Data Fig. 1 | Inbred strains vary in susceptibility to a NASH-model diet. a**, Degree of natural genetic variation between three inbred strains used in this study. **b**, Modeled weight gain assessed bi-weekly for each strain fed ad libitum with the Amylin liver NASH model diet, or a matched control diet. Main effects and their interaction were assessed for significance using a linear mixed model fit by maximum likelihood ['lmerModLmerTest'] and assessed by type III analysis of variance using Satterthwaite's method. Sample sizes are shown in Supplemental Table 3. **c**, Weekly weight gain in each strain on the AMLN diet. Box-whiskers boxes denote medians and first and third quartiles, and whiskers denote 1.5*IQR (interquartile range) as per ggplot2 defaults. All individual data points are overlayed in place of outlier values. P-values denote t-statistic probabilities for the diet*time interaction using Satterthwaite's method. Data were modeled as in b by subsetting on strain and reducing the main strain effect. Sample sizes are shown in Supplemental Table 3. **d**, Histopathological evidence of NASH in following 30 weeks of AMLN diet in each strain of mice using hematoxylin and eosin (left) and Sirius red (right) staining of mouse livers. Scale bars denote 100 microns. **e**, Histopathological scoring of NASH (top) and fibrosis (bottom) in each strain of mouse following 30 weeks of AMLN diet. Strain effects were assessed independently for NASH CRN and fibrosis score using a Kruskal-Wallis test (R, kruskal.test). N = 5 samples per group for histopathologic data. **f**, Unsupervised clustering of strain-specific differential genes from Kupffer cells from mice fed control or AMLN NASH-inducing diet, N = 2 per group for control diet Kupffer cells, N = 4 for A/J and C57BL/6J AMLN diet Kupffer cells and N = 2 for BALB/cJ AMLN diet Kupffer cells. Differential genes were defined using pairwise comparisons in DESeq2 with a Log2 Fold Change > 1 and an adjusted p value < 0.05 (Wald's test with multiple testing correction using Benjamini–Hochberg method)FDR adjusted p-value >< 0.05.

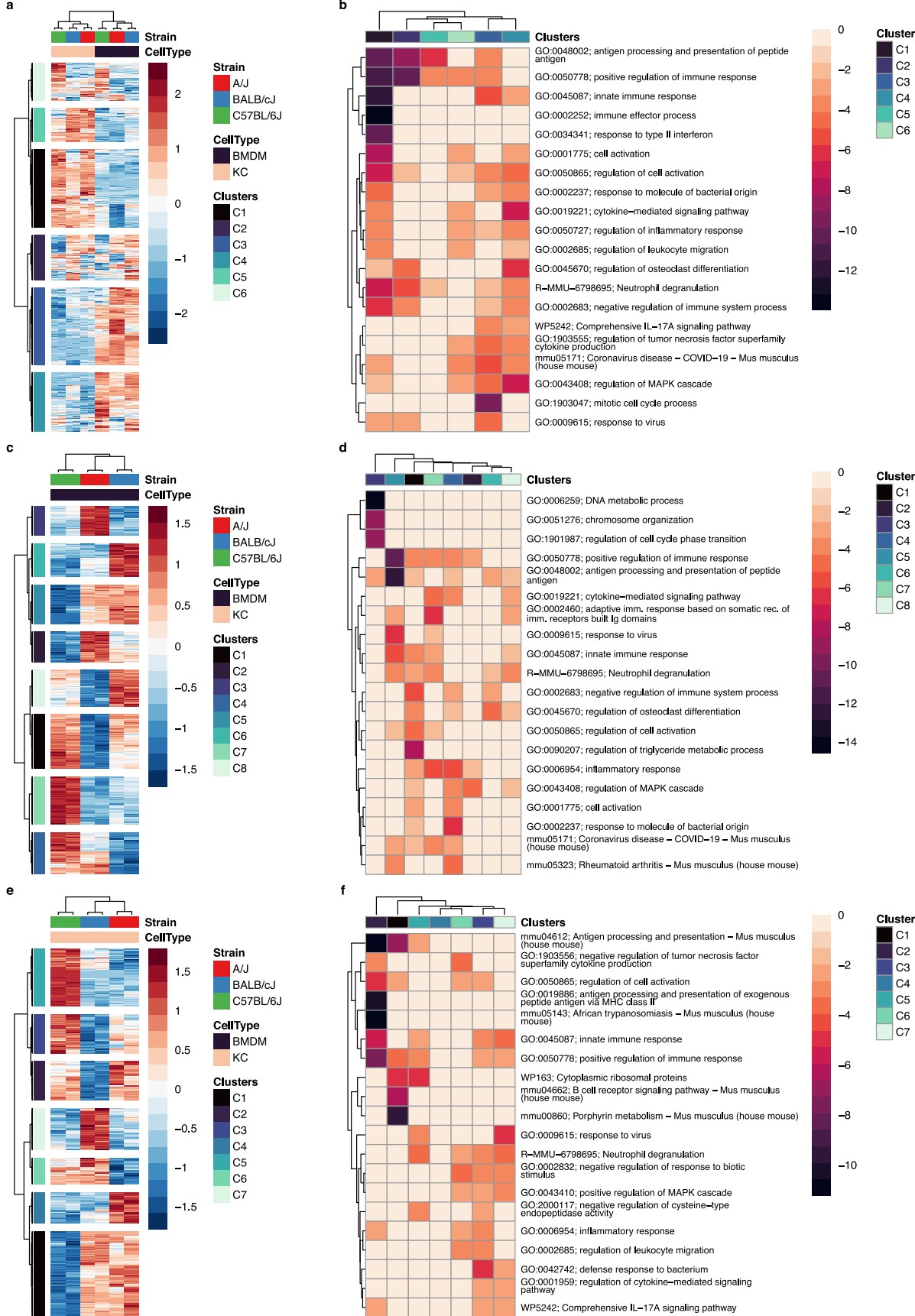

**Extended Data Fig. 2** | See next page for caption.

**Extended Data Fig. 2 | Gene-by-environment control of macrophage transcription. a**, Global comparison of differentially expressed genes between Kupffer cells or bone marrow-derived macrophages (BMDMs) of the indicated strain. Data represent the row z-score of log2(tpm + 1) values. **b**, Gene clusters identified in **a**. Data were subjected to Gene ontology analysis using Metascape.

Data indicate the log10(p-value). **c**,**e**, Global comparison of strain-specific gene expression only in BMDMs, as in **a**. **d**,**f**, Metascape analysis of results from **c** and **e**, as assessed in **b**. Gene lists for individual clusters **a**, **c** and **e** are provided in Supplemental Table 1.

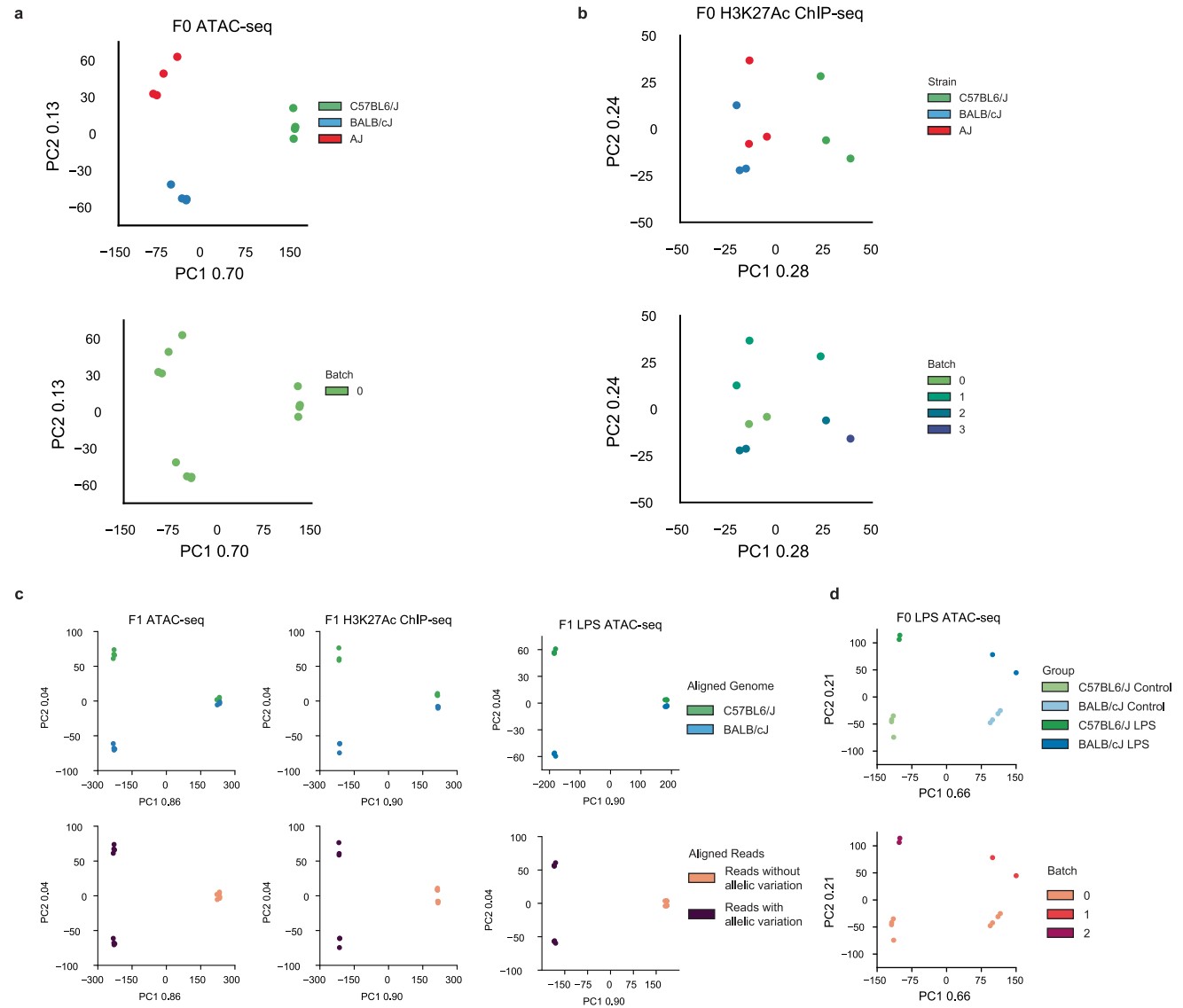

**Extended Data Fig. 3 | Principal component analysis of epigenetic datasets. a**, PCA of parental ATAC-seq data colored by strain (top) and batch (bottom). **b**, PCA of parental H3K27Ac ChIP-seq data colored by strain (top) and batch (bottom). **c**, PCA of F1 hybrid ATAC-seq and ChIP-seq data with data colored by strain and read alignment method, read alignment methods include perfectly aligned reads (pink) and perfectly aligned reads overlying mutations that discriminate between BALB/cJ and C57BL/6J (maroon). **d**, PCA of parental LPS treated Kupffer cells with control Kupffer cells as comparison, colored by strain (top) and batch (bottom).

**a** De novo motif enrichment **b** De novo motif enrichment

ATAC-seq H3K27ac ChIP-seq

| Rank | Motif | p-val | %Target %Bkgd | Best Match |
|------|-------|-------|---------------|------------|
| 1 | | 4240 | 58/11 | PU.1 |
| 2 | | 567 | 31/14 | C/EBP |
| 3 | | 546 | 19/7 | AP-1 |
| 4 | | 407 | 11/3 | IRF |
| 5 | | 360 | 67/50 | PAX2 |
| 6 | | 326 | 4/0.4 | CTCF |
| 7 | | 291 | 10/3 | RELA |
| 8 | | 246 | 13/6 | USF2 |
| 9 | | 152 | 7/3 | CHOP/MEF |
| 10 | | 136 | 12/7 | ATF |
| 11 | | 130 | 2/0.3 | PU.1/IRF |
| 12 | | 98 | 4/2 | MEF2 |
| 13 | | 94 | 7/3 | RUNX1 |
| 14 | | 84 | 1/0.2 | LXR/RXR |
| 15 | | 68 | 7/4 | SP1 |

| Rank | Motif | p-val | %Target %Bkgd | Best Match |
|------|-------|-------|---------------|------------|
| 1 | | 2393 | 48/12 | PU.1 |
| 2 | | 485 | 9/2 | PU.1/IRF |
| 3 | | 353 | 15/6 | AP-1 |
| 4 | | 309 | 32/18 | C/EBP |
| 5 | | 275 | 18/8 | ARNTL |
| 6 | | 263 | 6/1 | PU.1/Spic |
| 7 | | 117 | 7/3 | RELA |
| 8 | | 115 | 15/9 | MAFb |
| 9 | | 86 | 13/8 | NRF2 |
| 10 | | 66 | 12/8 | RUNX1 |
| 11 | | 56 | 2/0.7 | MEF2 |
| 12 | | 49 | 8/5 | SP1 |
| 13 | | 41 | 0.2/0.0 | ERRA |
| 14 | | 35 | 0.3/0.02 | CTCF |
| 15 | | 35 | 6/4 | CREM |

**Extended Data Fig. 4 | Motif enrichment analyses.** HOMER de novo motif analysis of shared accessible enhancers ('ATAC', ATAC-seq tags > 8) shared by all 3 strains and active enhancers ('H3K27Ac', H3K27Ac ChIP-seq tags > 16). **a**, Top 15 de novo motifs enriched at accessible enhancers as determined by HOMER. **b**, Top 15 de novo motifs enriched at active enhancers as determined by HOMER.

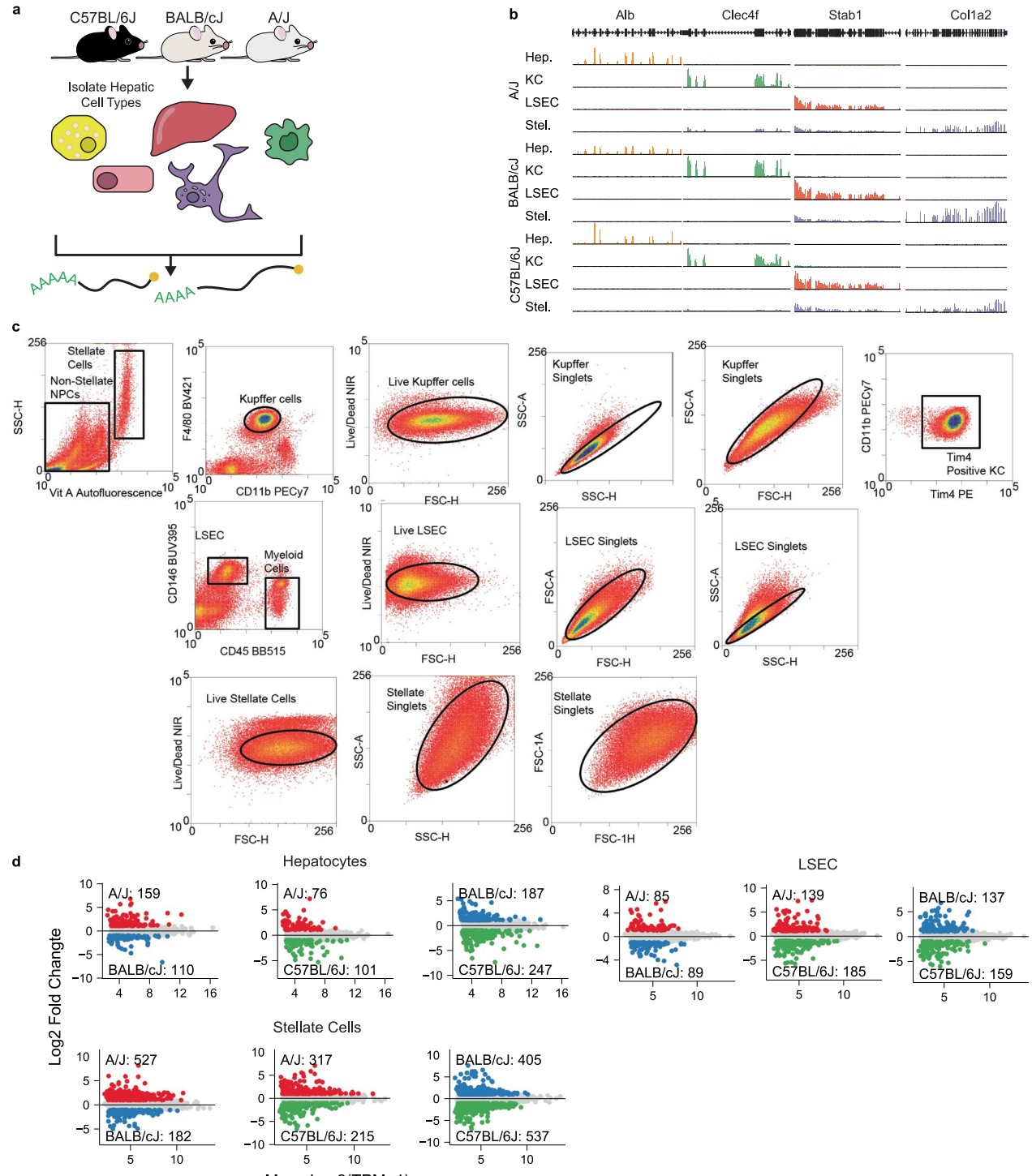

**Extended Data Fig. 5 | Data generation for network analysis. a**, Experimental schematic for isolation of hepatic cell types from inbred strains. **b**, Assessment of cell isolation purity at via RNA-seq signal at cell specific gene expression loci. Hepatocytes (yellow shades), Kupffer cells (green shades), and liver sinusoidal endothelial cells (LSEC) (red shades) were sorted with <1% contamination.

Hepatic stellate cell (purple shades) RNA-seq libraries displayed minor (<10%) contamination with LSECs and Kupffer cells (seen as RNA-seq signal in Clec4f and Stab1 loci). **c**, FACS strategy for stellate, LSEC, and Kupffer cell isolation. **d**, Strain-specific transcriptional variation in hepatocytes, LSECs, and Stellate cells. N = 4 samples per subgroup.

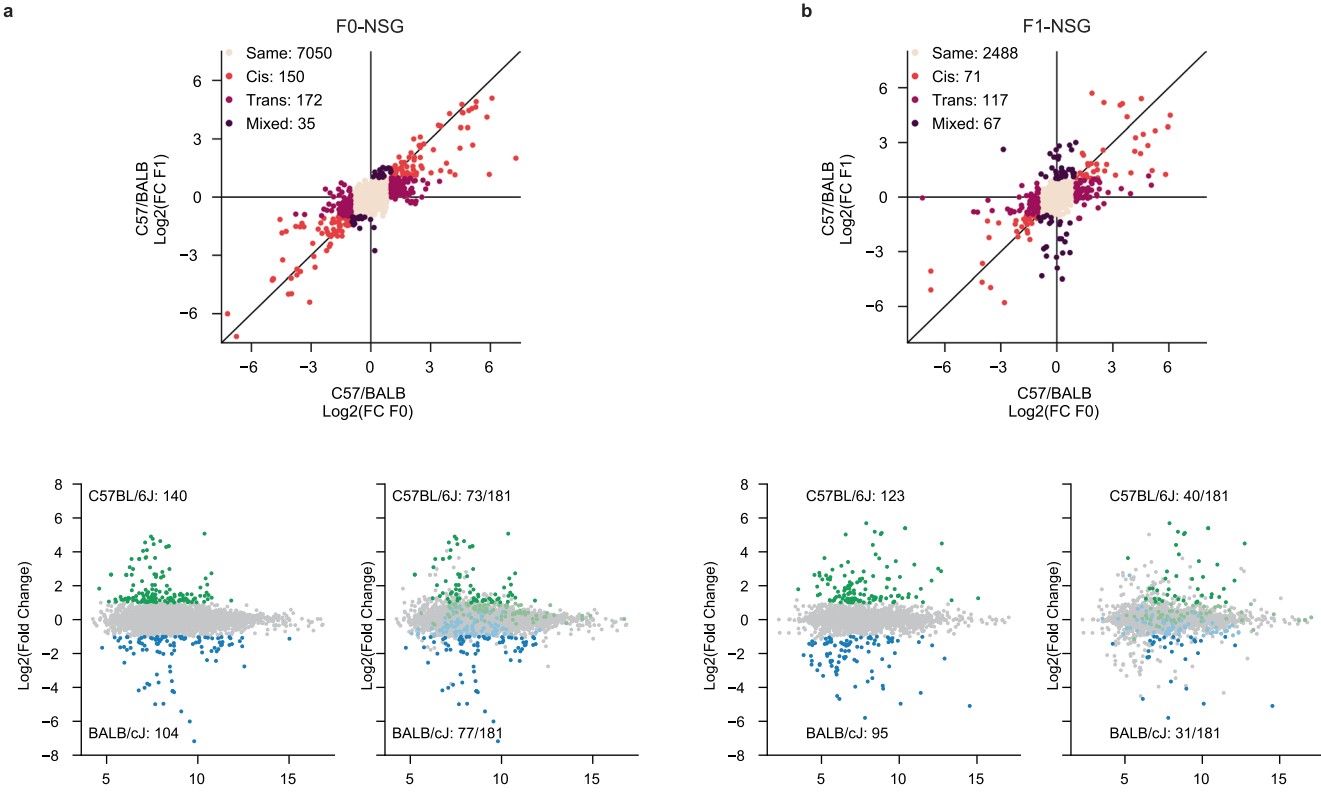

**Extended Data Fig. 6 | Strain- and allele-specific expression in NSG models.**
**a**, Comparison of strain-specific gene expression in parental C57BL/6J and BALB/cJ Kupffer cells and C57BL/6J and BALB/cJ Kupffer cells isolated from NSG hosts following bone marrow transplant. Top, cis trans plot. Bottom left, MA plot showing strain-specific gene expression in the F0-NSG model. Bottom right, MA plot of F0-NSG gene expression overlaid with F0 differential genes. Dark colored genes are strain specific in parental Kupffer cells and F0-NSG Kupffer cells, while light colored genes lose strain specificity in the F0-NSG model. **b**, Comparison of strain-specific expression in parental C57BL/6J and BALB/cJ Kupffer cells and allelic bias in CB6F1/J Kupffer cells isolated from NSG hosts following bone marrow transplant.

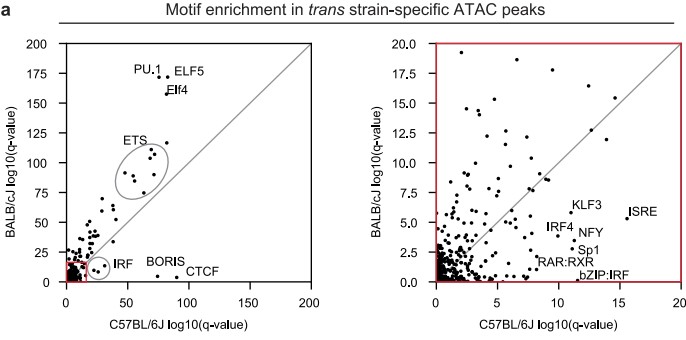

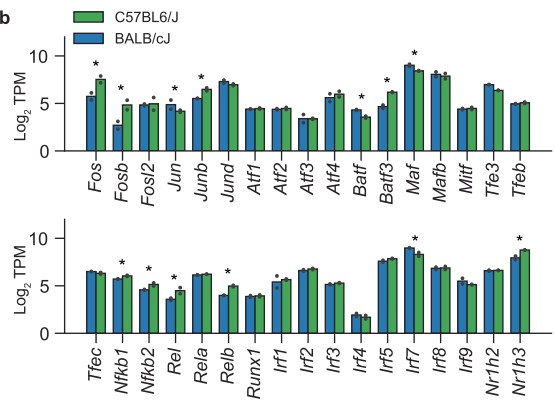

**c**

| | Rank | Motif | Best Match | P value |
|---|---|---|---|---|
| C57BL/6J *trans* gene enhancers | 1 | | SpiB | 1e-60 |
| | 2 | | Rel/RelA | 1e-25 |
| | 3 | | AP-1 | 1e-12 |
| BALB/cJ *trans* gene enhancers | 1 | | SpiC | 1e-40 |
| | 2 | | AP-1 | 1e-16 |
| | 3 | | ATF | 1e-11 |

**Extended Data Fig. 7 | Motif enrichment analyses of trans-associated data.**
**a**, Left, HOMER known motif enrichment in BALB/cJ trans-regulated ATAC-seq peaks (y-axis) and C57BL/6J trans-regulated ATAC-seq peaks (x-axis). Right, view of highlighted region in left panel. **b**, Expression of selected transcription factors in C57BL/6J and BALB/cJ F0 Kupffer cells. * indicates differential expression with DESeq2 adjusted p-value < 0.05 (Wald's test with multiple testing correction using Benjamini–Hochberg method) Exact p-values as follows: Fos 0.0017; Fosb 0.003; Jun 0.026; Junb 9.38e-9; Batf 0.047; Batf3 4.04e-10; Maf 8.15e-06; Nfkb1 0.004; Nfkb2 0.006; Rel 0.001; Relb 1.86e-6; Irf7 2.25e-5; Nr1h3 3.51e-7. **c**, Motif enrichment in ATAC-seq labeled enhancers associated with C57BL/6J and BALB/cJ trans genes.

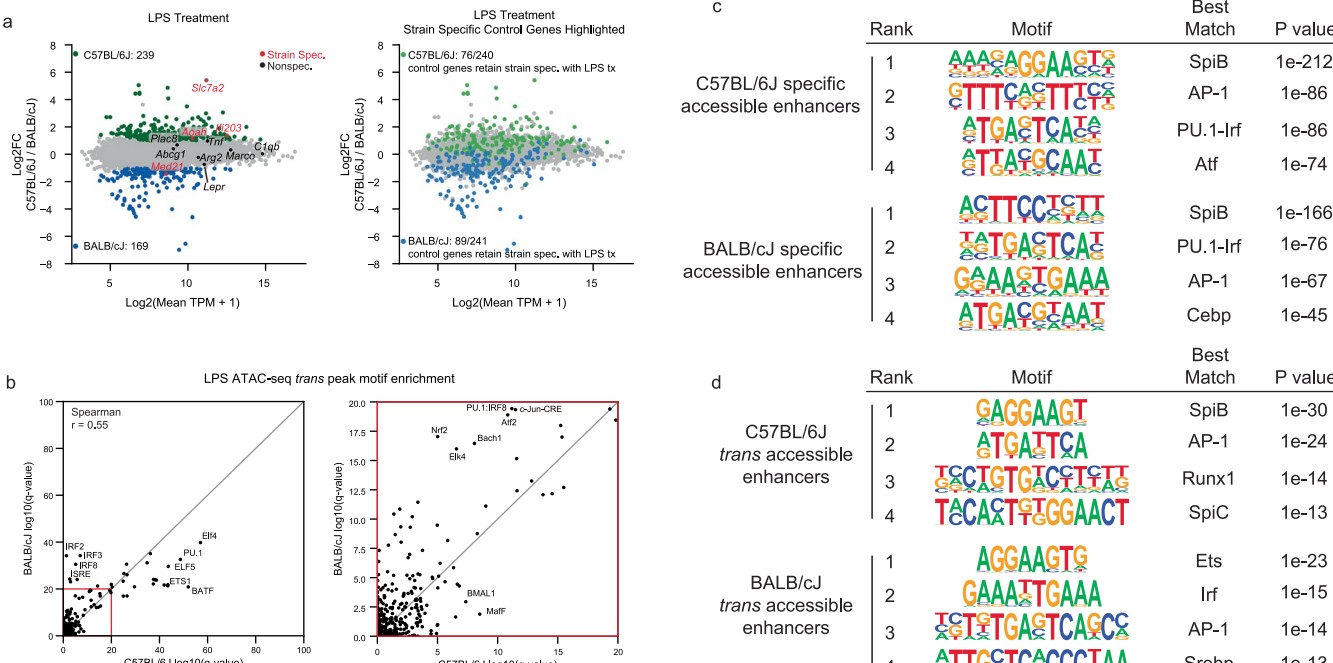

**Extended Data Fig. 8 | Strain-specific transcriptional regulation in response to LPS. a**, Effect of LPS treatment on homeostatic strain-specific genes. Left panel shows differential expression of transcripts in LPS treated C57BL/6J and BALB/cJ Kupffer cells. Select genes that display differential expression under LPS treatment are shown in red, while genes that are non-specific are shown in black. Right panel shows parental strain-specific genes overlayed onto strain-specific expression under LPS treatment. **b**, Motif enrichment in strain-specific trans peaks following LPS treatment. **c**, De novo motif enrichment of C57BL/6J and BALB/cJ specific accessible enhancers following LPS treatment. **d**, De novo motif enrichment of C57BL/6J and BALB/cJ specific trans enhancers following LPS treatment. P values in **c** and **d** calculated under binomial distribution as implemented by HOMER.

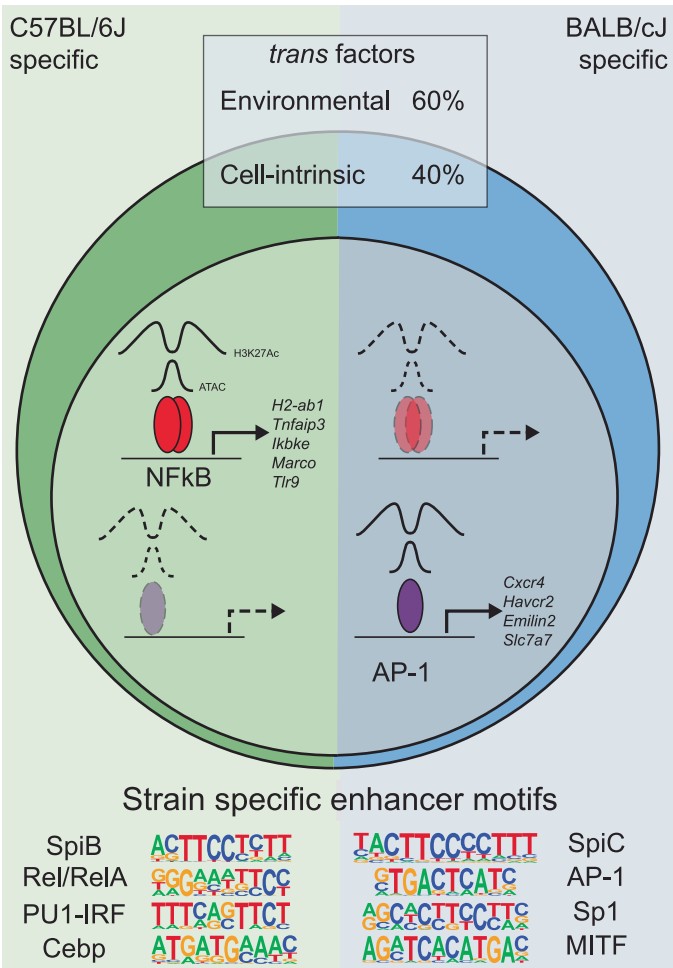

**Extended Data Fig. 9 | Summary model.** Model of trans-regulated strain-specific Kupffer cell transcriptional networks. A majority of trans effects are controlled by the strain-specific cellular environment, while a smaller fraction of trans effects are driven by cell-intrinsic differences in pathway activity. Trans differences in pathway activity induce differential transcription factor activation and gene expression. Examples of strain-specific transcription factor motifs and downstream genes are shown for BALB/cJ and C57BL/6J Kupffer cells.

# Reporting Summary

## Statistics

For all statistical analyses, confirm that the following items are present in the figure legend, table legend, main text, or Methods section.

| n/a | Confirmed | |
|---|---|---|
| ☐ | ☒ | The exact sample size (*n*) for each experimental group/condition, given as a discrete number and unit of measurement |
| ☒ | ☐ | A statement on whether measurements were taken from distinct samples or whether the same sample was measured repeatedly |
| ☐ | ☒ | The statistical test(s) used AND whether they are one- or two-sided<br>*Only common tests should be described solely by name; describe more complex techniques in the Methods section.* |
| ☐ | ☒ | A description of all covariates tested |
| ☒ | ☐ | A description of any assumptions or corrections, such as tests of normality and adjustment for multiple comparisons |
| ☐ | ☒ | A full description of the statistical parameters including central tendency (e.g. means) or other basic estimates (e.g. regression coefficient) AND variation (e.g. standard deviation) or associated estimates of uncertainty (e.g. confidence intervals) |
| ☐ | ☒ | For null hypothesis testing, the test statistic (e.g. *F*, *t*, *r*) with confidence intervals, effect sizes, degrees of freedom and *P* value noted<br>*Give P values as exact values whenever suitable.* |
| ☒ | ☐ | For Bayesian analysis, information on the choice of priors and Markov chain Monte Carlo settings |
| ☐ | ☒ | For hierarchical and complex designs, identification of the appropriate level for tests and full reporting of outcomes |
| ☐ | ☒ | Estimates of effect sizes (e.g. Cohen's *d*, Pearson's *r*), indicating how they were calculated |

*Our web collection on statistics for biologists contains articles on many of the points above.*

## Software and code

Policy information about availability of computer code

| Data collection | Cell sorting data was collected on a MoFlo Astrios EQ and processed with Summit (6.2 and 6.3). Immunoblotting data was collected using ChemiDoc XRS+ System using Image Lab (5.2.1). |
|---|---|
| Data analysis | **Sequencing Data Preprocessing**<br>Data was demultiplexed using bcl2fastq (v2.17). RNA-seq was mapped using STAR 2.5. H3K27Ac ChIP-seq and ATAC-seq data were mapped using bowtie4.1.2. ATAC-seq data were trimmed to 30 bp to remove sequencing adapters, for improved mapping efficiency. Strain specific genomes for BALB/cJ and A/J were generated from by replacing invariant positions of mm10 (https://hgdownload.soe.ucsc.edu/goldenPath/mm10/bigZips/) sequence with alleles reported in the Mouse Genome Project strain specific VCF files (ftp://ftp-mouse.sanger.ac.uk/REL-1303-SNPs_Indels-GRCm38/mgp.v3.snps.rsIDdbSNPv137.vcf.gz). mm10 was used as the C57BL/6J strain specific genome. Samples from parental strains of mice were mapped to the strain specific genome. Mapped reads were shifted to the chromosome coordinates of the mm10 genome build using MARGE.pl (v1.0) shift with -ind set to balbcj or aj for reads mapped to the BALB/cJ or A/J genome, respectively.<br><br>For samples from CB6F1/J samples, reads were mapped to the mm10 and BALB/cJ genome builds. Then the BALB/cJ mapped reads were shifted to the mm10 build with MMARGE v1.0 as above. Perfectly mapped reads spanning genetic mutations between BALB/cJ and mm10 were identified using the MMARGE.pl allele_specific_reads command with -ind set to BALB/cJ and a second time with -ind set to mm10 resulting in two SAM files for each biological sample: one SAM file containing reads perfectly mapped to the mm10 genome that spanned known DNA sequence differences relative to the BALB/cJ genome; and a second SAM file containing reads perfectly mapped to the BALB/cJ genome that spanning known DNA sequence differences relative to the reference mm10 genome.<br><br>Tag directories were called with HOMER (4.10) for each tag directory. ATAC-seq peaks were identified with HOMER using relaxed peak finding parameters "-L 0 -C 0 -fdr 0.9 -minDist 200 -size 200". IDR (v2.0.4) was used to test for reproducibility between ATAC-seq replicates. Peaksets |

from all pairwise IDR comparisons were merged for further analysis. Feature count matrices RNA-seq were generated using HOMER analyzeRepeats, or using annotatePeaks for ATAC-seq and ChIP-seq.

Statistical Analysis of Sequencing Data
Differential gene expression, histone acetylation, or open chromatin was assessed using DESeq2 v1.16. The raw p-values from DESeq2 for a given peak or gene were corrected for multiple testing using the Benjamini-Hochberg procedure. In some cases, ANOVA was used to assess expression differences of a priori defined candidate genes between many groups. Association of cis/trans with low, equal, and high basal regulation was performed with a Pearson Chi2 test (Python scipy.stats.contingency.chi2_contingency). Python packages used included numpy (1.20.2) and scipy (1.6.2), matplotlib (3.1.1), seaborn (0.9.0). R packages used included bioconductor-edger (3.28.0), gridExtra (2.3), cowplot (1.1.1), tibble (3.2.1), ggfortify (0.4.16), ggplotify (0.1.1), ggrepel (0.9.3) stringr(1.5.0), readr (2.1.4), ggplot2 (3.4.2), patchwork (1.1.2), scales (1.2.1), lubridate (1.9.2), dplyr (1.1.2), tidyr (1.3.0), and tidyverse (2.0.0), ggridges (0.5.4), ggpubr (0.6.0), forcats (1.0.0), stringr ( 1.5.0), purrr (1.0.1), and tibble (3.2.1)

Maggie Analysis
Maggie (1.2) analysis was performed with modifications to allow analysis of three pairwise comparisons. For each strain, positive sequences (either associated with increased ATAC-seq signal or increased H3K27Ac ChIP-seq signal) were extracted, in addition to the corresponding negative sequence from the comparator strain. Positive and negative sequences from all 6 possible foreground and background strains were concatenated and maximal motifs scores were calculated for each motifs available in JASPAR database and the net difference in max motif score between positive and negative sequences was calculated. Following calculation of net difference in max motif score for all sequences, a non-parametric Wilcoxon signed-rank two-sided test is used to assign significance to putative motifs by comparing the distribution of motif score differences to a null distribution centered on zero.

Niche-net
To assess putative strain-specific ligand activity, we first filtered the NicheNet ligand-target matrix to only consider ligands in which:

1. The ligand was expressed by a cell of the hepatic niche within that strain at > 10 TPM.
2. The receptor was expressed by Kupffer cells from that strain at > 10 TPM.

We also included selected metabolic ligands for which expression data were not available. We did not require that ligands or their receptors be differentially expressed by sender or receiver cells. Target genes were selected to be any gene that had significantly higher expression in a pairwise comparison of that strain ("union" gene set, adjusted p value < 0.05, log fold change > 2, TPM > 10 expression in Kupffer cells). As a background we considered all genes that were expressed at TPM > 10 in Kupffer cells. The NicheNet ligand activity score was then computed as the Pearson correlation coefficient between the ligand-target score and the binary vector indicating whether a target gene was differentially expressed. For heatmaps select top scoring ligands from each strain were aggregated and displayed with ligand z-scores. Ligand receptor interaction scores were displayed for ligand-receptor pairs with a receptor expressed by Kupffer cells in at least one strain. For the circos plot analysis ligand-target interaction scores were displayed as arrow thicknesses linking a ligand to its target gene. R packages used were circlize (0.4.11), dplyr (1.0.2), ggplot2 (3.3.5), tidyr (1.1.2), readr (1.4.0), forcats (0.5.0), stringr (1.4.0), purrr (0.3.4), nichenetr (1.0.0), A.C.Rsuite (1.0.0).

Statistical Analysis of
The effect of strain on NASH CRN and fibrosis score was assessed with a Kruskal-Wallis test (R, Kruskal.test). The effect of NASH or control diets between strains across time on mass was assessed with a linear mixed model fit by maximum likelihood. The hierarchical relationship of repeated measures on individual mice over time was controlled using a random effect term in the model (R, lme4::lmer(Mass ~ 1 + Week * Diet * Genotype + (1 + Week | Mouse), data = df1, REML = F). The significance of main effects or their interactions was assessed using Type III ANOVA with Satterthwaite's method (R, anova(fittedModel)). Within strain interactions of Week*Diet were assessed by sub-setting data on strain, and running a reduced linear mixed model (R, lmer(Mass ~ 1 + Week * Diet + (1 + Week | Mouse), data = strain_subset, REML = F). R packages used included glmmTMB (1.1.7), nlme (3.1-162), multcomp (1.4-25), tidyverse (2.0.0), sjPlot (2.8.14), lme4 (1.1-34), and lmerTest (3.1.3).

Analysis code is available on GitHub:
https://github.com/HunterBennett/KupfferCell_NaturalGeneticVariation

For manuscripts utilizing custom algorithms or software that are central to the research but not yet described in published literature, software must be made available to editors and reviewers. We strongly encourage code deposition in a community repository (e.g. GitHub). See the Nature Portfolio guidelines for submitting code & software for further information.

# Data

Policy information about availability of data

All manuscripts must include a data availability statement. This statement should provide the following information, where applicable:
- Accession codes, unique identifiers, or web links for publicly available datasets
- A description of any restrictions on data availability
- For clinical datasets or third party data, please ensure that the statement adheres to our policy

The datasets generated as part of the current study are available in the Gene Expression Omnibus repository. (GSE216164).

Processed data is made available at Zenodo: https://zenodo.org/record/7829622#.ZHogly-B1qs

Sequencing data was mapped to the publicly available GRCm38/mm10 genome: https://hgdownload.soe.ucsc.edu/goldenPath/mm10/bigZips/

Mouse strains genomes were generated using publicly available Mouse Genomes Project Variant Call Format (VCF) files: ftp://ftp-mouse.sanger.ac.uk/REL-1303-SNPs_Indels-GRCm38/mgp.v3.indels.rsIDdbSNPv137.vcf.gz

Datasets used in figure 3f were sourced from the Gene Expression Omnibus (GEO) Series GSE128337: https://www.ncbi.nlm.nih.gov/geo/query/acc.cgi?acc=GSE128337.

## Human research participants

Policy information about studies involving human research participants and Sex and Gender in Research.

| | |
|---|---|
| Reporting on sex and gender | Not applicable |
| Population characteristics | Not applicable |
| Recruitment | Not applicable |
| Ethics oversight | Not applicable |

Note that full information on the approval of the study protocol must also be provided in the manuscript.

# Field-specific reporting

Please select the one below that is the best fit for your research. If you are not sure, read the appropriate sections before making your selection.

☒ Life sciences ☐ Behavioural & social sciences ☐ Ecological, evolutionary & environmental sciences

For a reference copy of the document with all sections, see nature.com/documents/nr-reporting-summary-flat.pdf

# Life sciences study design

All studies must disclose on these points even when the disclosure is negative.

| | |
|---|---|
| Sample size | Sample size calculations were performed using a power analysis method detailed in Hart et al., J Comp Bio, 2014. According to these calculations a minimum of 2 samples would allow detection of 2-fold changes with >90% power. Therefore RNA-seq and ATAC-seq, and H3K27Ac ChIP-seq studies used a minimum of two biological replicates per cell subset.<br><br>We determined effect sizes of our own pathological scoring data from mouse NASH experiments and determined that we had >90% power to detect a 2-fold change in NASH CRN score at α = 0.05, and 43% power to detect a 60% fold change in fibrosis score at α = 0.05. |
| Data exclusions | Data quality was assessed using Spearman correlation between replicates Spearman correlation was calculated using TPM values for RNA-seq data, and tags overlying ATAC-seq peaks for ATAC-seq and H3K27Ac ChIP-seq data. Correlation was subset per strain, cell type, and condition and only replicates with good correlation (>0.80 typically, but determined on a case-by-case basis) were kept. |
| Replication | RNA-seq, ATAC-seq, and H3K27Ac ChIP-seq in Kupffer cells other non-parenchymal cell types from the inbred strains and CB6F1/J hybrids were performed in 2 independent experiments with highly correlated findings. All assays were successfully replicated 2 or more times and quantification and statistics are run on combined replicate experiments.<br><br>AMLN diet data included data from >5 replicated experiments.<br><br>RNA-seq from hepatocytes isolated from A/J, BALB/cJ, and C57BL/6J mice had n=2 independent samples but was not independently replicated as this data was only used for identifying putative ligands for use in the NicheNet algorithm. |
| Randomization | Data was generated from littermate mice ordered directly from the Jackson laboratories. Mice from the same shipment and, when possible, the same cage, were used within experimental groups. Littermates were assigned randomly for treatment with lipopolysaccharide or phosphate buffered saline. When assessing response of each strain to a NASH inducing diet, littermates were split randomly into separate cages and subsequently fed a NASH inducing diet or custom control diet for 30 weeks. |
| Blinding | Blinding was used in the assessment of liver pathology, which was performed by a board-certified pathologist. In this case the pathologist was given histopathologic slides with encoded IDs that were scored using the NASH CRN and fibrosis scores. These encoded IDs were then translated back to the original sample ID by the research team. Researchers were not blinded to groupings for other experiments as mice received identical treatments. |

# Reporting for specific materials, systems and methods

We require information from authors about some types of materials, experimental systems and methods used in many studies. Here, indicate whether each material, system or method listed is relevant to your study. If you are not sure if a list item applies to your research, read the appropriate section before selecting a response.

## Materials & experimental systems

| n/a | Involved in the study |
|---|---|
| ☐ | ☒ Antibodies |
| ☒ | ☐ Eukaryotic cell lines |
| ☒ | ☐ Palaeontology and archaeology |
| ☐ | ☒ Animals and other organisms |
| ☒ | ☐ Clinical data |
| ☒ | ☐ Dual use research of concern |

## Methods

| n/a | Involved in the study |
|---|---|
| ☐ | ☒ ChIP-seq |
| ☐ | ☒ Flow cytometry |
| ☒ | ☐ MRI-based neuroimaging |

# Antibodies

| | |
|---|---|
| Antibodies used | Anti-Mouse/Human CD11b PE/Cy7 (clone M1/70); BioLegend; 101216; RRID: AB_312799<br>Anti-Mouse CD146 BUV395 (clone ME-9F1); BD Biosciences; 740330; RRID: AB_2740063<br>Anti-Mouse CD16/32 (clone 93); BioLegend; 101302; RRID: AB_312801<br>Anti-Mouse CD45 Alexa488 (clone 30F11); BioLegend; 103122; RRID: AB_493531<br>Anti-Mouse CD45 BB515 (clone 30-F11); BD Biosciences; 564590; RRID: AB_2738857<br>Anti-Mouse Cx3cr1 Alex647 (clone SA011F11); BioLegend; 149004; RRID: AB_2564273<br>Anti-Mouse F4/80 BV421 (clone BM8); BioLegend; 123132; RRID: AB_11203717<br>Anti-Mouse Tim4 Alexa647 (clone RMT4-54); BioLegend; 127641; RRID: AB_2271648<br>Anti-Mouse Tim4 PE (clone RMT4-54); BioLegend; 130008; RRID: AB_2201843<br>Anti Mouse H-2Dd PE (clone 34-2-12); BioLegend; 110607; RRID: AB_313488<br>Anti Mouse H-2Kb Alexa647 (clone AF6-88.5); BioLegend; 116512 ; RRID: AB_492917<br>Anti-phospho-STAT3 (Tyr705) (clone D3A7); Cell Signaling Technology; 9145; RRID: AB_2491009<br>Anti-STAT3 (clone 124H6); Cell Signaling Technology; 9139; RRID:AB_331757<br>Donkey Anti-Mouse Immunoglobulins/HRP (Dako); no longer available on the manufacturer's website<br>Donkey Anti-Rabbit Immunoglobulins/HRP (Dako); no longer available on the manufacturer's website |
| Validation | Anti-Mouse/Human CD11b PE/Cy7 - validated by manufacturer in flow assays and internally using fluorescence minus one controls<br>Anti-Mouse CD146 BUV395 - validated by manufacturer in flow assays and internally using fluorescence minus one controls<br>Anti-Mouse CD16/32 - validated by manufacturer in flow assays<br>Anti-Mouse CD45 Alexa488 - validated by manufacturer in flow assays and internally using fluorescence minus one controls<br>Anti-Mouse CD45 BB515 - validated by manufacturer in flow assays and internally using fluorescence minus one controls<br>Anti-Mouse Cx3cr1 Alexa647 - validated by manufacturer in flow assays and internally using fluorescence minus one controls<br>Anti-Mouse F4/80 BV421 - validated by manufacturer in flow assays and internally using fluorescence minus one controls<br>Anti-Mouse Tim4 Alexa647 - validated by manufacturer in flow assays and internally using fluorescence minus one controls<br>Anti-Mouse Tim4 PE - validated by manufacturer in flow assays and internally using fluorescence minus one controls<br>Anti Mouse H-2Dd PE - validated by manufacturer in flow assays and internally using fluorescence minus one controls<br>Anti Mouse H-2Kb Alexa647 - validated by manufacturer in flow assays and internally using fluorescence minus one controls<br>Anti-phospho-STAT3 - validated by manufacturer in immunoblotting assays<br>Anti-STAT3 -  validated by manufacturer in immunoblotting assays<br><br>Detailed validation information is available on the product specification sheets. |

# Animals and other research organisms

Policy information about studies involving animals; ARRIVE guidelines recommended for reporting animal research, and Sex and Gender in Research

| | |
|---|---|
| Laboratory animals | Mus musculus laboratory mice were used in all experiments. Baseline studies were performed with male A/J, BALB/cJ, C57BL/6J and CB6F1/J (first generation intercross of male C57BL/6J and female BALB/cJ) mice age 8-10 weeks.<br><br>NOD-scid IL2Rgnull mice age 8-10 weeks were conditioned with the myeloablative agent busulfan at 25 mg/kg for 2 consecutive days. On the third day, mice were engrafted via retro orbital injection with magnetically enriched, lineage negative, hematopoietic stem cells (Miltenyi) from BALB/cJ, C57BL/6J, or CB6F1/J donors. Engraftment efficiency was monitored in peripheral blood after 4 and 8 weeks.<br><br>Mice were fed for up to 30 weeks with a NASH-model diet (Research Diets, D09100301) composed of 40 kcal% from fat, 20 kcal% from fructose, and 2% cholesterol by mass, or a custom defined control diet (Research Diet, D15100601) composed of 10% kcal from fat with 50 g inulin (a dietary fiber) per 4,057 kcal.<br><br>Mice were housed in individually ventilated cages in standard conditions at 22°C with 40 ± 5% relative humidity and a 12-h light/dark cycle. Water and standard laboratory diet were available ad libitum, unless indicated otherwise. |
| Wild animals | Study did not involve wild animals. |
| Reporting on sex | Sex was not considered as a variable in these studies due to poor susceptibility of female mice of the chosen strains to NASH models. |
| Field-collected samples | The study did not involve samples collected from the field. |

| Ethics oversight | All animals were maintained and all procedures performed were approved by the University of California San Diego Animal Care and Use Committee in accordance with an approved animal study protocol meeting AALAC standards. |

Note that full information on the approval of the study protocol must also be provided in the manuscript.

# ChIP-seq

## Data deposition

☒ Confirm that both raw and final processed data have been deposited in a public database such as GEO.

☒ Confirm that you have deposited or provided access to graph files (e.g. BED files) for the called peaks.

| Data access links
*May remain private before publication.* | Sequencing data generated in this study are available at GEO through accession number GSE216164.<br><br>Graph files (HOMER peaks) are available on GEO as well as Zenodo:<br>https://zenodo.org/record/7829622#.ZHogly-B1qs |
| Files in database submission | h3k27ac_AJ_Kupffer_Control_rep1_R1.fastq.gz<br>h3k27ac_AJ_Kupffer_Control_rep1_R2.fastq.gz<br>h3k27ac_AJ_Kupffer_Control_rep2_R1.fastq.gz<br>h3k27ac_AJ_Kupffer_Control_rep2_R2.fastq.gz<br>h3k27ac_AJ_Kupffer_Control_rep3_R1.fastq.gz<br>h3k27ac_AJ_Kupffer_Control_rep3_R2.fastq.gz<br>h3k27ac_BALBcJ_Kupffer_Control_rep1_R1.fastq.gz<br>h3k27ac_BALBcJ_Kupffer_Control_rep1_R2.fastq.gz<br>h3k27ac_BALBcJ_Kupffer_Control_rep2_R1.fastq.gz<br>h3k27ac_BALBcJ_Kupffer_Control_rep2_R2.fastq.gz<br>h3k27ac_BALBcJ_Kupffer_Control_rep3_R1.fastq.gz<br>h3k27ac_BALBcJ_Kupffer_Control_rep3_R2.fastq.gz<br>h3k27ac_C57_Kupffer_Control_rep1_R1.fastq.gz<br>h3k27ac_C57_Kupffer_Control_rep1_R2.fastq.gz<br>h3k27ac_C57_Kupffer_Control_rep2_R1.fastq.gz<br>h3k27ac_C57_Kupffer_Control_rep2_R2.fastq.gz<br>h3k27ac_C57_Kupffer_Control_rep3_R1.fastq.gz<br>h3k27ac_C57_Kupffer_Control_rep3_R2.fastq.gz<br>h3k27ac_cb6f1j_kupffer_chow_rep1_R1.fastq.gz<br>h3k27ac_cb6f1j_kupffer_chow_rep1_R2.fastq.gz<br>h3k27ac_cb6f1j_kupffer_chow_rep2_R1.fastq.gz<br>h3k27ac_cb6f1j_kupffer_chow_rep2_R2.fastq.gz<br>h3k27ac_cb6f1j_kupffer_chow_rep3_R1.fastq.gz<br>h3k27ac_cb6f1j_kupffer_chow_rep3_R2.fastq.gz<br>input_AJ_Kupffer_Control_rep1_R1.fastq.gz<br>input_AJ_Kupffer_Control_rep1_R2.fastq.gz<br>input_AJ_Kupffer_Control_rep2_R1.fastq.gz<br>input_AJ_Kupffer_Control_rep2_R2.fastq.gz<br>input_AJ_Kupffer_Control_rep3_R1.fastq.gz<br>input_AJ_Kupffer_Control_rep3_R2.fastq.gz<br>input_BALBcJ_Kupffer_Control_rep1_R1.fastq.gz<br>input_BALBcJ_Kupffer_Control_rep1_R2.fastq.gz<br>input_BALBcJ_Kupffer_Control_rep2_R1.fastq.gz<br>input_BALBcJ_Kupffer_Control_rep2_R2.fastq.gz<br>input_BALBcJ_Kupffer_Control_rep3_R1.fastq.gz<br>input_BALBcJ_Kupffer_Control_rep3_R2.fastq.gz<br>input_C57_Kupffer_Control_rep1_R1.fastq.gz<br>input_C57_Kupffer_Control_rep1_R2.fastq.gz<br>input_cb6f1j_kupffer_chow_rep1_R1.fastq.gz<br>input_cb6f1j_kupffer_chow_rep1_R2.fastq.gz<br>input_cb6f1j_kupffer_chow_rep2_R1.fastq.gz<br>input_cb6f1j_kupffer_chow_rep2_R2.fastq.gz<br>input_cb6f1j_kupffer_chow_rep3_R1.fastq.gz<br>input_cb6f1j_kupffer_chow_rep3_R2.fastq.gz<br><br><br>h3k27ac_f0_raw.txt<br>h3k27ac_f1_raw.txt |
| Genome browser session<br>(e.g. UCSC) | https://genome.ucsc.edu/s/hunterrb/glass_kupffer_strains_manuscript |

## Methodology

| Replicates | N=3 independent replicates were used for each experimental subgroup. Samples with pearson correlation of H3K27Ac ChIP-seq signal annotated over ATAC-seq peaks >0.8 were kept for analysis. |

| Sequencing depth | Inbred strain samples were sequenced to a depth of ~5 million reads. F1 samples were sequenced to a depth of ~50 million reads to allow sufficient detection of perfectly mapped reads spanning mutations between the parental genomes. |
|---|---|
| Antibodies | Active Motif, 39685, anti-H3K27ac, RRID: AB_2793305, Clone: MABI 0309 |
| Peak calling parameters | Our pipeline did not call H3K27Ac ChIP-seq peaks, instead ChIP-seq signal was annotated over relevant ATAC-seq peaks identified using the irreproducible discovery rate (IDR v2.0) algorithm. IDR ATAC-seq peaks were annotated using HOMER (v4.10) annotate_peaks command using -size 1000 flag to set a 2000bp window around the ATAC-seq peak. |
| Data quality | Data were examined on the UCSC genome browser to ensure efficacy of the immunoprecipitation step. Outlier samples were screened for using Spearman correlation between samples within each subgroup. All samples with a Spearman correlation > 0.90 were retained for further analysis. |
| Software | H3K27Ac ChIP-seq was mapped to strain specific genomes (mm10 was used as the C57BL/6J strain specific genome) generated from Mouse Genome Project VCF files. Mapping was performed using bowtie 0.12.7. Samples from parental strains of mice were mapped to the strain specific genome. Mapped reads were shifted to the chromosome coordinates of the mm10 genome build using MARGE.pl (v1.0) shift with -ind set to balbcj or aj for reads mapped to the BALB/cJ or A/J genome, respectively. Tag directories were generated from SAM alignment files using HOMER v4.10. |

## Flow Cytometry

### Plots

Confirm that:

☒ The axis labels state the marker and fluorochrome used (e.g. CD4-FITC).

☒ The axis scales are clearly visible. Include numbers along axes only for bottom left plot of group (a 'group' is an analysis of identical markers).

☒ All plots are contour plots with outliers or pseudocolor plots.

☒ A numerical value for number of cells or percentage (with statistics) is provided.

### Methodology

| Sample preparation | Mouse livers were retrograde perfused for 3 min at a rate of 5-7 mL/min through the inferior vena cava with HBSS without Ca2+ or Mg2+ supplemented with 0.5 mM EGTA, 0.5 mM EDTA, and 20 mM HEPES. Perfusions were then switched to 40 mL of a digestion buffer, held at 37°C , comprised of HBSS with Ca2+ and Mg2+ supplemented with 0.033 mg/mL of Liberase TM (Roche), 20 mg/mL DNaseI (Worthington), and 20 mM HEPES. Livers were then excised, minced, and digested for an additional 20 min in vitro at 37°C with gentle rotation in 20 mL of fresh digestion buffer. The perfusion and digestion steps were performed in the presence of 1 mM flavopiridol to offset transcriptional changes associated with digestion. After tissue digestion, cells were passed through a 70-micron cell strainer and hepatocytes removed by 2 low-speed centrifugation steps at 50 X G for 2 min. Non-parenchymal cells in the supernatant were further separated from debris by pelleting for 15 min at 600 X G in 50 mL of 20% isotonic Percoll (Sigma Aldrich) at room temperature. Cells were then washed from Percoll containing buffer and suspended in 10 mL 28% OptiPrep (Sigma Aldrich) and carefully underlaid beneath 3 mL of wash buffer. The resulting gradient was centrifuged at 1,400 X G for 25 min at 4°C with no break, and cells enriched at the interface were saved and subjected to isotonic erythrocyte lysis. Cells were washed after erythrocyte lysis and immediately used purified by cells sorting. |
|---|---|
| Instrument | Beckman Coulter Mo-Flo Astrios EQ configured with spatially separated 355 nm, 405 nm, 488 nm, 561 nm, and 642 nm lasers |
| Software | The Astrios was controlled and set up using Summit software. Some post-sort analyses were done using FlowJo (10.4.1). |
| Cell population abundance | Post-sort purities routinely exceeded 95% as assessed by reacquisition on the same cell sorter. Transcriptomics is sensitive to contamination and we could observe indications that our method for purifying hepatic stellate cells had varied contamination with Kupffer cells. |
| Gating strategy | Kupffer cells were defined as 355:494/20Low (inferred as retinol autofluorescence), SSCLow, CD146Neg, CD45Pos, F4/80High, CD11bIntermediate, Live, Singlets. Liver sinusoidal endothelial cells were defined at 355:494/20Low (inferred as retinol autofluorescence), SSCLow, CD45Neg, CD146Pos, Live, Singlets. Hepatic stellate cells were defined as 355:494/20High (inferred as retinol autofluorescence), SSCIntermediate, Live, Singlets. Singlets were identified using SSC-H/SSC-A and FSC-H/FSC-A for all populations. |

☒ Tick this box to confirm that a figure exemplifying the gating strategy is provided in the Supplementary Information.

