## [Peer Review File · Nature Immunology]

Peer Review Information

Journal: Nature Immunology

Manuscript Title: Discrimination of cell-intrinsic and environment-dependent effects of natural genetic variation on Kupffer cell epigenomes and transcriptomes

Corresponding author name(s): Professor Christopher Glass, Dr. Ty Troutman

Reviewer Comments & Decisions:

Decision Letter, initial version:
--

8th Jan 2023

Dear Chris,

Thank you for providing a detailed point-by-point response to the referees' comments on your manuscript entitled, "Systematic analysis of transcriptional and epigenetic effects of genetic variation in Kupffer cells enables discrimination of cell intrinsic and environment-dependent mechanisms". As noted previously, while they find your work of considerable potential interest, they have raised quite substantial concerns that must be addressed. In light of these comments, we cannot accept the current manuscript for publication, but would be very interested in considering a revised version that addresses these concerns along the lines proposed in your rebuttal response.

Much of the revision appears to be reanalysis of datasets already in hand and textual clarifications of the methods/data analysis as well as better visual depiction of the results. Upon discussion with the other editors, we are inclined to waive doing additional F1 BM > NSG recipients.

Please revise your manuscript taking into account all reviewer and editor comments, please highlight all changes in the manuscript text file in Microsoft Word format. When you submit the revised manuscript please select the "Resource" article option, as we think the study befits that designation.

* If you have not done so already please begin to revise your manuscript so that it conforms to our

Article format instructions at <http://www.nature.com/ni/authors/index.html>. Refer also to any guidelines provided in this letter.

The Reporting Summary can be found here:

[REDACTED]

If you wish to submit a suitably revised manuscript we would hope to receive it within 6 months. If you cannot send it within this time, please let us know. We will be happy to consider your revision so long as nothing similar has been accepted for publication at Nature Immunology or published elsewhere.

Nature Immunology is committed to improving transparency in authorship. As part of our efforts in this direction, we are now requesting that all authors identified as 'corresponding author' on published papers create and link their Open Researcher and Contributor Identifier (ORCID) with their account on the Manuscript Tracking System (MTS), prior to acceptance. ORCID helps the scientific community achieve unambiguous attribution of all scholarly contributions. You can create and link your ORCID from the home page of the MTS by clicking on 'Modify my Springer Nature account'. For more information please visit www.springernature.com/orcid.

Thank you for the opportunity to review your work.

Kind regards,

Laurie

Laurie A. Dempsey, Ph.D.
Senior Editor
Nature Immunology
l.dempsey@us.nature.com
ORCID: 0000-0002-3304-796X

Referee expertise:

Referee #1: Systems immunology

Referee #2: Systems immunology

Referee #3: Fibrosis

Reviewers' Comments:

Reviewer #1:

Remarks to the Author:

Glass and colleagues have performed a solid study focused on integrating genetic and epigenetic contributions to Kupfer cell phenotypes. Kupffer cells are relatively understudied using the most modern approaches, so the data the authors have produced are an important contribution and they should be broadly useful.

In the context of the above, several concerns arise ranging from the primary data to the experimental setup, to the presentation of the results.

General comments:

- The primary epigenomic data are voluminous but there is little if any indication of data quality; well-used measures such as FRIP are not presented. More importantly, there is no indication in the text or methods specifying how the authors handled differences in data quality between two data sets when making comparisons.

- The finding of strain-specific differences environmental factors influencing Kupffer cell phenotypes is not unexpected. In this context, the authors highlighted phenotypic differences in strain susceptibility and resistance to NASH but did not comment on general mechanisms of how the observed genetic variation might influence differential susceptibility to NASH

- The authors propose that strain-specific Kupffer cells engrafted into a common host enabled quantitative assessment of cis versus trans effects of genetic variation on gene expression, and also quantification of cell autonomous effects. However, the presentation of these results is rather opaque. The authors should consider partitioning the text and figure panels into a more logical flow.

- It would be of value to compare hepatocytes, hepatic stellate cells, endothelial cell and other supportive cell types in the liver to improve desegregation of cell autonomous vs non-cell autonomous effects during NASH progression
- The authors could consider passaging F1 hybrids into NSG mice as a stronger comparator to the C57BL/6J and BALB/cJ NSG transplants
- The authors showed that changes in *Lepr* expression in Kupffer cells from C57BL/6 mice are associated with NASH induction. It is not clear how this finding relates to NASH susceptibility. Do similar/dissimilar expression changes occur upon feeding NASH-resistant mice an AMLN diet, and could they contribute to differences in NASH susceptibility?
- The analysis of parental vs NSG-passaged vs F1 hybrid Kupffer cells to untangle mechanisms related to cis vs trans and cell-intrinsic vs cell-extrinsic gene regulation is intriguing, but additional clarifying text will aid interpretation and relevance of the findings

Specific comments:

- Scaling the RNA expression data, e.g., $\log_2(\text{TPM}+1)$, may improve interpretability of the gene expression bar plots. As well, it is not clear if statistical tests were performed for gene expression comparisons (Fig 3d/f/g, Fig 5e, Fig 6c)
- There could be additional explanatory text to accompany the NicheNet analysis. In Figure 3c, it is not stated what the width of each line in the circos plot represents
- In Figure 6 and the accompanying text, the delineation between 'low basal', 'similar basal', and 'high basal' is not intuitive and seem to describe both basal and LPS-induced gene expression, which is not clear by the category names alone. In some cases, 'similar basal' and 'equal basal' are used interchangeably. Further clarification of how the categories were formed will be helpful
- It is unclear from the text what is meant by "LPS responsive strain" and "LPS non-responsive strain" (line 354-355, 370-373). Perhaps this should be changed to strain-specific LPS responsive genes or strain-specific LPS responsive enhancers?
- The legend corresponding to Extended Data Figure 1f indicates that bone marrow derived macrophages and Kupffer cells were hierarchically clustered, but Extended Data Figure 1f only shows clustering of Kupffer cells

Reviewer #2:

Remarks to the Author:

This paper seeks to analyze the effects of natural genetic variation on Kupffer cell gene expression through use of three inbred mouse strains (A/J, BALB/cJ, C57BL/6J) that display phenotypic differences in liver disease (in particular non-alcoholic steatohepatitis, NASH) as well as chimeric and F1 hybrid mouse models based on these strains. The study tries to quantify the extent of cis vs. trans effects of genetic variation on gene expression, and of cell-autonomous vs. non-cell-autonomous trans

effects, through a fairly complex experimental design:

- (i) first RNA-seq data is collected in the three strains in both Kupffer cells and bone-marrow derived macrophages (BMDMs) to identify strain-differential gene expression specific to Kupffer cells (Fig. 1);
- (ii) ATAC-seq and H3K27ac ChIP-seq is next generated to identify shared and differentially accessible active enhancers and non-acetylated (“poised”) enhancers and to identify motifs associated with strain-specific active enhancers (Fig. 2);
- (iii) a computational model of cellular signaling called NicheNet on RNA-seq from hepatocytes, stellate cells, and liver sinusoidal cells (LSECs) in these strains to ask if ligands expressed by these cell types could predict strain-specific gene expression patterns in Kupffer cells; follow-up validation experiments for one of the predictions — the effect of the top scoring ligand Lep, active in the BALB/cJ (and to a lesser extent, C57BL/6J) liver niche but not in A/J, on STAT3 phosphorylation downstream of Lepr signaling — are performed (Fig. 3);
- (iv) to try to quantify cis vs. trans and cell-autonomous vs. non-cell-autonomous effects between BALB/cJ and C57BL/6J, bone marrow from these strains is transplanted into NOD mice that have been treated with busulfan to deplete resident Kupffer cells, allowing generation of monocyte-derived Kupffer cells from donor progenitors, and strain-specific expression differences (with matching NOD host environments) are compared to allele-specific effects in Kupffer cells in F1 mice, though there are caveats to this design (Fig. 4);
- (v) ATAC-seq and H3K27ac ChIP data in parental vs. F1 Kupffer cells are compared to identify cis vs. trans effects (Fig. 5);
- (vi) strain-specific expression and ATAC-seq response to LPS stimulation (for BALB/cJ and C57BL/6J) are examined and coupled with motif analysis to find stronger cis effects in this setting (Fig. 6).

This is an interesting if somewhat descriptive study addressing important questions of how natural genetic variation impacts gene expression; given the strong focus on cis effects in current studies of the genetics of gene regulation, the authors take a valuable wider view to consider the widespread impact of trans effects, including non-cell-autonomous effects. The take-home message seems to be this statement from the Discussion: “In concert, systematic analysis of the transcriptomes and regulatory landscapes of Kupffer cells in A/J, BALB/cJ and C57BL/6J mice, in NSG chimeras, and in F1-hybrids, reveals dominant trans effects of natural genetic variation under homeostatic conditions and dominant cis effects on the response to LPS.” However, there are missing experimental and computational analysis steps in reaching this conclusion. While aspects of the analysis — like mapping sequencing data to F1 hybrid genomes — demonstrate the computational expertise of this lab, other parts are less satisfying — like trying to make quantitative arguments in Venn diagram fashion after imposing a significance threshold. The experiments outlined in point (iv) have clear caveats in comparing monocyte-derived Kupffer cells in immunodeficient mice to Kupffer cells of non-hematopoietic origin in immunocompetent F1 mice; there is also the missing comparison of F1 bone marrow transferred to NSG mice. In places, the analysis depends on previously published but not yet widely used algorithms (MAGGIE motif analysis, NicheNet) that would benefit from some background exposition. More comments are given below.

Major comments

- (1) Initial expression analysis. The heatmap in Fig. 1a seems minimally informative and at least should be larger. Perhaps the clusters of genes displaying different strain-specific patterns (from the hierarchical clustering) could be explored using gene ontology enrichments?
- (2) Initial epigenomic analyses. The term “poised” is not standard for accessible peaks lacking

H3K27ac (often connotes H3K4me1 signal) and should probably not be used here. A few sentences of exposition on MAGGIE would be useful to orient the reader. Moreover, since the MAGGIE paper describes a statistical test for pairs of sequences, i.e. coming from a pair of genomes, it is not clear how these pairwise analyses across 3 strains are summarized in Fig. 2g.

(3) NicheNet results. A brief description of NicheNet would be helpful. The main result for which there are follow-up experiments — i.e. the strain-specific effect of the ligand Lep on BALB/cJ, shown in Fig. 3b — is a bit confusing since it seems that none of the niche cell types seem to express much Lep (in any strain) based on Fig. 3a. Where is the endogenous Lep coming from? The immunoblot assessment of phospho-STAT3 in liver treatment after leptin injection is not a cell type specific assessment showing the response is limited to Kupffer cells, though *Lep* expression seems limited to Kupffer cells. It would seem important to fill in some of the gaps in these validation experiments.

(4) Chimeric mouse experiments. This is an interesting but complex experimental design with many caveats. Comparing hematopoietic-derived Kupffer cells in immunodeficient mice with embryonic-derived Kupffer cells in immunocompetent F1 mice seems problematic; cross-referencing against published hematopoietic vs. embryonic Kupffer cell expression changes is inadequate to address the issue. Perhaps the physiologic stress of the busulfan treatment and bone marrow transplant contributes to non-cell-autonomous trans effects here, and these effects are not necessarily relevant in the context of e.g. genetic variation and liver disease? At minimum, there is a missing comparison of F1 bone marrow transplantation into NSG mice; these (hematopoietic) F1 Kupffer cells could be directly compared with the corresponding F0 strain Kupffer cells in a matching NSG host environment to address cell-autonomous trans effects. In general, the quantitative analysis is a bit weak here — based on Venn diagram style quantification.

Minor comments

^[1]_{SEP} - “Dependent on context, as much as 80% of allele-specific differences in cis regulatory activity can be linked to local variants” — please provide a citation for this statement^[1]_{SEP}

- It might be worth discussing that the pseudo genome alignment strategy, which edits the reference genome using the VCF file, ignores large structural variants that may contain important regulatory elements.

- Some phrase left out on line 206: “asked whether ligands expressed by could predict”

- Fig. 5: typos in labeling, “prental” -> “parental”^[1]_{SEP}

Reviewer #3:

Remarks to the Author:

Summary

This manuscript attempts to address the challenge of identifying causal genes for trans non-coding genetic risk variants, and the cell types in which their phenotype manifests, by constructing genetically diverse murine macrophage gene regulatory networks correlated with cellular function. The study's main contribution is to link natural genetic variation in several common mouse strains with Kupffer cell (KC) function by identifying strain-differential genes via RNAseq, ATACseq, and H3K27ac ChIPSeq in vivo and in vitro. This is a model for understanding the effects of genetic variation in other immune cell types. However, the findings do not knit together to clearly explain strain-specific fibrosis

protection or into a clearly defined differential gene regulatory network. The manuscript would fit better with Genes & Immunity due to its focused impact on immunogenomics research, potentially classified as a resource paper.

Denotes Major Comments | Minor Comments

Results

Figure 1 – defines transcriptional strain differences in KCs and BMDMs and implicates some immune-functions in that difference

Figure 1a – why was a TPM threshold of 8 chosen? How many genes clear that threshold, and are they all displayed in the main figure?

1e-j: what constitutes being a AJ/Specific, BLAB/CJ specific or C57BL/6J specific gene?

Recommend condensing Extended Data Figure 1 & 2 into one

Figure 2 – defines chromatin strain differences in KCs using ATACseq and H3K27ac ChIPseq, and implicates strain-specific differences at loci of trans-acting immune-related genes

Recommend using UpSetPlots instead of Venn diagrams

Figure 3 – Defines strain-specific differences in hepatocyte, SC, and LSEC ligand-receptor usage (e.g. signaling), and IDs Leptin signaling as a driver of strain-specific transcriptional difference in KCs
How was Leptin signaling and Leptr expression selected for further follow-up out of the NicheNet top scoring ligands?

Is leptin signaling also strain-specifically regulated in the NSG transplant model?

Figure 4 – Analysis of strain-specific transcriptomics differences of KCs in NSG transplant model

Recommend using UpSetPlots instead of Venn diagrams

Figure 5 – Analysis of ATACseq and H3K27a ChIPseq differences from an experiment in Figure 4

How are the enriched motifs from ChIPseq connected to the transcriptional differences ID'd in figure 4? Are these genes (in 5d/e) the top-regulated genes in 4? Where are they in 4b/c?

Figure 6

Please define your differentially regulated genes and motifs from Figures 1-5 in this LPS model. Are they also differentially responding to LPS?

Methods

Why were DESeq 2 and HOMER chosen for transcriptomics analysis instead of limma and ComBat?

Reviewers were not given access to GSE216164; thus it is not possible to confirm that all datasets generation during the study are available (please provide if a revision review is initiated)

Extended Data Table 1 & 2 should be provided in the .csv format, not as PDFs

All code for any analysis and generated figures should be provided to the public in Git repo; it is not sufficient to provide it upon request

Your processed datasets should be deposited at figshare, specifically

Mapped read count files to mm10 and BLAB/CJ genome builds for all transcriptomics and epigenomics

NicheNet object, and full model output files

DESeq2 objects, and full model output files

Discussion

What is the key story in the extensive data generated? A toy model or figure illustrating the key strain-specific differences in KCs would be helpful to the reader

Author Rebuttal to Initial comments

Detailed responses to Reviewers' comments

We thank the reviewers for their time in consideration of our manuscript, and for the constructive comments provided. We address these comments as italicized and inset point-by-point responses below.

Reviewer #1

Glass and colleagues have performed a solid study focused on integrating genetic and epigenetic contributions to Kupffer cell phenotypes. Kupffer cells are relatively understudied using the most modern approaches, so the data the authors have produced are an important contribution and they should be broadly useful.

In the context of the above, several concerns arise ranging from the primary data to the experimental setup, to the presentation of the results.

We thank reviewer 1 for their positive feedback and constructive attention to our manuscript. We address specific concerns below.

General comments:

- The primary epigenomic data are voluminous but there is little if any indication of data quality; well-used measures such as FRIP are not presented. More importantly, there is no indication in the text or methods specifying how the authors handled differences in data quality between two data sets when making comparisons.

We agree that quality metrics are an important component of peer review and we regret not providing these parameters in the original submission. We have compiled these summary statistics, including FRIP, library clonality, and total reads in Extended Data Table 4. We have also provided a set of principal component analyses illustrating the dominant effect of strain and LPS treatment on our data. Overall, the data quality is highly comparable across sample groups, allowing us to perform analysis without directly adjusting for differences in quality between samples.

- The finding of strain-specific differences environmental factors influencing Kupffer cell phenotypes is not unexpected. In this context, the authors highlighted phenotypic differences in strain susceptibility and resistance to NASH but did not comment on general mechanisms of how the observed genetic variation might influence differential susceptibility to NASH

Linking cell-specific effects of natural genetic variation to mechanisms underlying the differences in strain susceptibility and resistance to NASH is a major long-term goal. In fact, this was one of the original objectives of these studies. However, in the course of performing this work, it became evident that pursuing any one of the leads that emerged from our systematic analysis

would be a major undertaking and that the work would be of greatest impact by clearly establishing the quantitative contributions of cis and trans effects on Kupffer cell transcriptomes and epigenetic landscapes and providing the underlying data sets as a community resource. It was for this reason that we did not present the data on NASH as a main figure or refer to NASH in the abstract. However, to respond directly to this comment, we revised the discussion to include the following statement: Lines 436-445

'An important future objective will be investigation if observed effects of natural genetic variation on Kupffer cell gene expression are linked to mechanisms of liver diseases. Overall, our findings suggest that interactions of multiple genes and cell types make quantitative contributions to phenotypic differences, in line with implications of most genome wide association studies. However, our findings also indicated strain-specific pathways correlated with sensitivity or resistance to the NASH phenotype. Notable candidate pathways identified here include signaling through the leptin receptor, which has anti-inflammatory roles in Kupffer cells and is expressed highest by Kupffer cells from NASH-resistant BALB/cJ mice. Additionally, Kupffer cells from the NASH-sensitive C57BL/6J strain had increased trans-acting chromatin activity at elements predicted to bind NF κ B, a transcription factor strongly linked to macrophage-mediate pathogenesis in many chronic inflammatory diseases.'

- The authors propose that strain-specific Kupffer cells engrafted into a common host enabled quantitative assessment of cis versus trans effects of genetic variation on gene expression, and also quantification of cell autonomous effects. However, the presentation of these results is rather opaque. The authors should consider partitioning the text and figure panels into a more logical flow.

We thank Reviewer 1 for the comment and note that Reviewer 2 also suggested modification of Figure 4 and the corresponding text. In response to these comments, we have provided additional analyses to link trans genes to specific transcriptional pathways and restructured Figure 4 to emphasize how the use of both NSG transplant model and the F1 hybrid model allow for quantitative assessment of cis and trans genetic variation in Kupffer cells. (Lines 240-288)

First, we sought to connect the findings in our study to findings in human genetics research. Large scale human cohort studies have shown that cis effects tend to have larger effect size than trans effects, while the total number of trans variants outnumbers cis effects (Liu et al., Cell, 2019, PMID 31051098, Battle et al., Genome Research 2014, PMID 24092820). We show that our cell-specific model of in vivo genetic variation agrees with human eQTL research, as the parental fold change in expression associated with trans regulated genes is smaller than the fold change associated with cis regulated genes (Figure 4c). We also note that our estimate that ~60% of gene expression is driven by trans effects is similar to the estimate made by previous studies of proteome quantitative trait loci from genetically diverse outbred mice (Chick et al., 2016 Nature, PMID 27309819).

We note that recent work studying complex traits in human genetics shows that trans eQTLs associated with disease tend to converge on modules of co-expressed genes (Vosa et al., Nature Genetics 2021, PMID 34475573; Liu et al., Cell, 2019, PMID 31051098). Therefore, identifying trans regulated gene modules can discern the biology underlying divergent complex traits. We hypothesized that C57BL/6J and BALB/cJ specific Kupffer cell trans genes would therefore be enriched for distinct pathways present in the strain-specific extracellular environment. We found that BALB/cJ and C57BL/6J trans genes were enriched for distinct gene ontology terms and that

promoters of BALB/cJ and C57BL/6J trans genes were enriched for distinct transcriptional regulators (Figure 4d-e). Notably, C57BL/6J Kupffer cell trans genes were enriched for genes involved in leukocyte activation and TLR signaling, while C57BL/6J trans promoters were enriched for the NFkB-p65 motif, suggesting that differential activation of NFkB signaling may be responsible for a portion of the differential effects seen in parental strains. This finding suggests that trans gene analysis can be used to recover different strain specific environmental regulators.

We agree with Reviewer 2 and Reviewer 3 that the venn diagram analysis provided limited insight when comparing differentially expressed transcripts across the three models. In response to this, we replaced the venn diagrams with an Upset plot showing the overlap between C57BL/6J and BALB/cJ specific transcripts across the three comparisons (Figure 4f). This figure emphasizes that the F0-NSG model shares more strain specific genes with the parental F0 strains in comparison with the F1-NSG model.

Taken together, the new analyses emphasize the key finding that the engraftment of strain specific Kupffer cells into a common host allows for discrimination and quantification of cell intrinsic versus cell extrinsic genetic variation, suggesting that roughly 2/3 of trans acting variation exerts its effects through cell-extrinsic (environmental) mechanisms.

- It would be of value to compare hepatocytes, hepatic stellate cells, endothelial cell and other supportive cell types in the liver to improve desegregation of cell autonomous vs non-cell autonomous effects during NASH progression

We agree that considerable insight could be derived from further analysis of the other cell types of the liver. Towards this goal, the transcriptomes of hepatocytes, stellate cells and sinusoidal endothelial cells presented in this manuscript provide a strong starting point for this effort and thereby represent what we expect will be a valuable resource. We note that extending the analysis of each cell type at the level described here for Kupffer cells would go beyond the scope of what can be presented in a single paper. However, we wish to emphasize that our focus in these studies is directed at Kupffer cells because they are an important example of a tissue resident macrophage that adopts an environment-dependent phenotype, as exemplified in Figure 1A and Extended Data Figure 2. It is therefore our intention to position this work as a general approach to understanding how natural genetic variation influences macrophage phenotypes for the readers of Nature Immunology.

- The authors could consider passaging F1 hybrids into NSG mice as a stronger comparator to the C57BL/6J and BALB/cJ NSG transplants

We thank Reviewer 1 for this suggestion, also suggested by Reviewer 2. We agree that transplantation of F1 HSCs into NSG mice will better control for possible confounding effects of the busulfan HSC transplant protocol. Additionally, as stated by reviewer 2, this new experiment controls for gene-environment effects that arise from differences in ontogeny. Comparisons of F0 and F1 cells in the NSG transplant model is necessary to exclude gene-environment effects that arise from differences in cellular ontogeny (embryonic KCs versus monocyte origin KCs). A notable limitation with this system is that gene-environment effects existing only in embryonic Kupffer cells are missed.

We transplanted HSCs from CB6F1/J mice into busulfan depleted NSG recipients. The resulting F1-NSG hybrids were maintained for 8 weeks to allow CB6F1/J bone marrow derived monocytes

to enter the liver and differentiate into Kupffer cells. We then sorted CB6F1/J Kupffer cells and performed RNA-sequencing on them. We compared RNA-seq results from parental F0 mice, F0 Kupffer cells harvested from transplanted NSG mice (F0-NSG) and F1 Kupffer cells harvested from transplanted NSG mice (F1-NSG).

We found that the total number of strain-specific and allele specific genes were similar in both studies (244 strain-specific genes in the F0-NSG, 218 allele-specific genes in the F1-NSG). Comparison of strain specific genes from the three models showed that parental F0 Kupffer cells and F0-NSG Kupffer cells shared more strain specific genes than either F0 model shared with the allele specific genes in the F1-NSG model. This consistent with the role of the intracellular environment in maintaining a portion of strain specific gene expression (Fig. 4g, Extended Data Figure 6).

We then directly comparing trans genes across the three model systems to resolve cell-intrinsic and cell-extrinsic trans effects. The F0-NSG transplants and F1-NSG transplant Kupffer cells share a similar hepatic extracellular environment. Thus, genes with strain-specific gene expression bias in parental Kupffer cells and equivalent expression in F0-NSG and F1-NSG Kupffer cells are likely driven by environmental differences unique to the parental liver (Fig. 4f). In contrast, donor derived Kupffer cells from the F1-NSG model contain both parental genomes in a shared cellular and nuclear environment. Consequently, genes exhibiting strain-specific expression in Kupffer cells from parental mice and the F0-NSG transplant model and equivalent expression in donor-derived F1-NSG model likely arise from differences in intracellular signaling or transcription factor activity. We found overall similar results using the F1-NSG model compared to the F1 embryonic derived Kupffer cell model. The majority (71/121) of transcripts with established allelic bias were trans-regulated in both models and likely driven by environmental trans effects in the parental hepatic environment that are lost in both the NSG and F1-hybrid models. Many remaining transcripts (42/121) were trans-regulated in F1-hybrids but not the engraftment model, suggesting they are caused by cell-autonomous differences in upstream of transcription factor activity. A small proportion (8/121) genes were trans-regulated in NSG chimeras but not F1-hybrids. Examples of each type of trans gene are shown in Fig. 4h. Based on these data, we estimate that 60% of trans mediated genetic variation in Kupffer cells is driven by cell non-autonomous differences in environmental signals, while 40% is due to cell-autonomous differences in signaling or transcriptional activity.

- The authors showed that changes in *Lepr* expression in Kupffer cells from C57BL/6 mice are associated with NASH induction. It is not clear how this finding relates to NASH susceptibility. Do similar/dissimilar expression changes occur upon feeding NASH-resistant mice an AMLN diet, and could they contribute to differences in NASH susceptibility?

*Expression of *Lepr* was highest in Kupffer cells from NASH resistant mice (Fig. 3d). Further, NASH resistant mice had minimal accumulation of monocyte-derived Tim4⁺ Kupffer cells during experimental NASH, in stark contrast to livers from NASH susceptible C57BL/6J mice. This is relevant because ablation of embryonic Kupffer cell with a genetic model (*Clec4f-Cre R26-*Isl-DTR**) demonstrated that monocyte-derived Kupffer cells may be incompetent for normal *Lepr* expression as observed in steady state embryonic Kupffer cells (Fig. 3g). Although these data are intriguing correlates, we cannot conclude that a causal relationship exists between KC expression of *Lepr* and protection from experimental NASH without performing formal experiments disrupting *Lepr* expression in Kupffer cells of mice of NASH susceptible and NASH-resistant genetic backgrounds. However, we now specifically mention this relationship in the discussion as a*

direction for future research as indicated in our response to the previous question on NASH susceptibility from Reviewer 1. (Lines 436-445)

- The analysis of parental vs NSG-passaged vs F1 hybrid Kupffer cells to untangle mechanisms related to cis vs trans and cell-intrinsic vs cell-extrinsic gene regulation is intriguing, but additional clarifying text will aid interpretation and relevance of the findings

We thank the reviewer for this feedback. We have sought additional feedback from colleagues regarding clarification of our primary message and hope our revised text is a substantial improvement. (Lines 263-280)

Specific comments:

- Scaling the RNA expression data, e.g., $\log_2(\text{TPM}+1)$, may improve interpretability of the gene expression bar plots. As well, it is not clear if statistical tests were performed for gene expression comparisons (Fig 3d/f/g, Fig 5e, Fig 6c)

We thank Reviewer 1 for pointing out the omission of statistical testing for gene expression comparisons. We have added notations to the revised figures and their corresponding legends indicating the statistical tests performed in Figure 3d/f/g, 5e, and 6c. We have also used \log_2 transformed TPM values to improve interpretability of bar charts with large differences in expression such as 3f, 3g, and 6c.

- There could be additional explanatory text to accompany the NicheNet analysis. In Figure 3c, it is not stated what the width of each line in the circos plot represents

We thank Reviewer 1 for bringing this to our attention. We have added clarifying text accompanying Figure 3 and additional text in the methods explaining our use of NicheNet (Lines 200-212). NicheNet is a model of cellular signaling built from public data sources. The NicheNet model is a matrix receptor by gene matrix of weighted scores predicting the ability of a particular receptor to activate expression of target genes. NicheNet ranks ligands by performing Pearson correlation of a vector of differential genes ('1' for a differential gene, '0' for a non-differential gene) with the target gene prediction row of each receptor. We have also added text to the Revised Figure 3 legend stating that the width of each line in the circos plot represents the NicheNet model's ligand activation score for a receptor-gene interaction (Line 519).

- In Figure 6 and the accompanying text, the delineation between 'low basal', 'similar basal', and 'high basal' is not intuitive and seem to describe both basal and LPS-induced gene expression, which is not clear by the category names alone. In some cases, 'similar basal' and 'equal basal' are used interchangeably. Further clarification of how the categories were formed will be helpful

We thank Reviewer 1 for pointing out that the definitions of the 'low basal', 'equal basal', and 'high basal' were incompletely described in the original manuscript. These categories were used to separate mRNA transcripts with strain specific LPS induction (i.e. transcripts that are induced by LPS in C57BL/6J Kupffer cells but not in BALB/cJ Kupffer cells). Low basal transcripts are associated with low expression in the strain lacking LPS induction, while high basal transcripts are associated with high expression in the strain lacking LPS induction. Equal basal transcripts have equivalent expression in each strain, but only display LPS induction in one strain. We

showed in a recent paper that these categories are associated with distinct mechanisms of epigenetic regulation in vitro using bone marrow derived macrophages treated with IL-4 (PMID: 3632331). We have added text to the revised manuscript that clarifies this point (Lines 350-358).

- It is unclear from the text what is meant by “LPS responsive strain” and “LPS non-responsive strain” (line 354-355, 370-373). Perhaps this should be changed to strain-specific LPS responsive genes or strain-specific LPS responsive enhancers?

We thank Reviewer 1 for raising this question. We clarified in the revised manuscript that it is the genes that are selectively responsive in the strain and not the entire strain itself that is LPS responsive. This is an important correction (Lines 342-358).

- The legend corresponding to Extended Data Figure 1f indicates that bone marrow derived macrophages and Kupffer cells were hierarchically clustered, but Extended Data Figure 1f only shows clustering of Kupffer cells

We thank Reviewer 1 for bringing this error to our attention. It has been corrected in the revised manuscript.

Reviewer #2

(Remarks to the Author)

This paper seeks to analyze the effects of natural genetic variation on Kupffer cell gene expression through use of three inbred mouse strains (A/J, BALB/cJ, C57BL/6J) that display phenotypic differences in liver disease (in particular non-alcoholic steatohepatitis, NASH) as well as chimeric and F1 hybrid mouse models based on these strains. The study tries to quantify the extent of cis vs. trans effects of genetic variation on gene expression, and of cell-autonomous vs. non-cell-autonomous trans effects, through a fairly complex experimental design:

(i) first RNA-seq data is collected in the three strains in both Kupffer cells and bone-marrow derived macrophages (BMDMs) to identify strain-differential gene expression specific to Kupffer cells (Fig. 1);

(ii) ATAC-seq and H3K27ac ChIP-seq is next generated to identify shared and differentially accessible active enhancers and non-acetylated (“poised”) enhancers and to identify motifs associated with strain-specific active enhancers (Fig. 2);

(iii) a computational model of cellular signaling called NicheNet on RNA-seq from hepatocytes, stellate cells, and liver sinusoidal cells (LSECs) in these strains to ask if ligands expressed by these cell types could predict strain-specific gene expression patterns in Kupffer cells; follow-up validation experiments for one of the predictions — the effect of the top scoring ligand Lep, active in the BALB/cJ (and to a lesser extent, C57BL/6J) liver niche but not in A/J, on STAT3 phosphorylation downstream of Lepr signaling — are performed (Fig. 3);

(iv) to try to quantify cis vs. trans and cell-autonomous vs. non-cell-autonomous effects between BALB/cJ and C57BL/6J, bone marrow from these strains is transplanted into NOD mice that have been treated with busulfan to deplete resident Kupffer cells, allowing generation of monocyte-derived Kupffer cells from donor progenitors, and strain-specific expression differences (with matching NOD host environments) are compared to allele-specific effects in Kupffer cells in F1 mice, though there are caveats to this design (Fig. 4);

(v) ATAC-seq and H3K27ac ChIP data in parental vs. F1 Kupffer cells are compared to identify cis vs.

trans effects (Fig. 5);

(vi) strain-specific expression and ATAC-seq response to LPS stimulation (for BALB/cJ and C57BL/6J) are examined and coupled with motif analysis to find stronger cis effects in this setting (Fig. 6).

This is an interesting if somewhat descriptive study addressing important questions of how natural genetic variation impacts gene expression; given the strong focus on cis effects in current studies of the genetics of gene regulation, the authors take a valuable wider view to consider the widespread impact of trans effects, including non-cell-autonomous effects. The take-home message seems to be this statement from the Discussion: “In concert, systematic analysis of the transcriptomes and regulatory landscapes of Kupffer cells in A/J, BALB/cJ and C57BL/6J mice, in NSG chimeras, and in F1-hybrids, reveals dominant trans effects of natural genetic variation under homeostatic conditions and dominant cis effects on the response to LPS.” However, there are missing experimental and computational analysis steps in reaching this conclusion. While aspects of the analysis — like mapping sequencing data to F1 hybrid genomes — demonstrate the computational expertise of this lab, other parts are less satisfying — like trying to make quantitative arguments in Venn diagram fashion after imposing a significance threshold. The experiments outlined in point (iv) have clear caveats in comparing monocyte-derived Kupffer cells in immunodeficient mice to Kupffer cells of non-hematopoietic origin in immunocompetent F1 mice; there is also the missing comparison of F1 bone marrow transferred to NSG mice. In places, the analysis depends on previously published but not yet widely used algorithms (MAGGIE motif analysis, NicheNet) that would benefit from some background exposition. More comments are given below.

We thank Reviewer 2 for their comprehensive and constructive review of our manuscript. The summary encapsulates most of the messages we tried to convey. We would clarify that in point (vi), we utilized motif mutation analysis, not motif analysis, in an attempt to capture functional outcomes of evolution on a layer of transcriptional regulation in Kupffer cells. We recognize this clarification was probably understood by Reviewer 2 based on their mention of our use of MAGGIE in the following paragraph.

We recognize and agree that Venn diagram fashion analyses are not the best quantitative analytical method. We used this approach because the methods we adopted for analysis of allelic biases in F1 experiments required filtering of sequencing reads containing strain-specific genetic variants. As a consequence, we were not able to *directly* compare data sets from F0/NSG to F1 because the variant filtered F1 sequences are derived from a very different read distribution. In the revised manuscript we sought to improve the quantitative analysis associated with Figure 4.

In our original submission we found that *trans* genes outnumbered *cis* genes at a fold change cutoff of 2, but we recognize the potential for thresholding bias in the analysis of the relative proportion of cis and trans effects presented in the original version of Figure 4. To address the concern, we assessed the total number of cis and trans genes and their cumulative distribution at three different fold change thresholds (2, 1.5, and 0 – the fold change 0 threshold simply requiring the adjusted p value to meet significance threshold of 0.05)

a-c, cumulative distribution of fold change for cis and trans genes at different fold change thresholds. d, figure reproduced from Liu et. al, Cell, 2019, 31051098, showing relative effect size (quantified here as |Z|, where Z² is the relative amount of genetic variance associated with each SNP) of cis and trans-eQTL variants identified in study of human blood eQTLs. e, table showing fractional contribution of trans effects to differential genes identified in parental (F0) Kupffer cells from C57BL/6J and BALB/cJ mice.

At each of these fold change thresholds, trans genes outnumber cis genes, but cis genes retain a larger effect size as evidenced by the significant difference in the cumulative distributions, indicating that our results are robust to bias introduced by thresholding. Notably, this pattern has also been observed in human eQTL studies, as in the figure shown here (reproduced from Liu et. al, Cell, 2019, 31051098).

In addition, our estimate of the fractional contribution of trans effects to gene expression is quite stable across these three fold change thresholds, and is similar to the estimate made using data from a prior study of bulk mouse liver transcriptomics and proteomics (~60% of gene expression heritability in trans, Chick et al., Nature 2016. PMID 27309819) as well as estimates derived from

human genetics (Liu et al., Cell, 2019, PMID 31051098). These data support our conclusion that a majority of strain specific differences in mRNA expression are driven by trans acting genetic effects.

Major comments

(1) Initial expression analysis. The heatmap in Fig. 1a seems minimally informative and at least should be larger. Perhaps the clusters of genes displaying different strain-specific patterns (from the hierarchical clustering) could be explored using gene ontology enrichments?

We agree that Fig. 1a was minimally informative for interpreting the relative contributions of genetic effects on macrophage transcriptomes. Our reason for displaying the data in this fashion was because the heatmap demonstrated that environmental effects (in vitro versus in vivo) dominated the transcriptional differences within the data set as a whole, in agreement with other studies. This observation provided impetus for identifying the genetic contribution to Kupffer cell transcription in physiological relevant settings due to the sensitivity of macrophage transcription to environmental differences. To make this point we have replaced Fig 1a. with a principal component analysis of the global data.

Additionally, we have expanded on the original global gene expression analyses as requested, and present the results in a new Extended Data Figure 2 and Extended Data Table 1. This figure includes the requested larger heatmap originally in Fig. 1A, as well as subsets focusing on cell type-specific differential genes between the three strains. These three heatmaps together demonstrate that the environmental effect on gene expression is larger than the still large genetic component. Further, we have included the top gene ontology terms for the major hierarchical clusters in Extended Data Figure 2. These changes support our original purpose in defining environmental effects and facilitate interpretation of strain-specific effects on macrophage transcription. We are grateful for this recommendation.

(2) Initial epigenomic analyses. The term “poised” is not standard for accessible peaks lacking H3K27ac (often connotes H3K4me1 signal) and should probably not be used here. A few sentences of exposition on MAGGIE would be useful to orient the reader. Moreover, since the MAGGIE paper describes a statistical test for pairs of sequences, i.e. coming from a pair of genomes, it is not clear how these pairwise analyses across 3 strains are summarized in Fig. 2g.

We agree that the term poised is often used to refer to regions marked by H3K3me1, we have revised the manuscript to use the more appropriate terms “accessible” and “open” when referring to regions marked by ATAC sequencing. We also recognize that H3K27ac is a surrogate of activity and not a direct measure of activity, and state this directly in our use of this mark (Fig. 2, Lines 131-193, 499-511)

We have expanded the methods section to provide additional explanation for how multiple pairwise comparisons are integrated to generate the significance values presented in Fig. 2g (Lines 814-826). Briefly, the MAGGIE model analyzes differences in motif score between “positive sequences” that have an epigenetic feature of interest (H3K27Ac ChIP-seq, ATAC-seq,

transcription factor specific ChIP-seq) in comparison to “negative” sequences lacking that feature in a corresponding mouse strain. In order to compare the effect of mutations on epigenetic features across three strains in one analysis, we concatenated positive sequences (and their corresponding negative sequences) from each of the six possible pairwise comparisons of the strains into one MAGGIE analysis.

(3) NicheNet results. A brief description of NicheNet would be helpful. The main result for which there are follow-up experiments — i.e. the strain-specific effect of the ligand Lep on BALB/cJ, shown in Fig. 3b — is a bit confusing since it seems that none of the niche cell types seem to express much Lep (in any strain) based on Fig. 3a. Where is the endogenous Lep coming from? The immunoblot assessment of phospho-STAT3 in liver treatment after leptin injection is not a cell type specific assessment showing the response is limited to Kupffer cells, though *Lepr* expression seems limited to Kupffer cells. It would seem important to fill in some of the gaps in these validation experiments.

We agree with Reviewer 1 and Reviewer 2 that an expanded explanation of the NicheNet analysis will benefit the reader. We have expanded on the use of NicheNet as described above in the response to Reviewer 1 (Lines 200-212, 824-829)

With regards to the identification of leptin, we note that Kupffer cells reside in the hepatic sinusoids and are exposed to portal blood. Given this location, we decided to include a selection of hormones in our NicheNet analysis that are present in blood but not expressed in the liver itself. Endogenous leptin is exclusively derived from adipose tissue. This point has been clarified in the revised manuscript (Lines 223-224).

We agree that phospho-STAT3 is not cell specific, but as pointed out by Reviewer 2, BALB/cJ and C57BL/6J Kupffer cells express at least 2-3 fold more leptin receptor mRNA than other hepatic cell types (Figure 3d).

(4) Chimeric mouse experiments. This is an interesting but complex experimental design with many caveats. Comparing hematopoietic-derived Kupffer cells in immunodeficient mice with embryonic-derived Kupffer cells in immunocompetent F1 mice seems problematic; cross-referencing against published hematopoietic vs. embryonic Kupffer cell expression changes is inadequate to address the issue. Perhaps the physiologic stress of the busulfan treatment and bone marrow transplant contributes to non-cell-autonomous trans effects here, and these effects are not necessarily relevant in the context of e.g. genetic variation and liver disease? At minimum, there is a missing comparison of F1 bone marrow transplantation into NSG mice; these (hematopoietic) F1 Kupffer cells could be directly compared with the corresponding F0 strain Kupffer cells in a matching NSG host environment to address cell-autonomous trans effects. In general, the quantitative analysis is a bit weak here — based on Venn diagram style quantification.

We agree that transplantation of F1 HSCs into NSG mice will better control for possible confounding effects of the busulfan HSC transplant. Additionally, as stated by reviewer 2, this new experiment controls for gene-environment effects that arise from differences in ontogeny. Comparisons of F0 and F1 cells in the NSG transplant model is necessary to exclude gene-environment effects that arise from differences in cellular ontogeny (embryonic KCs versus

monocyte origin KCs). A notable limitation with this system is that gene-environment effects existing only in embryonic Kupffer cells are missed.

We transplanted HSCs from CB6F1/J mice into busulfan depleted NSG recipients. The resulting F1-NSG hybrids were maintained for 8 weeks to allow CB6F1/J bone marrow derived monocytes to enter the liver and differentiate into Kupffer cells. We then sorted CB6F1/J Kupffer cells and performed RNA-sequencing on them. We compared RNA-seq results from parental F0 mice, F0 Kupffer cells harvested from transplanted NSG mice (F0-NSG) and F1 Kupffer cells harvested from transplanted NSG mice (F1-NSG).

We found that the total number of strain-specific and allele specific genes were similar in both studies (244 strain-specific genes in the F0-NSG, 218 allele-specific genes in the F1-NSG). Comparison of strain specific genes from the three models showed that parental F0 Kupffer cells and F0-NSG Kupffer cells shared more strain specific genes than either F0 model shared with the allele specific genes in the F1-NSG model. This consistent with the role of the intracellular environment in maintaining a portion of strain specific gene expression (Fig. 4g, Extended Data Figure 6).

We then directly comparing trans genes across the three model systems to resolve cell-intrinsic and cell-extrinsic trans effects. The F0-NSG transplants and F1-NSG transplant Kupffer cells share a similar hepatic extracellular environment. Thus, genes with strain-specific gene expression bias in parental Kupffer cells and equivalent expression in F0-NSG and F1-NSG Kupffer cells are likely driven by environmental differences unique to the parental liver (Fig. 4f). In contrast, donor derived Kupffer cells from the F1-NSG model contain both parental genomes in a shared cellular and nuclear environment. Consequently, genes exhibiting strain-specific expression in Kupffer cells from parental mice and the F0-NSG transplant model and equivalent expression in donor-derived F1-NSG model likely arise from differences in intracellular signaling or transcription factor activity. We found overall similar results using the F1-NSG model compared to the F1 embryonic derived Kupffer cell model. The majority (71/121) of transcripts with established allelic bias were trans-regulated in both models and likely driven by environmental trans effects in the parental hepatic environment that are lost in both the NSG and F1-hybrid models. Many remaining transcripts (42/121) were trans-regulated in F1-hybrids but not the engraftment model, suggesting they are caused by cell-autonomous differences in upstream of transcription factor activity. A small proportion (8/121) genes were trans-regulated in NSG chimeras but not F1-hybrids. Examples of each type of trans gene are shown in Fig. 4h. Based on these data, we estimate that 60% of trans mediated genetic variation in Kupffer cells is driven by cell non-autonomous differences in environmental signals, while 40% is due to cell-autonomous differences in signaling or transcriptional activity.

As discussed in our initial response to Reviewer 2, we have sought to improve the quantitative analysis of this section by incorporating motif analysis and linking the results to work in human eQTL studies.

Minor comments

- "Dependent on context, as much as 80% of allele-specific differences in cis regulatory activity can be linked to local variants" — please provide a citation for this statement^[1]_{SEP}

The statement was taken from our prior analysis of effects of natural genetic variation on gene expression in bone marrow derived macrophages (Link et al., Cell 2018, PMID 29779944). We will provide the reference in the revised manuscript (Lines 56-57).

- It might be worth discussing that the pseudo genome alignment strategy, which edits the reference genome using the VCF file, ignores large structural variants that may contain important regulatory elements.

We thank Reviewer 2 for raising this point. It is correct that we have not considered large structural variants. These are relatively infrequent in comparison to SNPs and InDels in the strain comparisons performed in these studies but could account for some strain specific differences. We include a statement referring to this possibility in the methods of the revised manuscript (Lines 757-760).

- Some phrase left out on line 206: "asked whether ligands expressed by could predict"
- Fig. 5: typos in labeling, "prental" -> "parental"s

Thank you for catching these mistakes. We have corrected both errors.

Reviewer #3

(Remarks to the Author)

Summary

This manuscript attempts to address the challenge of identifying causal genes for trans non-coding genetic risk variants, and the cell types in which their phenotype manifests, by constructing genetically diverse murine macrophage gene regulatory networks correlated with cellular function. The study's main contribution is to link natural genetic variation in several common mouse strains with Kupffer cell (KC) function by identifying strain-differential genes via RNAseq, ATACseq, and H3K27ac ChIPSeq in vivo and in vitro. This is a model for understanding the effects of genetic variation in other immune cell types. However, the findings do not knit together to clearly explain strain-specific fibrosis protection or into a clearly defined differential gene regulatory network. The manuscript would fit better with Genes & Immunity due to its focused impact on immunogenomics research, potentially classified as a resource paper.

We thank Reviewer 3 for the careful review of our manuscript and the appreciation of its relevance to understanding the effects of genetic variation in other immune cell types. We wish to emphasize that while using the data sets to advance understanding of strain-specific fibrosis protection is a long-term goal, it is not the goal of this manuscript. The strains used for these studies were initially chosen because of their different responses to a NASH inducing diet as a starting point for this effort and to enhance the resource value of the corresponding data sets. However, our findings (in agreement with the field) are most consistent with multiple genes, acting in multiple cell types, having quantitative contributions to these phenotypic differences. Because each candidate gene becomes a subject for in depth analysis, performing these additional studies would go beyond the scope of what can be presented in a single manuscript. In preparing the manuscript, we were careful to not use the terms NASH or fibrosis in the abstract and we made no specific conclusions about mechanisms underlying strain-specific phenotypes. We further clarify the point that our findings support roles of multiple genes acting in multiple cell types to determine strain-specific susceptibility to NASH which has been added in the Discussion (Lines 436-445) and is included below.

'An important future objective will be investigation if observed effects of natural genetic variation on Kupffer cell gene expression are linked to mechanisms of liver diseases. Overall, our findings suggest that interactions of multiple genes and cell types make quantitative contributions to phenotypic differences, in line with implications of most genome wide association studies. However, our findings also indicated strain-specific pathways correlated with sensitivity or resistance to the NASH phenotype. Notable candidate pathways identified here include signaling through the leptin receptor, which has anti-inflammatory roles in Kupffer cells and is expressed highest by Kupffer cells from NASH-resistant BALB/cJ mice. Additionally, Kupffer cells from the NASH-sensitive C57BL/6J strain had increased trans-acting chromatin activity at elements predicted to bind NF κ B, a transcription factor strongly linked to macrophage-mediate pathogenesis in many chronic inflammatory diseases.'

With respect to a clearly defined transcriptional network, we think that the present studies confirm and extend prior work demonstrating important roles of various transcription factors and upstream signaling pathways in establishing Kupffer cell identity and function. However, the major conceptual advance that we hope to convey is that the systematic analysis of epigenetic landscapes from genetically diverse mice can be used as a way of investigating how enhancers integrate cis and trans effects of non-coding genetic variation. We agree that the manuscript would fit with Genes and Immunity, but we also think that the approach and findings would be of interest to the broader readership of Nature Immunology.

Denotes Major Comments | Minor Comments

Results

Figure 1 – defines transcriptional strain differences in KCs and BMDMs and implicates some immune-functions in that difference

Figure 1a – why was a TPM threshold of 8 chosen? How many genes clear that threshold, and are they all displayed in the main figure?

We chose the TPM threshold of 8 to provide a sufficiently high level of expression in at least one strain that genes exhibiting expression below this level in other strains could be confidently determined to be strain specific. We also considered the TPM threshold to be enriched for genes that are expressed at functionally significant level. This is the threshold that we applied consistently across the RNA-seq analyses in this paper. All genes meeting this threshold are displayed in Figure 1a.

1e-j: what constitutes being an AJ/Specific, BLAB/CJ specific or C57BL/6J specific gene?

Recommend condensing Extended Data Figure 1 & 2 into one

Strain specific genes were selected if they were expressed at a significantly higher level in one strain compared to both other comparison strains (i.e. Irak3 is expressed higher in C57BL/6J Kupffer cells than in both BALB/cJ and A/J Kupffer cells and is therefore C57BL/6J strain specific). We have clarified this description in the legend to Figure 1 in the

revised manuscript and are grateful for pointing out the need for improved clarity (Lines 488-491).

We prepared a draft figure condensing Extended Data Figures 1 and 2 into a single Figure, but the panels do not fit on a single printable page at readable font sizes. Further, Extended Data Figure 2 now includes additional analyses targeted to improving the informative nature of our studies.

Figure 2 – defines chromatin strain differences in KCs using ATACseq and H3K27ac ChIPseq, and implicates strain-specific differences at loci of trans-acting immune-related genes
Recommend using UpSetPlots instead of Venn diagrams

We thank Reviewer 3 for bringing our attention to this useful form of data visualization. We agree that UpSet plots better capture the structure of our epigenomic data and emphasize the fact that the majority of open and active chromatin regions are shared across the three strains in our study. All Venn diagrams have been converted to UpSetPlots in the revised manuscript (Figure 2c, Figure 4g).

Figure 3 – Defines strain-specific differences in hepatocyte, SC, and LSEC ligand-receptor usage (e.g. signaling), and IDs Leptin signaling as a driver of strain-specific transcriptional difference in KCs
How was Leptin signaling and Lepr expression selected for further follow-up out of the NicheNet top scoring ligands?

As discussed in the response to reviewer 2, leptin is expressed exclusively by adipocytes and thus was not a gene represented in the RNA-seq data sets obtained from hepatocytes, stellate cells, endothelial cells or Kupffer cells. However, given Kupffer cell's physiologic location in the liver sinusoids we included additional hormonal signals derived from other sources that could act on receptors expressed by Kupffer cells, including Leptin. Leptin was amongst the top 5 scoring ligands from our NicheNet analysis of BALB/cJ specific genes and is of interest given its role in energy homeostasis and metabolic syndrome. We elected to validate the hypothesis that Leptin signaling would exhibit strain specific differences because of the differential expression of the Leptin receptor in A/J Kupffer cells compared to BALB/cJ and C57BL/6J Kupffer cells and the ability to readily detect the consequences of Leptin receptor activity at the level of the whole liver by assessing the phosphorylation status of STAT3 (Mageda et al., PNAS 2009, PMID 19620723; Kiguchi et al., Cell Metabolism 2012, PMID 22768838).

Is leptin signaling also strain-specifically regulated in the NSG transplant model?

We cannot directly answer this question because the NSG transplant studies used BALB/cJ and C57BL/6J donor mice, whereas the strain specific expression of Lepr was in A/J mice. Leptin receptor is not expressed in NSG transplanted Kupffer cells from BALB/cJ or C57BL/6J mice. This is likely due to the monocytic origin of NSG transplant Kupffer cells. As discussed in the manuscript, RNA-seq data from two recent papers (Sakai et al., Immunity 2019, PMID 31587991; Seidman et al, Immunity 2020, PMID 32362324) show that the leptin receptor is not expressed in monocyte derived Kupffer cells (Figure 3f-g).

Figure 4 – Analysis of strain-specific transcriptomics differences of KCs in NSG transplant model
Recommend using UpSetPlots instead of Venn diagrams

As discussed in the response to Figure 2, we appreciate the recommendation of the UpSet plot as a mechanism for capturing the comparison of multiple gene sets as performed in this study. We have replaced the Venn diagrams in Figure 4g as suggested by Reviewer 3. Please see the discussion of Figure 4 revisions made in the response to Reviewer 1 for additional description of changes made.

Figure 5 – Analysis of ATACseq and H3K27a ChIPseq differences from an experiment in Figure 4
How are the enriched motifs from ChIPseq connected to the transcriptional differences ID'd in figure 4?
Are these genes (in 5d/e) the top-regulated genes in 4? Where are they in 4b/c?

We thank Reviewer 3 for pointing out the opportunity to integrate epigenomic data F1 more fully with the F1 RNA-seq data presented in Figure 4. We have provided two additional threads of analysis to address this comment.

First, we provide global analysis of our epigenetic and transcriptional data from the F1 and F0 mice to show that allelic bias in ATAC-seq data is correlated with allelic bias in H3K27Ac ChIP-seq data, and that allelic bias in promoter H3K27Ac ChIP-seq and ATAC-seq tags is associated with allelic bias in associated mRNA transcripts (Figure 5b, c, Lines 298-300, 315-319, 563-571). This analysis suggests that our motif analysis as shown in figure 5d has important implications for the regulation of strain specific transcription in the F0 Kupffer cells.

Secondly, as discussed in the response to Reviewer 1, our revised version of Figure 4 highlights distinct functions and distinct upstream regulators associated with C57BL/6J and BALB/cJ specific trans genes. In the revised version of Figure 5 we show that NFkB-p65 signaling is also enriched in C57BL/6J trans enhancers generally (Figure 5d), and specifically in trans enhancers associated with the C57BL/6J trans genes identified in Figure 4 (Figure 5e). Additionally, we provide browser tracks of representative trans regulated loci putatively controlled by strain

specific transcription factor activity (of NFkB in C57BL/6J and AP-1 in BALB/cJ, Figure 5f). Collectively, Figures 4 and 5 suggest that trans analysis of the transcriptome and epigenome can be used to identify specific differentially activated gene regulatory networks in Kupffer cells from inbred strains of mice.

Figure 6

Please define your differentially regulated genes and motifs from Figures 1-5 in this LPS model. Are they also differentially responding to LPS?

We now include a supplementary figure showing expression of BALB/cJ and C57BL/6J specific control genes from Figure 1 in the LPS model (Extended data Figure 7, Lines 940-943)). Overall, approximately 1/3 of strain specific genes retain their specificity 2 hours after treatment in LPS (Extended Data Figure 7a). For further comparison of genes expression across the different experiments in this study, we have provided TPM files from LPS treatment studies in GSE216164. We regret that Reviewer 3 was not able to access GSE216164 during the review process, as we provided the token in our submission documents. The token is now also included in the Resource Availability section in the revision Line 866.

We have also included supplementary figures and data illustrating motif enrichment analysis results following treatment with LPS. Extended Data Figure 7b shows results of known motif enrichment in BALB/cJ and C57BL/6J specific trans regulated peaks in the LPS treatment model (comparable to Figure 5d). We also included a list of the top 4 enriched de novo motifs in C57BL/6J and BALB/cJ specific accessible enhancers (both all enhancers and trans enhancer subsets) following treatment with LPS (Extended data Figure 7c-d). Lastly, we have added a supplementary table of known motif enrichment analysis for each of the core sets of peaks used in this study, to allow direct comparison of enrichment scores for the 426 default HOMER motifs across the conditions used in this study (Extended Data Table 6). Collectively, this highlights important insights comparing motif analysis results at homeostasis and under LPS and emphasizes the resource value of our study - as readers can explore changes in motif enrichment or expression through the supplemental data.

Methods

Why were DESeq 2 and HOMER chosen for transcriptomics analysis instead of limma and ComBat?

Limma and DESeq2 are similar analytical methods using slightly different normalization techniques. HOMER is used by this lab to generate sequencing count matrices from mapped fastq files in a manner analogous to featureCounts (PMID: 24227677). To our knowledge, this is not a feature of limma, ComBat, nor DeSeq2. Although featureCounts leads HOMER in citations counts (14,066 vs 9,820), the HOMER software suite is widely used in genomics analyses.

ComBat-seq is batch correction software for pooling samples across multiple temporal batches. However, we have no evidence of batch effects on initial principal component analysis that would necessitate such a correction (Extended Data Table 4, Extended Data Figure 3). We now provide an additional Extended Data Figure illustrating PCA by batch

demonstrating the dominant effect of strain on our data (see response to Reviewer 1 above).

Reviewers were not given access to GSE216164; thus it is not possible to confirm that all datasets generation during the study are available (please provide if a revision review is initiated)

We provided and confirmed a reviewer token for GSE216164 was included in submission documents for our manuscript (Nature Immunology Reporting Summary). As mentioned above, we are not sure why this information was not accessible to Reviewer 3, but the token is now also included here (ezqdkyiebrynpsz) and in the Resource Availability section in the revision at Line 866.

Extended Data Table 1 & 2 should be provided in the .csv format, not as PDFs

We have converted Extended Data Tables into .xlsx or .txt files.

All code for any analysis and generated figures should be provided to the public in Git repo; it is not sufficient to provide it upon request

*We appreciate the reviewer's commitment to open science and have endeavored to make the analysis in this manuscript broadly accessible. We have deposited the code used for analysis and figure generation within the following Github repository:
https://github.com/HunterBennett/KupfferCell_NaturalGeneticVariation.*

The processed data files necessary to perform analysis can be found at the following Zenodo repository: <https://zenodo.org/record/7829622>.

Your processed datasets should be deposited at figshare, specifically
Mapped read count files to mm10 and BALB/cJ genome builds for all transcriptomics and epigenomics
NicheNet object, and full model output files
DESeq2 objects, and full model output files

Mapped read count files are provided in GSE216164, which can be accessed as described above. We have added quality control summaries in Extended Data Table 4 as requested by Reviewer 1 and 3 and regret having not included these data in the initial submission. Regarding DESeq2 objects, we perform our differential gene analysis using the HOMER `getDifferentialExpression.pl` wrapper, and thus do not have direct access to the R object files. In lieu of this we have included an excel spreadsheet with DESeq2 results for comparisons assessed in the manuscript (Extended Data Table 5). Further, code necessary to reproduce our results has been posted to a public GitHub repository (https://github.com/HunterBennett/KupfferCell_NaturalGeneticVariation) and the processed data files necessary for analysis are publicly available on Zenodo (<https://zenodo.org/record/7829622>).

Discussion

What is the key story in the extensive data generated? A toy model or figure illustrating the key strain-specific differences in KCs would be helpful to the reader

We have prepared a model figure summarizing the trans regulated transcriptional networks identified in our study to improve audience access to our message (Extended Data Figure 9). This is an important addition, and we are grateful for the suggestion.

Decision Letter, first revision:

13th Jul 2023

Dear Chris,

Thank you for submitting your revised manuscript "Systematic analysis of transcriptional and epigenetic effects of genetic variation in Kupffer cells enables discrimination of cell intrinsic and environment-dependent mechanisms" (NI-A34682B). It has now been seen by the original referees and their comments are below. The reviewers find that the paper has improved in revision, and therefore we'll be happy in principle to publish it in Nature Immunology, pending minor revisions to satisfy the referees' final requests and to comply with our editorial and formatting guidelines.

We will now perform detailed checks on your paper and will send you a checklist detailing our editorial and formatting requirements in about a week. Please do not upload the final materials and make any revisions until you receive this additional information from us.

If you had not uploaded a Word file for the current version of the manuscript, we will need one before beginning the editing process; please email that to immunology@us.nature.com at your earliest convenience.

Thank you again for your interest in Nature Immunology Please do not hesitate to contact me if you have any questions.

Kind regards,

Laurie

Laurie A. Dempsey, Ph.D.
Senior Editor
Nature Immunology
l.dempsey@us.nature.com
ORCID: 0000-0002-3304-796X

Reviewer #1 (Remarks to the Author):

The revised manuscript demonstrates a significant improvement to the clarity and overall rigor of the study. Although the manuscript lacks mechanistic insight overall, particularly in relation to the effect of natural genetic variation on strain-specific liver disease, the descriptive findings and concepts reported here summarize a significant volume of work which will be of high utility for the

community.

The authors have performed sufficient additional analyses and experiments (such as the inclusion of F1-NSG hybrids) to strengthen the robustness of this work, and the expanded discussion section now positions future objectives in the context of their findings, which adds broader relevance to this research.

Some minor errors were introduced in the revised version of the manuscript and should be amended prior to publication. Specific points include but are not limited to:

- Typos in Expanded Data Figure 1 ("FDR adjusted p-value > 0.05" should be < 0.05)
- Line 213-214: use of "BALB/cJ specific" seems inaccurate to describe Fads1, which was identified in both A/J and BALB/cJ strains based on Figure 3c
- Line 355, 357, 358: Although the authors have improved the clarity of this section, the use of the terms "LPS responsive strain" and "LPS non-responsive strain" is still misleading
- Line 438: grammar
- Line 537 (in the legend of Figure 4b): it is unclear what is meant by 'left' and 'right'

Reviewer #2 (Remarks to the Author):

The authors have largely addressed the comments from the previous review, including adding the control of transplantation of F1 HSCs into NSG mice to examine cell-autonomous vs cell-non-autonomous trans effects. However, the text description of the chimeric mouse experiments (both new text and older text) is very confusing and needs to be clarified for the final version of the manuscript (see below). Otherwise, the paper is a valuable resource (and as another reviewer commented, could indeed be presented as a Resource article) that provides experimental and computational strategies to distinguish between cis vs trans effects of genetic variation on gene expression (and accessibility), and cell-autonomous vs non-cell-autonomous trans effects; the approach is still only semi-quantitative (see below), and perhaps these analysis limitations can be acknowledged in the Discussion.

(1) Text description of chimeric mouse experiments. The logic of the heatmap in Fig. 4h largely makes sense and is presented in terms of $\log_{FC}(\text{BALB}/\text{C57})$ (or resp. $\log_{FC}(\text{C57}/\text{BALB})$) in the original F0 data, in the F0 NSG data, and in the F1 NSG data (allelic imbalance in the NSG setting). But the text uses phrases like "genes exhibiting ... equivalent expression in donor-derived F1 NSG model" — presumably this means "no significant allelic imbalance", but the reader wonders "equivalent to what?". Other confusing points:

- "cell extrinsic/intrinsic" are terms used in the figure only, while cell-non-autonomous and cell-autonomous are used in the main text
- the description on the 71/121 and 42/121 genes in the text seems to contradict the heatmap? Seems like 71/121 genes are trans regulated only in the parental cells, but are not strain-differential (and therefore not trans regulated) in the NSG model?
- a trans-regulated gene by definition here refers to a gene with strain-differential expression for a pair of strains but no significant allelic imbalance in the corresponding F1 model; so "trans" is

associated with a pair of strains. It is therefore not clear what “trans-regulated in F1-hybrids” means - “NSG F0 specific trans genes”: this is a small set of genes, but the nomenclature seems counter-intuitive, as there is no strain-differential expression in the NSG F0 setting (based on Fig. 4h).

(2) Limitations of the quantitative approach. Here a strain-differential gene is described as trans-regulated if there is little allelic imbalance in F1 cells (which control for trans effects), but the definition is based on a cut-off; perhaps it is more accurate to say that all genes are regulated both by trans and cis effects to different degrees, but we do not have a model to quantify the relative impact of these effects using F0 and F1 data.

Typos:

- line 278-279: “more differential genes the F1-NSG model ...”
- line 299: “is positively correlated biases in both ATAC-seq ...”

Author Rebuttal, first revision:

Point-by-point response:

We appreciate the reviewers for their attention and comments on our revised manuscript. Of note, both reviewers found aspects of the manuscript to be unclear at times. We have given careful attention to these comments and edited the manuscript in hopes of improving clarity. Additional reviewer comments have been addressed in-line below:

Reviewer #1:

Remarks to the Author:

The revised manuscript demonstrates a significant improvement to the clarity and overall rigor of the study. Although the manuscript lacks mechanistic insight overall, particularly in relation to the effect of natural genetic variation on strain-specific liver disease, the descriptive findings and concepts reported here summarize a significant volume of work which will be of high utility for the community.

The authors have performed sufficient additional analyses and experiments (such as the inclusion of F1-NSG hybrids) to strengthen the robustness of this work, and the expanded discussion section now positions future objectives in the context of their findings, which adds broader relevance to this research.

We are grateful to reviewer 1 for the kind comments and attention to improving the quality of our reported research. The nature of the presented study is indeed descriptive, and our goal is that we and others can use the data and described framework to better define disease-relevant mechanisms for genetic variation in control of tissue macrophages.

Some minor errors were introduced in the revised version of the manuscript and should be amended prior to publication. Specific points include but are not limited to:

Corrected

- Typos in Expanded Data Figure 1 (“FDR adjusted p-value > 0.05” should be < 0.05)

Corrected

- Line 213-214: use of “BALB/cj specific” seems inaccurate to describe Fads1, which was identified in both A/J and BALB/cj strains based on Figure 3c

Corrected

- Line 355, 357, 358: Although the authors have improved the clarity of this section, the use of the terms “LPS responsive strain” and “LPS non-responsive strain” is still misleading

We have endeavored to improve this section further.

- Line 438: grammar

Corrected

- Line 537 (in the legend of Figure 4b): it is unclear what is meant by ‘left’ and ‘right’

Changed to correctly reflect the panel in the panel.

Reviewer #2:

Remarks to the Author:

The authors have largely addressed the comments from the previous review, including adding the control of transplantation of F1 HSCs into NSG mice to examine cell-autonomous vs cell-non-autonomous trans effects. However, the text description of the chimeric mouse experiments (both new text and older text) is very confusing and needs to be clarified for the final version of the manuscript (see below). Otherwise, the paper is a valuable resource (and as another reviewer commented, could indeed be presented as a Resource article) that provides experimental and computational strategies to distinguish between cis vs trans effects of genetic variation on gene expression (and accessibility), and cell-autonomous vs non-cell-autonomous trans effects; the approach is still only semi-quantitative (see below), and perhaps these analysis limitations can be acknowledged in the Discussion.

We thank reviewer 2 for the careful attention to our revision. We acknowledge the challenging nature of our results text for figure 4 and have revised the section for clarity. We are still in

agreement that our categorical intersections of differentially expressed genes from F0 and F1 mice is only semi-quantitative. To borrow your words, it is correct to state that “we do not have a model to quantify the relative impact of these effects using F0 and F1 data” and now appropriately acknowledge this limitation in the text.

(1) Text description of chimeric mouse experiments. The logic of the heatmap in Fig. 4h largely makes sense and is presented in terms of $\log_{2}FC(\text{BALB}/\text{C57})$ (or resp. $\log_{2}FC(\text{C57}/\text{BALB})$) in the original F0 data, in the F0 NSG data, and in the F1 NSG data (allelic imbalance in the NSG setting). But the text uses phrases like “genes exhibiting ... equivalent expression in donor-derived F1 NSG model” — presumably this means “no significant allelic imbalance”, but the reader wonders “equivalent to what?”. Other confusing points:

Indeed, the intent of our phrasing “equivalent expression in donor-derived F1 NSG model” is an absence of detectable “significant allelic imbalance.” We have adopted your recommended language.

- “cell extrinsic/intrinsic” are terms used in the figure only, while cell-non-autonomous and cell-autonomous are used in the main text

Thank you for pointing out this discrepancy. We have adjusted the figure text to match language in the main text.

- the description on the 71/121 and 42/121 genes in the text seems to contradict the heatmap? Seems like 71/121 genes are trans regulated only in the parental cells, but are not strain-differential (and therefore not trans regulated) in the NSG model?

The genes in the upper section of the heatmap differentially regulated in the parental cells but do not meet threshold criteria ($\log_{2}FC$ and p_{adj}) to be categorized as differentially regulated in parental cells or F1 cells recovered from NSG transplant models. Thus, we interpret this result to mean these genes were regulated through non-cell autonomous trans effects mediated by the environment of the NSG liver, as opposed to cis-acting effects of genetic variants.

- a trans-regulated gene by definition here refers to a gene with strain-differential expression for a pair of strains but no significant allelic imbalance in the corresponding F1 model; so “trans” is associated with a pair of strains. It is therefore not clear what “trans-regulated in F1-hybrids” means

Here our meaning is genes with significant allelic imbalance in F1-hybrids from NSG mice but not differential expression in parental cells transplanted into NSG mice. We recognize that this gene list is small and the biological consequence unclear.

- “NSG F0 specific trans genes”: this is a small set of genes, but the nomenclature seems counter-intuitive, as there is no strain-differential expression in the NSG F0 setting (based on Fig. 4h).

Our meaning for this definition was based on the following:

- *These genes were differentially regulated in parental Kupffer cells from the native mice*
- *These genes have significant allelic imbalance in F1 Kupffer cells recovered from transplanted NSG mice*
- *These genes do not have significant differences in expression in parental Kupffer cells recovered from transplanted NSG mice*

We defined these as “NSG F0 specific genes” from presumed environment-dependent regulation through trans-acting effects from the NSG hepatic environment. It is possible that these genes were mis-categorized due to the semi-quantitative approach and may result from a combination of cis- and trans-effects.

(2) Limitations of the quantitative approach. Here a strain-differential gene is described as trans-regulated if there is little allelic imbalance in F1 cells (which control for trans effects), but the definition is based on a cut-off; perhaps it is more accurate to say that all genes are regulated both by trans and cis effects to different degrees, but we do not have a model to quantify the relative impact of these effects using F0 and F1 data.

This is a valuable suggestion and we have modified our text with your recommended phrasing.

Typos:

- line 278-279: “more differential genes the F1-NSG model ...”

- line 299: “is positively correlated biases in both ATAC-seq ...”

Corrected, thank you for catching and sharing these mistakes with us.

Final Decision Letter:

Dear Chris,

I am delighted to accept your manuscript entitled "Discrimination of cell-intrinsic and environment-dependent effects of natural genetic variation on Kupffer cell epigenomes and transcriptomes" for publication in an upcoming issue of Nature Immunology.

Over the next few weeks, your paper will be copyedited to ensure that it conforms to Nature Immunology style. Once your paper is typeset, you will receive an email with a link to choose the appropriate publishing options for your paper and our Author Services team will be in touch regarding any additional information that may be required.

After the grant of rights is completed, you will receive a link to your electronic proof via email with a

request to make any corrections within 48 hours. If, when you receive your proof, you cannot meet this deadline, please inform us at rjsproduction@springernature.com immediately.

Please note that *Nature Immunology* is a Transformative Journal (TJ). Authors may publish their research with us through the traditional subscription access route or make their paper immediately open access through payment of an article-processing charge (APC). Authors will not be required to make a final decision about access to their article until it has been accepted. [Find out more about Transformative Journals](https://www.springernature.com/gp/open-research/transformative-journals).

Your paper will be published online soon after we receive your corrections and will appear in print in the next available issue. Content is published online weekly on Mondays and Thursdays, and the embargo is set at 16:00 London time (GMT)/11:00 am US Eastern time (EST) on the day of publication. Now is the time to inform your Public Relations or Press Office about your paper, as they might be interested in promoting its publication. This will allow them time to prepare an accurate and satisfactory press release. Include your manuscript tracking number (NI-A34682C) and the name of the journal, which they will need when they contact our office.

About one week before your paper is published online, we shall be distributing a press release to news organizations worldwide, which may very well include details of your work. We are happy for your institution or funding agency to prepare its own press release, but it must mention the embargo date and *Nature Immunology*. Our Press Office will contact you closer to the time of publication, but if you

or your Press Office have any enquiries in the meantime, please contact press@nature.com.

Also, if you have any spectacular or outstanding figures or graphics associated with your manuscript - though not necessarily included with your submission - we'd be delighted to consider them as candidates for our cover. Simply send an electronic version (accompanied by a hard copy) to us with a possible cover caption enclosed.

Please note that we encourage the authors to self-archive their manuscript (the accepted version before copy editing) in their institutional repository, and in their funders' archives, six months after publication. Nature Portfolio recognizes the efforts of funding bodies to increase access of the research they fund, and strongly encourages authors to participate in such efforts. For information about our editorial policy, including license agreement and author copyright, please visit www.nature.com/ni/about/ed_policies/index.html

Kind regards,

Laurie

Laurie A. Dempsey, Ph.D.

Senior Editor
Nature Immunology
l.dempsey@us.nature.com
ORCID: 0000-0002-3304-796X